# UNDERSTANDING HETEROPHILY FOR GRAPH NEURAL NETWORKS

## ABSTRACT

Graphs with heterophily have been regarded as challenging scenarios for Graph Neural Networks (GNNs), where nodes are connected with dissimilar neighbors through various patterns. In this paper, we present theoretical understandings of the impacts of different heterophily patterns for GNNs by incorporating the graph convolution (GC) operations into fully connected networks via the proposed Heterophilous Stochastic Block Models (HSBM), a general random graph model that can accommodate diverse heterophily patterns. Firstly, we show that by applying a GC operation, the separability gains are determined by two factors, i.e., the Euclidean distance of the neighborhood distributions and $\sqrt{\mathbb{E}[\deg]}$, where $\mathbb{E}[\deg]$ is the averaged node degree. It reveals that the impact of heterophily on classification needs to be evaluated alongside the averaged node degree. Secondly, we show that the topological noise has a detrimental impact on separability, which is equivalent to degrading $\mathbb{E}[\deg]$. Finally, when applying multiple GC operations, we show that the separability gains are determined by the normalized distance of the $l$-powered neighborhood distributions. It indicates that the nodes still possess separability as $l$ goes to infinity in a wide range of regimes. Extensive experiments on both synthetic and real-world data verify the effectiveness of our theory.

## 1 INTRODUCTION

Graph Neural Networks (GNNs) have demonstrated remarkable superiority in processing the graph-structured data (Kipf & Welling, 2017; Velickovic et al., 2018; Hamilton et al., 2017; Xu et al., 2019; Gilmer et al., 2017; Wang et al., 2021). Unfortunately, the majority of GNNs are designed for the homophilous scenario, where the connected nodes usually possess similar features or labels. When this homophilous assumption is not satisfied, i.e., in the heterophilous scenario, typical GNNs may fail to handle the graphs and even perform worse than a fully connected network (Pei et al., 2020; Zhu et al., 2020). To tackle this heterophily problem, several heterophily-specific GNNs are proposed, by developing adaptive aggregation schemes for the similar/dissimilar neighbors (Pei et al., 2020; Bo et al., 2021; Suresh et al., 2021; He et al., 2022; Ma et al., 2022a; Li et al., 2023) or excavating appropriate information from the high-ordered/non-local neighbors (Zhu et al., 2020; Chien et al., 2021; Liu et al., 2021; Yang et al., 2021; Jin et al., 2021; Song et al., 2023).

Unfortunately, according to the recent studies (Ma et al., 2022b; Luan et al., 2022), which have revisited the impact of heterophily on typical GNNs, heterophily may not be harmful to the classification results. On the graphs with certain heterophily patterns, a typical GNN, e.g., Graph Convolutional Network (GCN) (Kipf & Welling, 2017), can achieve competitive or even superior performances compared to the well-designed heterophily-specific GNNs (Ma et al., 2022b; Platonov et al., 2023).

This observation raises a pivotal question: *What types of heterophily patterns in graphs are beneficial or harmful to typical GNNs for their performances on multi-class node classifications*? Accordingly, Ma et al. (2022b); Yan et al. (2022) present a preliminary analysis on the binary classification task. Ma et al. (2022b) empirically observes that GCN can effectively handle the graphs, whose nodes with the same label possess similar neighborhood distributions. Inspired by the gradient computed via SGC(Wu et al., 2019), Luan et al. (2022) introduces a post-aggregation node similarity metric to assess the impact of heterophily. Despite their efforts, the fundamental nature of this issue remains inadequately explored. Therefore, it is crucial to thoroughly investigate:

*How do different heterophily patterns contribute to the multi-class node classifications?*

In this paper, we theoretically investigate this question on the random graph models, which are widely utilized to analyze the behavior of GNNs for various vital problems (Baranwal et al., 2021; Keriven et al., 2021; Wei et al., 2022; Baranwal et al., 2023; Wu et al., 2023). Since the conventional Erdős–Rényi Model (Erdős et al., 1960) and Stochastic Block Model (Holland et al., 1983) sample edges uniformly or only differentiate the inter-/intra-category edges, they lack the capacity to effectively model various heterophily patterns. To address this limitation, we propose a more general model, named Heterophilous Stochastic Block Models (HSBM). In HSBM, edges are sampled based on the blocks/classes of the connected nodes. Thus, nodes within the same class exhibit class-specific neighborhood distributions. This feature allows HSBM to accommodate diverse heterophily patterns. Besides, our proposed HSBM incorporates node-wise neighborhood distribution perturbations, which simulate the topology noises in real-world graphs.

Based on our HSBM, we construct an analytical framework to explore the impacts of GNNs on graphs with diverse heterophily patterns. We analyze the variations of the separability between each pair of classes, by applying the graph aggregator operations. Specifically, we utilize GCN (Kipf & Welling, 2017), one of the most popular GNNs, in our illustrations. Our theoretical results can be summarized as follows.

1. For the impact of a Graph Convolution (GC) operation, we demonstrate that the separability gain of each pair of classes is determined by two factors, i.e., the Euclidean distance of their respective neighborhood distributions and the square root of the averaged degree $\sqrt{\mathbb{E}\left[\deg\right]}$. This result indicates that the effect of the heterophily pattern on classification requires to be evaluated based on $\sqrt{\left[\mathbb{E}\deg\right]}$, e.g., very similar neighborhood distributions may boost the classification if the average node degree is large. Based on this analysis, we can form the categories of good/mixed/bad heterophily patterns.

2. For the impact of the topological noise on separability, we reveal that this impact is detrimental. When a Gaussian noise is employed, this impact is equivalent to reducing the averaged degree by a factor of $1/(1 + r\delta^2)$, where $r > 0$ is a constant and $\delta^2$ is the variance of the topological noise. This result indicates that for a specific heterophily pattern, enhancing the node-wise neighborhood distribution perturbations will result in a decrement in separability, similar to reducing the node degrees.

3. For the impact of multiple GC operations along with a stronger density assumption, we demonstrate that the separability gains are determined by the normalized distance of the $l$-powered neighborhood distributions. This result suggests that the nodes still possess separability in various regimes, even when *over-smoothing* occurs. However, as $l$ approaches infinity, though the relative differences between the nodes are still maintained, their absolute values become exponentially smaller. Therefore, the classification accuracy eventually decreases, due to the *precision limitations* of the floating-point data formats.

Our theoretical results are verified via extensive experiments on both the synthetic and real-world data, and they may serve for two practical utilizations. Firstly, when GNNs are integrated into specific applications, our theoretical insights can offer guidance to the construction of their graphs and help to understand the performance with their specific heterophily patterns. Secondly, our work introduces a fresh perspective on heterophily and over-smoothing, coupled with a novel GNN analytical framework, which may further boost the development of innovative methods.

## 2 PRELIMINARIES

**Notations.** Let $\mathcal{G} = (\boldsymbol{A}, \boldsymbol{X})$ be an attributed graph, where $\boldsymbol{A} = [a_{ij}] \in \{0, 1\}^{n \times n}$ denotes the adjacency matrix and $\boldsymbol{X} \in \mathbb{R}^{n \times d}$ represents the node features. $n$ and $d$ represent the numbers of nodes and node features, respectively. $a_{ij}$ is 1 iff there is an incoming edge from node $j$ to node $i$, or 0 otherwise. $\boldsymbol{D}$ represents the diagonal degree matrix corresponding to the adjacency matrix $\boldsymbol{A}$, where $D_{ii} = \sum_j a_{ij}$. $[n] = \{0, 1, 2, \cdots, n-1\}$ is denoted as the set of nodes. For each node $i \in [n]$, $\Gamma_i = \{j | a_{ij} = 1\}$ represents the set of its neighbors. For a node classification task, $\boldsymbol{Y} = [y_i] \in [c]^n$ represents the labels of nodes and $c$ is the number of classes.

**Graphs with Heterophily.** The metrics the homophily/heterophily measures the fraction of intra-class edges within different scopes (Pei et al., 2020; Zhu et al., 2020; Lim et al., 2021). Typically, the

homophily ratio can be defined from the node-level, i.e., $\mathcal{H}(\mathcal{G}) = \frac{1}{n}\sum_{i\in[n]}\frac{\sum_{j\in\Gamma_i}(y_i=y_j)}{D_{ii}}$, where $\mathcal{H}(\mathcal{G}) \in [0,1]$ (Pei et al., 2020). A high homophily ratio indicates that the graph is with strong homophily, while a graph with strong heterophily has a small homophily ratio. However, the homophily ratio of a certain heterophily pattern does not align with its impacts on node classifications. It is crucial to delve deeper into the underlying mechanisms of how heterophily functions.

**Graph Convolutional Network.** Graph Convolutional Network (GCN) (Kipf & Welling, 2017), known as one of the most popular GNN, is employed to analyze the impact of different heterophily patterns in our work. In a graph convolutional layer, nodes updates their representations via

$$\boldsymbol{X}^{(l+1)} = \sigma\left(\tilde{\boldsymbol{A}}\boldsymbol{X}^{(l)}\boldsymbol{W}^{(l)}\right), \tag{1}$$

where $\boldsymbol{W}^{(l)} \in \mathbb{R}^{d_{in}^{(l)}\times d_{out}^{(l)}}$ is the parameter matrix of layer $l$, $\tilde{\boldsymbol{A}} = \boldsymbol{D}^{-1}\boldsymbol{A}$, $\boldsymbol{A}$ is the adjacency matrix containing self-loops, and $\boldsymbol{X}^{(0)} = \boldsymbol{X}$. For simplicity, the non-linear function $\sigma(\cdot)$ is neglected in our analysis, as is widely adopted (Baranwal et al., 2021; Wu et al., 2023).

**Contextual Stochastic Block Models.** Contextual Stochastic Block Models (CSBMs) (Deshpande et al., 2018) are typical random graph models, where nodes are randomly sampled by class-specific Gaussian distribution and edges are sampled by Bernoulli distribution, with a parameter of $p$ if the two connected nodes belongs to the same class, or $q$ otherwise. It is extensively employed to analyze the theoretical performance of Graph Neural Networks (Baranwal et al., 2021; Keriven et al., 2021; Wei et al., 2022; Baranwal et al., 2023; Wu et al., 2023). Specifically, for the heterophily issue, Ma et al. (2022b) presents a preliminary result of GCN regarding the impact of different $p$ and $q$ on the two-block CSBMs, while Luan et al. (2023) analyzes the performance of GNNs with low-pass/high-pass filters. Mao et al. (2023) analyzes on the graphs containing nodes with homophily and heterophily patterns simultaneously. However, these efforts only focus on the simple binary node classifications, which is not general enough to analyze diverse heterophily patterns. The impact of complicated heterophily patterns in the multi-class classifications has not been analyzed.

## 3 PROPOSED DATA MODELS

In this section, we propose more general data models compared with CSBM, named Heterophilous Stochastic Block Models (HSBM), to accommodate diverse heterophily patterns.

In HSBM, each node $i \in [n]$ is independently sampled from $c$ blocks/classes, with probability of $\boldsymbol{\eta} = (\eta_0, \cdots, \eta_{c-1})$, where $\eta_k > 0$ and $\sum_k \eta_k = 1$. The class of node $i$ is denoted as $\varepsilon_i$. Each edge, connecting nodes $i$ and $j$, is sampled by a Bernoulli distribution, with a parameter $m_{\varepsilon_i\varepsilon_j}$. The collected $\mathbf{M} = [m_{ij}]$ is the probability matrix which represents the edge generation probabilities between different classes. For each class $k \in [c]$, $\bar{p}_k = \sum_{t\in[c]}\eta_t m_{kt}$ is denoted as its averaged edge generation probability. Then, $\bar{D}_k = n\bar{p}_k$ is the averaged node degree within class $k$. Denoting $\hat{m}_{kt} = \eta_t m_{kt}/\bar{p}_k$, $\hat{\mathbf{m}}_k = (\hat{m}_{k0}, \hat{m}_{k1}, \cdots, \hat{m}_{k(c-1)})$ is the neighborhood distribution of the nodes within class $k$, where $\hat{m}_{kt}$ represents the proportion of nodes within class $t$ in the neighborhood of a node belonging to class $k$. The collected matrix $\hat{\mathbf{M}} = [\hat{m}_{ij}]$ is denoted as the neighborhood distribution matrix. Then, by choosing different edge generation probability matrix $\mathbf{M}$, we can generate graphs with different patterns of heterophily, i.e., $\hat{\mathbf{M}} = [\hat{m}_{ij}]$.

**Topological Noise.** In real-world networks, the neighborhood distributions of nodes within the same class are center around a specific distribution as well as possess certain perturbation. Inspired by this observation, a random node-wise topological noise is introduced in our HSBM. Specifically, for each node $i$, its incoming edges are generated according to the distribution $\hat{\mathbf{m}}(i) = \hat{\mathbf{m}}_{\varepsilon_i} + \boldsymbol{\Delta}_i$, where $\boldsymbol{\Delta}_i$ is the random noise. Note that in practice, the generated distribution should be legal, i.e., $[\hat{\mathbf{m}}(i)]_j > 0$ and $\sum_j[\hat{\mathbf{m}}(i)]_j = 1$.

**Gaussian Node Features.** Since our intention is to study the effects of topology within different heterophily patterns, following CSBM, the Gaussian node features are employed. Specifically, in each block/class $k \in [c]$, the features of each node are generated by Gaussian distributions, i.e., $\mathcal{N}(\mathbf{x}; \boldsymbol{\mu}_k, \sigma^2\mathbf{I})$. For simplicity, $\boldsymbol{\mu}_k$ is orthometric to each other and has the same length, i.e., for any $k, t \in [c]$ and $k \neq t$, we have $\boldsymbol{\mu}_k^T\boldsymbol{\mu}_t = 0$ and $||\boldsymbol{\mu}_k||_2 = ||\boldsymbol{\mu}_t||_2$. $\gamma = ||\boldsymbol{\mu}_k - \boldsymbol{\mu}_t||_2$ is denoted as the distance between two Gaussian means. Finally, denote $(\boldsymbol{X}, \boldsymbol{A}) = \text{HSBM}(n, c, \sigma, \{\boldsymbol{\mu}_k\}, \boldsymbol{\eta}, \mathbf{M}, \{\boldsymbol{\Delta}_i\})$ as the graph data generated by our HSBM.

## 3.1 ASSUMPTIONS.

Our analysis is constructed based on the following two gentle assumptions.

**Assumption 1.** $\bar{p}_k \asymp \bar{p}_t$, $\forall k, t \in [c]$.

**Assumption 2.** $\bar{p}_k = \omega \left( \log^2 n / n \right)$, $\forall k \in [c]$.

Assumption 1 claims that nodes within different classes possess similar averaged degree. Assumption 2 states that the graph is not too sparse and enable some numerical properties of graphs, e.g., the number of nodes within each class and node degree concentrate to their expectations. A detailed discussion of these two assumptions is provided in Appendix A.1.

## 4 THEORETICAL RESULTS

Our analysis is established on the multi-class node classifications. Its primary objective is to accurately categorize samples into their respective classes while distinguishing them from samples belonging to other classes. To achieve this, a classifier is required to establish a total of $\binom{c}{2}$ classification boundaries to separate samples from different categories. By minimizing the error rate, the optimal classification boundaries are established based on posterior probabilities. Specifically, $\forall k, t \in [c]$ with $k < t$, the condition $\mathbb{P}[y = k \mid \mathbf{x}] = \mathbb{P}[y = t \mid \mathbf{x}]$ is established as the optimal classification boundary, as constructed by the Bayes classifier.

In this section, we develop the analytical framework and present our theoretical results. We begin by defining the two-class separability for each ideal boundary and then proceed to analyze the variation of separability when applying GC operations on graphs with different patterns of heterophily. Note that although our analysis is based on the ideal Bayes classifier, the analyzed results can be achieved through MLPs, as detailed in Appendix D. All the proofs can be found in Appendix A.

## 4.1 SETTING UP THE BASELINE

For each node $i \in [n]$, $\zeta_i(k)$ is denoted as the event where $\mathbb{P}[y = \varepsilon_i \mid \boldsymbol{X}_i] \geq \mathbb{P}[y = k \mid \boldsymbol{X}_i]$, which indicates whether node $i$ is more likely to classified into its ground truth class $\varepsilon_i$ rather than into class $k$. Then, the two-classes separability between classes $t$ and $k$ is defined.

**Definition 1.** $\forall t, k \in [c]$ and $t \neq k$, the separability between class $t$ and class $k$ is defined as

$$S(t, k) = \frac{\eta_t}{\eta_t + \eta_k} \mathbb{E}_{i \in \mathcal{C}_t} \mathbb{P}\left[\zeta_i(k)\right] + \frac{\eta_k}{\eta_t + \eta_k} \mathbb{E}_{i \in \mathcal{C}_k} \mathbb{P}\left[\zeta_i(t)\right], \tag{2}$$

*where $\mathbb{E}_{i \in \mathcal{C}_t} \mathbb{P}\left[\zeta_i(k)\right]$ represents the fraction of nodes within class $t$ can be successfully classified regrading to class $k$.*

The two-classes separability $S(t, k)$ can be regarded as the expected accuracy when only considering the classification among nodes within these two classes. It assesses the effectiveness of the boundary $\mathbb{P}[y = k \mid \mathbf{x}] = \mathbb{P}[y = t \mid \mathbf{x}]$ for the corresponding two-classes subtask. Apparently, $S(t, k) = S(k, t)$ holds for Eq. (2). Besides of its impact on the two-class subtask, we claim that it also holds significance for the overall classification by setting bounds for the minimum error rate, as illustrated in Appendix E. Then, we proceed to establish the separability of the raw node features $\boldsymbol{X}$.

**Theorem 1.** *Given $(\boldsymbol{X}, \boldsymbol{A}) = \mathrm{HSBM}\left(n, c, \sigma, \{\boldsymbol{\mu}_k\}, \boldsymbol{\eta}, \mathbf{M}, \{\boldsymbol{\Delta}_i\}\right)$, we get the following properties over data $\boldsymbol{X}$.*

1. *$\forall t, k \in [c]$ and $t \neq k$,*

$$\mathbb{E}_{i \in \mathcal{C}_t} \mathbb{P}\left[\zeta_i(k)\right] = \Phi\left(\frac{\gamma}{2\sigma} + \frac{\sigma}{\gamma} \ln\left(\frac{\eta_t}{\eta_k}\right)\right), \tag{3}$$

   *where $\Phi(\cdot)$ is the cumulative distribution function of the standard Gaussian distribution.*

2. *Without loss of generality, let $\eta_t \geq \eta_k$. Then, $S(t, k)$ is an increasing function with respect to both $\frac{\gamma}{\sigma}$ and $\frac{\eta_t}{\eta_k}$.*

The formula of $\mathbb{E}_{i \in \mathcal{C}_t} \mathbb{P}[\zeta_i(k)]$ is presented in Theorem 1, which represents the fraction of nodes within class $t$ that can be successfully classified with respect to class $k$. When $\eta_t = \eta_k$, i.e., nodes within the two classes have the same proportion, we obtain that $\mathbb{E}_{i \in \mathcal{C}_t} \mathbb{P}[\zeta_i(k)] = \mathbb{E}_{i \in \mathcal{C}_k} \mathbb{P}[\zeta_i(t)] = \Phi\left(\frac{\gamma}{2\sigma}\right)$. Notably, both $\mathbb{E}_{i \in \mathcal{C}_t} \mathbb{P}[\zeta_i(k)]$ and $\mathbb{E}_{i \in \mathcal{C}_k} \mathbb{P}[\zeta_i(t)]$ are positively correlated with $\frac{\gamma}{\sigma}$. However, in more general cases, where the number of nodes within two classes are imbalanced, i.e., $\eta_t \neq \eta_k$, the situations become more complex. W.l.o.g, let $\eta_t \geq \eta_k$. According to Eq. (3), as $\frac{\gamma}{\sigma}$ increases, the $\mathbb{E}_{i \in \mathcal{C}_t} \mathbb{P}[\zeta_i(k)]$ exhibits a larger increment compared to $\mathbb{E}_{i \in \mathcal{C}_k} \mathbb{P}[\zeta_i(t)]$. Specifically, When $\frac{\gamma}{\sigma}$ increases from a small value, the latter may even decrease. This indicates that as $\frac{\gamma}{\sigma}$ increases, classes with a larger number of samples experience more significant benefits, while classes with fewer samples derive fewer benefits and may even encounter a decrease in the classification accuracy. Furthermore, the second part of Theorem 1 claims that the separability $S(t, k)$, which represents the overall impact of $\mathbb{E}_{i \in \mathcal{C}_t} \mathbb{P}[\zeta_i(k)]$ and $\mathbb{E}_{i \in \mathcal{C}_k} \mathbb{P}[\zeta_i(t)]$, consistently increases as $\frac{\gamma}{\sigma}$ increases. Hence, we can assess the impact of GC operations by examining the variation of $\frac{\gamma}{\sigma}$.

## 4.2 IMPACT OF HETEROPHILY FOR GRAPH CONVOLUTION

Subsequently, we incorporate a Graph Convolution (GC) operation and analyze the separability of the aggregated features.

**Theorem 2.** *Given $(\boldsymbol{X}, \boldsymbol{A}) = \mathrm{HSBM}\left(n, c, \sigma, \{\boldsymbol{\mu}_k\}, \boldsymbol{\eta}, \mathbf{M}, \{\mathbf{0}\}\right)$, for each categories $t, k \in [n]$ and $t \neq k$, with probability at least $1 - 1/\mathrm{Ploy}(n)$, we have*

$$\mathbb{E}_{i \in \mathcal{C}_t} \mathbb{P}[\zeta_i(k)] = \Phi\left(\frac{\gamma}{2\sigma} \frac{F_{tk}}{\varsigma_n} + \frac{\sigma}{\gamma} \frac{\varsigma_n}{F_{tk}} \ln\left(\frac{\eta_t}{\eta_k}\right)\right), \tag{4}$$

*over the aggregated features $\tilde{\boldsymbol{X}} = \boldsymbol{D}^{-1} \boldsymbol{A} \boldsymbol{X}$, where $F_{tk} = \frac{1}{\sqrt{2}} \|\sqrt{\bar{D}_k} \hat{\mathbf{m}}_k - \sqrt{\bar{D}_t} \hat{\mathbf{m}}_t\|$, $\varsigma_n = 1 \pm o_n(1)$, and $o_n(1)$ is the error term.*

Theorem 2 illustrates the impact of the GC operation on the two-classes separability. Compared with the formula of $\mathbb{E}_{i \in \mathcal{C}_t} \mathbb{P}[\zeta_i(k)]$ in Theorem 1, its effect is equivalent to scaling the $\frac{\gamma}{\sigma}$ by a factor of $\frac{F_{tk}}{\varsigma_n}$, with high probability. Therefore, according to the monotonicity of $S(t, k)$, when $F_{tk} > \varsigma_n$, the GC operation enhances the separability $S(t, k)$; otherwise, it degrades the separability. Here, $o_n(1)$ is an error item introduced by sampling, which can be neglected as $n$ goes to sufficient large. Therefore, $F_{tk}$ is the separability gains of a GC operation, which is determined by the Euclidean distance of the class-specific degree times neighborhood distribution. Specifically, according to Assumption 1, we can assume that nodes within different classes have approximately the same degree, denoted as $\bar{D} \approx \bar{D}_k$. Then, $F_{tk}$ becomes $\sqrt{\bar{D}/2}\|\hat{\mathbf{m}}_k - \hat{\mathbf{m}}_t\|$, which indicates that the average node degree and the distance between neighborhood distributions possess complementary effects on the separability gains. For example, even though nodes within class $t$ and $k$ possess similar (not the same) neighborhood distributions, this heterophily pattern may also boost the classification with a large average node degree. Therefore, we can conclude that whether a heterophily pattern has a positive/negative impact for the classification should be evaluated along with the node degree.

Based on this analysis, we can form the categories of good/mixed/bad heterophily patterns.

**Definition 2.** *Given $(\boldsymbol{X}, \boldsymbol{A}) = \mathrm{HSBM}\left(n, c, \sigma, \{\boldsymbol{\mu}_k\}, \boldsymbol{\eta}, \mathbf{M}, \{\mathbf{0}\}\right)$, a good heterophily pattern is where*

$$\min_{k \neq t} \frac{1}{\sqrt{2}} \|\sqrt{\bar{D}_k} \hat{\mathbf{m}}_k - \sqrt{\bar{D}_t} \hat{\mathbf{m}}_t\| > \varsigma_n, \tag{5}$$

*and a bad heterophily pattern is where*

$$\max_{k \neq t} \frac{1}{\sqrt{2}} \|\sqrt{\bar{D}_k} \hat{\mathbf{m}}_k - \sqrt{\bar{D}_t} \hat{\mathbf{m}}_t\| < \varsigma_n, \tag{6}$$

*where $\varsigma_n = 1 \pm o_n(1)$ and $o_n(1)$ is the error term. Otherwise, we get a mixed heterophily pattern.*

When applying a GC operation, good heterophily patterns improve the separability of every pair of classes, consequently boosting the overall performance. On the contrary, bad heterophily patterns decrease the separability of all pairs of classes, resulting in lower performances. Besides, mixed heterophily patterns possess varying impacts on the separability of class pairs, enhancing it for some pairs while reducing it for others, leading to a mixed impact on the separability.

### 4.3 IMPACT OF TOPOLOGICAL NOISE

In real-world graphs, nodes belonging to the same class typically exhibit similar but slightly different neighborhood distributions. Intuitively, the greater the magnitude of these differences is, the more challenging to process the graph becomes. To assess the impact of this perturbation, we introduce the topological noise $\boldsymbol{\Delta}_i$ for each node $i \in [n]$ within our HSBM.

**Theorem 3.** *Given* $(\boldsymbol{X}, \boldsymbol{A}) = \mathrm{HSBM}\,(n, c, \sigma, \{\boldsymbol{\mu}_k\}, \boldsymbol{\eta}, \mathbf{M}, \{\boldsymbol{\Delta}_i\})$, *with each topological noise* $\boldsymbol{\Delta}_i$ *independently sampled from* $\mathcal{N}\,(\mathbf{0}, \delta\mathbf{I})$, *assuming that* $d = c$ *and* $\forall k \in [c], \bar{D} = \bar{D}_k$, *with probability at least* $1 - 1/\mathrm{Ploy}\,(n)$, *we have*

$$\mathbb{E}_{i \in \mathcal{C}_t} \mathbb{P}\left[\zeta_i(k)\right] = \Phi\left(\frac{\gamma}{2\sigma}\frac{F'_{tk}}{\varsigma_n} + \frac{\sigma}{\gamma}\frac{\varsigma_n}{F'_{tk}}\ln\left(\frac{\eta_t}{\eta_k}\right)\right) \tag{7}$$

*over data* $\tilde{\boldsymbol{X}} = \boldsymbol{D}^{-1}\boldsymbol{A}\boldsymbol{X}$, *where* $F'_{tk} = \frac{1}{\sqrt{2}}\|\frac{1}{\rho_k}\hat{\mathbf{m}}_k - \frac{1}{\rho_t}\hat{\mathbf{m}}_t\|$, $\rho_t = \sqrt{\frac{\gamma^2\delta^2}{2\sigma^2} + \frac{1}{\bar{D}_t}}$, *and* $\varsigma_n = 1 \pm o_n(1)$.

Theorem 3 shows that topological noise has a detrimental impact on separability, which is equivalent to degrading the average degree by a factor of $\frac{1}{1+r\delta^2}$, where $r = \frac{\gamma^2\bar{D}_k}{2\sigma^2} > 0$. It indicates that for the specific heterophily patterns, a larger $\delta$ results in a more significant reduction in separability. Note that we can directly generalize Definition 2 by utilizing $F'_{tk}$ when considering the topological noise.

### 4.4 IMPACT OF STACKING MULTIPLE GRAPH CONVOLUTIONS

As the number of GC operations increases, the over-smoothing dilemma (Li et al., 2018; Oono & Suzuki, 2020) indicates that the features of different nodes tend to converge to the same values, thereby losing their separability. In this subsection, we present the separability gains of stacking multiple GC operations and introduce a novel and different insight: nodes can still maintain their separability, even in the presence of over-smoothing.

**Theorem 4.** *Given* $(\boldsymbol{X}, \boldsymbol{A}) = \mathrm{HSBM}\,(n, c, \sigma, \{\boldsymbol{\mu}_k\}, \boldsymbol{\eta}, \mathbf{M}, \{\mathbf{0}\})$, *assume that* $\forall k, t \in [c], \bar{D} = \bar{D}_t$, $\eta_t = \eta_k$, $\sum_t m_{tk} \asymp \sum_t m_{kt}$, *and* $\sum_t m_{\varepsilon_i t}m_{t\varepsilon_j} = \omega(\log n^2/n)$. *When each class possesses exactly* $\frac{n}{c}$ *nodes, with probability of at least* $1 - 1/\mathrm{Ploy}\,(n)$, *we have*

$$\mathbb{E}_{i \in \mathcal{C}_t} \mathbb{P}\left[\zeta_i(k)\right] = \Phi\left(\frac{\gamma}{2\sigma}\frac{F^{(l)}_{tk}}{\varsigma^{(l)}_n} + \frac{\sigma}{\gamma}\frac{\varsigma^{(l)}_n}{F^{(l)}_{tk}}\ln\left(\frac{\eta_t}{\eta_k}\right)\right), \tag{8}$$

*over data* $\boldsymbol{X}^{(l)} = \left(\boldsymbol{D}^{-1}\boldsymbol{A}\right)^l \boldsymbol{X}$, *where*

$$F^{(l)}_{tk} = \sqrt{\frac{c\|\hat{\mathbf{m}}^{(l)}_k - \hat{\mathbf{m}}^{(l)}_t\|^2_2}{\sum_{k_1, k_2 \in [c]} \|\hat{\mathbf{m}}^{(l)}_{k_1} - \hat{\mathbf{m}}^{(l)}_{k_2}\|^2_2} \frac{\bar{D}}{\log n}} \tag{9}$$

*for* $l > 1$ *and* $\varsigma^{(l)}_n = 1 \pm lo_n(1)$.

Different from the previous analysis (Wu et al., 2023), Theorem 4 is obtained by fully considering the correlation of the aggregated node features. It requires a stronger density assumption, i.e., $\bar{p}_k = \omega\left(\log n/\sqrt{n}\right)$, obtained according to $\sum_t m_{\varepsilon_i t}m_{t\varepsilon_j} = \omega(\log n^2/n)$, while a more general version of Theorem 4 with less assumptions is provided in Theorem A1.

Theorem 4 claims that when applying $l$ GC operations, the separability gains $F^{(l)}_{tk}$ is determined by the normalized distance of the $l$-powered neighborhood distributions. Although the distance $\|\hat{\mathbf{m}}^{(l)}_k - \hat{\mathbf{m}}^{(l)}_t\|$ tends to converge to 0 as $l$ goes to infinity, its normalized version can also possess a large values through the normalization term.

**Proposition 1.** *When* $\hat{\mathbf{M}}$ *is non-singular, the approximation of* $F^{(l)}_{tk}$ *in Eq. (112) is always larger than 0 and* $\sum_{t,k} F^{(l)}_{tk} > \sqrt{c}\bar{D}/\log n$.

Specifically, Proposition 1 further claims that there exists various regimes that the separability gains never become 0, i.e., when $\hat{\mathbf{M}}$ is non-singular, $F^{(l)}_{tk} > 0$ holds for every $l$. It reveals that the separability of each pair of classes is larger than $\frac{1}{2}$, i.e., the nodes exhibit a certain degree of separability.

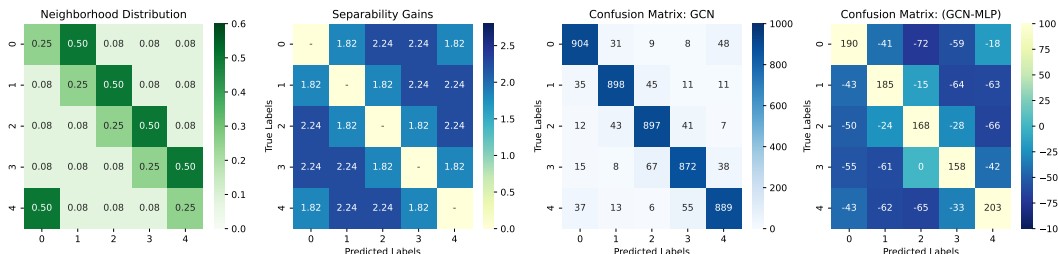

(a) Example of good heterophily pattern with $a = 0.25$. The accuracy of GCN is 89.20.

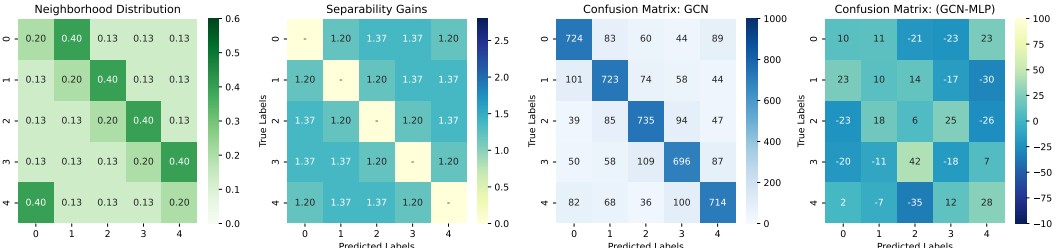

(b) Example of mixed heterophily pattern with $a = 0.2$. The accuracy of GCN is 71.84.

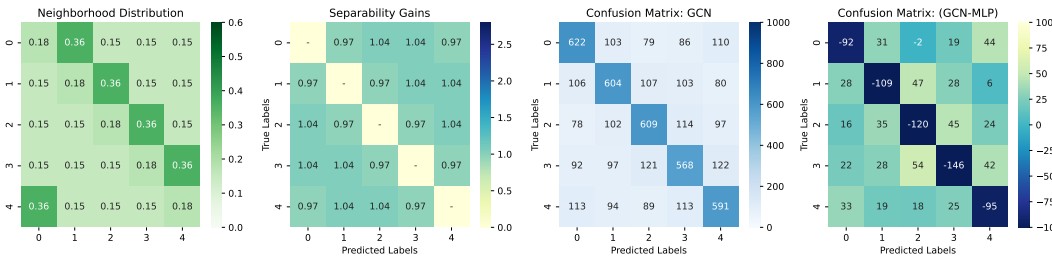

(c) Example of bad heterophily pattern with $a = 0.18$. The accuracy of GCN is 59.88.

Figure 1: Examples of different heterophily patterns. The accuracy of MLP is 71.12.

Besides, $\sum_{t,k} F_{tk}^{(l)} > \sqrt{c}\bar{D}/\log n$ further depicts the separability of at least one pair of classes never becomes to $\frac{1}{2}$, which indicates that nodes still possess certain separability as $l$ goes to infinity.

However, although the relative differences between the nodes persist, as $l$ increases, their absolute values become exponentially smaller (Oono & Suzuki, 2020). Therefore, due to the *precision limitations* of the floating-point data formats, these differences eventually cannot be captured by the classifier, leading to the decrement in the classification accuracy.

## 5 EXPERIMENTS

### 5.1 SYNTHETIC DATA

Here, we verify our theory on the synthetic data generated from our HSBM model. We set the number of nodes as $n = 1000$, the number of classes as $c = 5$, and the dimension of node features as $d = 5$. For each class $k \in [c]$, its nodes are sampled by a Gaussian distribution with a mean of $e_k$, which is a standard basis vector $[0, ..., 1, ..., 0]$ with a 1 at position $k$, and a standard variance of $0.6$. The averaged node degree for each class is $25$. We employ a family of heterophily patterns, where $\hat{m}_{ij}$ is $a$ when $i = j$, $2a$ when $j = i + 1$ and $\frac{1}{3} - a$ otherwise. Besides, results on more heterophily patterns and large-scale graphs can be found in Appendix B.2 and Appendix B.3, respectively. A one-layered MLP is employed as the baseline. One or multiple GC operations are incorporated into MLP to obtain the corresponding GCN.

**Impact of Different Heterophily Patterns.** Fig. 2(a) shows the results with different heterophily patterns as $a \in [0, 0.33]$. As can be observed, the overall change in accuracy aligns with the varia-

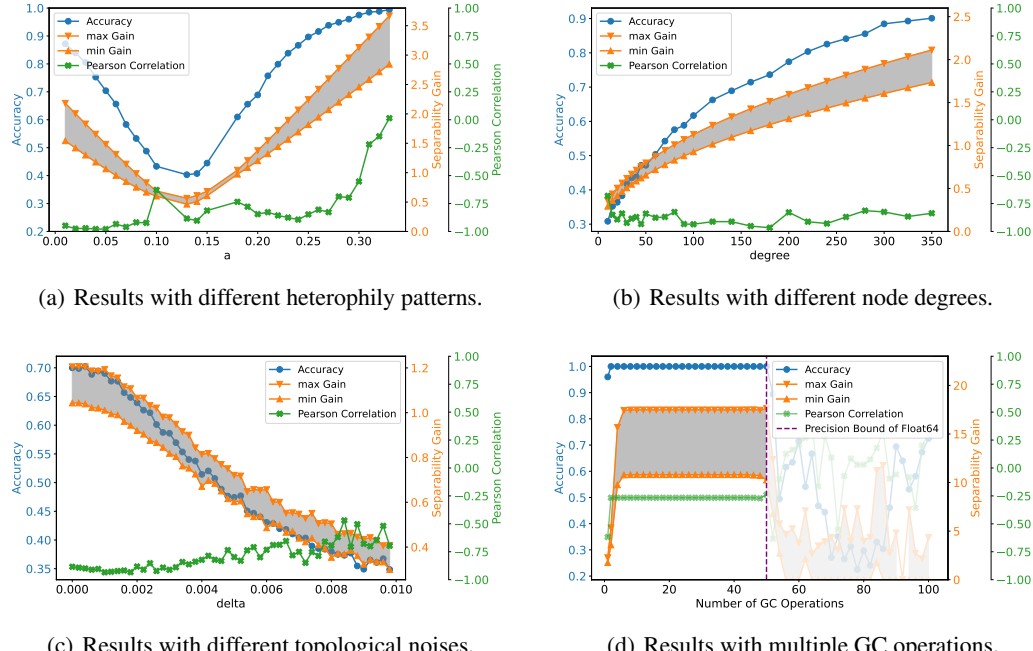

(a) Results with different heterophily patterns.

(b) Results with different node degrees.

(c) Results with different topological noises.

(d) Results with multiple GC operations.

Figure 2: Results on synthetic data. (a)-(c) present the results of MLP incorporated one GC operation, while (d) displays the results with multiple GC operations. In each figure, the gray region, enclosed by the minimum gain and maximum gain curves, represents the area of separability gains. The Pearson Correlation represents the Pearson correlation coefficient between the separability gains and the differences in the confusion matrix with and without the GC operations.

tion in the region of separability gains, which demonstrates the effectiveness of Theorem 2. Besides, the Pearson correlation between the separability gains and the differences in the confusion matrix with and without the GC operations are highly negative. It indicates that different heterophily patterns function by altering the pair-wise classification boundaries. These impacts can be captured by the proposed separability gains. Specifically, Fig. 1 illustrates three typical examples of different heterophily patterns with $a = 0.25, 0.20, 0.18$, respectively. In the good heterophily pattern where $a = 0.25$, the separability gains are relatively high for all cases; hence, GCN performs better than MLP in all the two-classes subtasks, i.e., all non-diagonal elements of the confusion matrix (GCN-MLP) are negative. On the contrary, in the bad heterophily pattern shown in Fig. 1(c), the separability gains are all smaller than $\varsigma_n$, leading to the increment of the number of misclassified nodes. Besides, Fig. 1(b) shows a example of mixed heterophily pattern, wherein the separability between two classes improves for some pairs while deteriorates for others. In summary, although these three examples exhibit similar heterophily patterns, their differences in the distances between the neighborhood distributions lead to distinct impacts on the separability of nodes.

**Impact of Node Degrees.** Fig. 2(b) shows the results with different averaged node degrees when $a = 0.20$. The averaged degree is varied within the range of $[5, 350]$ to assess its impact. It is evident that as the averaged degree increases, the separability gains also increase, resulting in an overall improvement of accuracy. During this process, the heterophily pattern with $a = 0.2$ transforms from being a bad pattern to becoming a good one, which verifies that the averaged degree possess a complementary effects alongside heterophily patterns, as claimed Theorem 2.

**Impact of Topological Noise.** Fig. 2(c) shows the results with different topological noises when $a = 0.20$, where the variance of noise, $\delta$, ranging from 0 to 0.01. Note that in practice, the empirical neighborhood distributions may be influenced by sampling, especially when $\delta$ is large. Thus, we utilize the empirical $\hat{M}$ to calculate the separability gains. As can be observed, as the variance of topological noises increases, the separability gains decrease, leading to a reduction of the overall accuracy, which verifies the results of Theorem 3.

**Impact of stacking Multiple GC Operations.** Fig. 2(d) shows the results with stacking multiple GC operations. Here, the precise separability gains in Theorem A1 are utilized. As can be observed, as $l$ increases, the variations in separability gains closely mirror the fluctuation in accuracy, which

Table 1: Results on the real-world datasets. Note that A.R, A.Y, and S.P. are the abbreviations of Amazon-ratings, Arxiv-year and Snap-patents.

|  | Cora | Chameleon | Workers | Actor | A.R. | Squirrel | A.Y. | S.P. |
|---|---|---|---|---|---|---|---|---|
| Acc(MLP) | 77.68 | 54.29 | 76.81 | 35.00 | 50.58 | 36.63 | 36.40 | 31.50 |
| Acc(GCN) | 90.04 | 70.33 | 78.66 | 30.33 | 48.01 | 60.67 | 42.44 | 34.42 |
| Max Gain | 1.7823 | 1.4220 | 1.1508 | 0.0364 | 0.4504 | 0.4101 | 1.6912 | 1.4820 |
| Min Gain | 1.4216 | 0.2242 | 1.1508 | 0.0093 | 0.0838 | 0.0846 | 0.4255 | 0.2198 |
| Type of Het. | Good | Good | Good | Bad | Mixed | Mixed | Mixed | Mixed |

verifies the effectiveness of our analysis. Both of them increase during the initial several GC operations, and then consistently maintain at high values. However, when $l$ approaches 50, both the separability gains and accuracy experience a sudden decrease. As in a further investigation in Appendix B.4, the aggregated node features encounter a *precision limitation* issue, i.e., the employed float64 data format cannot accurately represent the node features. Specifically, when the relative difference among different node features are smaller than the precision of Float64, the difference cannot be precisely obtained. By increasing the precision of the data format, the point of descent can be postponed.

## 5.2 REAL WORLD DATA

Here, we verify our theory on eight real-world node classification datasets, i.e., Cora, Chameleon, Workers, Actor, Amazon-ratings, Squirrel, as well as two large-scale datasets, i.e., Arxiv-year and Snap-patents. Their statistics are provided in Appendix C.1. On each dataset, a two-layered MLP is employed as the baseline. We incorporate one GC operation in the second layer of MLP to construct the corresponding GCN. Both MLP and GCN run with the same settings, as provided in Appendix C.2. Note that in the real-world datasets, both the node features and topology are highly correlated. Intuitively, this correlation can degrade the variance of the features among the aggregated nodes, like in Appendix A.5.2. Thus, a lower $\varsigma_n$ is required to understand real-world data results.

As shown in Tab. 1, Cora, Chameleon, and Workers exhibit significant separability gains, indicating a good heterophily pattern that enhances the separability among nodes within each class pair. On the contrary, Actor shows minimal separability gains, resulting in reduced distinguishability among nodes, which can be characterized as bad heterophily. Besides, Amazon-ratings, Squirrel, Arxiv-year, and Snap-patents display both substantial and minimal gains, contributing to improved separability in some class pairs while reducing it in others, illustrating mixed heterophily patterns. Therefore, the overall performance of GCN on these two datasets may either surpass or fall behind that of MLP. A more comprehensive discussion can be found in Appendix C.3. These results on real-world datasets further validate the effectiveness of our theory.

## 6 CONCLUSION AND FUTURE WORK

This paper presents theoretical understandings of the impacts of different heterophily patterns for GNNs by incorporating the graph convolution (GC) operations into fully connected networks via the proposed Heterophilous Stochastic Block Models (HSBM), a general random graph model that can accommodate diverse heterophily patterns. We present three novel insights. Firstly, we show that by applying a GC operation, the separability gains are determined by the Euclidean distance of the neighborhood distributions and $\sqrt{\mathbb{E}[\deg]}$, where $\mathbb{E}[\deg]$ is the averaged node degree. Secondly, we show that the topological noise has a detrimental impact on separability. Finally, when applying multiple GC operations, we show that the separability gains are positively correlated to the normalized distance of the $l$-powered neighborhood distributions, which indicates that the nodes still possess separability as $l$ goes to infinity in various regimes.

However, there are still some limitations that need to be improved in future work. First, our theory is constructed based on Gaussian node features. It may be beneficial to extend the analysis to features with more general distributions. Second, our analysis relies on the assumption of the independence among node features and among edges. Exploring the distribution of nodes and edges with complex dependency relationships could potentially provide more valuable insights. These limitations are beyond the scope of our HSBM and are left as the future work.

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

# A   THEORETICAL PROOFS

## A.1   ASSUMPTIONS.

Our analysis is constructed based on the following two gentle assumptions.

**Assumption 1.** $\bar{p}_k \asymp \bar{p}_t, \forall k, t \in [c]$.

**Assumption 2.** $\bar{p}_k = \omega \left( \log^2 n/n \right), \forall k \in [c]$.

Assumption 1 claims that nodes within different classes possess similar averaged degrees. This assumption allows us to approximate $\bar{D} \approx \bar{D}_k$, to make our results more clear and easy to understand. It's worth noting that many literatures also implicitly or explicitly assume that $\bar{D} = \bar{D}_k$. For example, the averaged degrees of different classes are the same when utilizing the CSBM model to generate graphs, with each class exhibiting the same number of nodes (Baranwal et al., 2023; Wu et al., 2023). Our assumption allows the averaged degrees of different classes to possess the same order of magnitude, which covers a broader regime. Also, we observe this property is frequently satisfied in many real-world datasets, as confirmed in Appendix C.1. Assumption 2 states that the graph is not too sparse and enable some numerical properties of graphs, e.g., the number of nodes within each class and node degree concentrate to their expectations. This assumption is similar to those in the random graph model literatures, e.g., (Baranwal et al., 2021).

## A.2   PROOF OF THEOREM 1

### A.2.1   BAYESIAN CLASSIFIER

**Lemma A1** (Bayesian Classifier). *Given* $(\boldsymbol{X}, \boldsymbol{A}) = \mathrm{HSBM}\left(n, c, \sigma, \{\boldsymbol{\mu}_k\}, \boldsymbol{\eta}, \mathbf{M}, \{\boldsymbol{\Delta}_i\}\right)$, *the Bayesian optimal classifier over data* $\boldsymbol{X}$ *is*

$$h^*(\boldsymbol{x}) = \operatorname*{argmax}_{k \in [c]} \left( \langle \boldsymbol{x}, \boldsymbol{\mu}_k \rangle + \sigma^2 \ln \eta_k \right). \tag{10}$$

*Proof.* The Bayesian optimal classifier, which is based on the criterion of maximum a posteriori probability, is defined as

$$h^*(\boldsymbol{x}) = \operatorname*{argmax}_{k \in [c]} \left( \mathbb{P}[y = k \mid \mathbf{x} = \boldsymbol{x}] \right). \tag{11}$$

Then, according to the Bayes' theorem, the posteriori probability is

$$
\begin{aligned}
\mathbb{P}[y = k \mid \mathbf{x} = \boldsymbol{x}] &= \frac{\mathbb{P}[y = k]\mathbb{P}[\boldsymbol{x} \mid y = k]}{\sum_{t \neq k} \mathbb{P}[y = t]\mathbb{P}[\boldsymbol{x} \mid y = t]} \\
&= \frac{1}{1 + \sum_{t \neq k} \frac{\eta_t \cdot \mathbb{P}[\boldsymbol{x} \mid y = t]}{\eta_k \cdot \mathbb{P}[\boldsymbol{x} \mid y = k]}}.
\end{aligned}
\tag{12}
$$

Denoting $\varphi_t = \eta_t \cdot \mathbb{P}[\boldsymbol{x} \mid y = t]$,

$$
\begin{aligned}
\frac{\varphi_t}{\varphi_k} &= \frac{\eta_t \cdot \frac{1}{\sqrt{2\pi}\sigma} \exp\left(-\frac{(\boldsymbol{x} - \boldsymbol{\mu}_t)^T(\boldsymbol{x} - \boldsymbol{\mu}_t)}{2\sigma^2}\right)}{\eta_k \cdot \frac{1}{\sqrt{2\pi}\sigma} \exp\left(-\frac{(\boldsymbol{x} - \boldsymbol{\mu}_k)^T(\boldsymbol{x} - \boldsymbol{\mu}_k)}{2\sigma^2}\right)} \\
&= \frac{\eta_t}{\eta_k} \exp\left(\frac{\langle \boldsymbol{x}, \boldsymbol{\mu}_t \rangle - \langle \boldsymbol{x}, \boldsymbol{\mu}_k \rangle}{\sigma^2}\right).
\end{aligned}
\tag{13}
$$

For any $k_1, k_2 \in [c]$ and $k_1 \neq k_2$, denote $\phi = \sum_{t \neq k_1, k_2} \varphi_t$. Then,

$$
\begin{aligned}
&\mathbb{P}[y = k_1 \mid \mathbf{x} = \boldsymbol{x}] \geq \mathbb{P}[y = k_2 \mid \mathbf{x} = \boldsymbol{x}] \\
\iff & \sum_{t \neq k_2} \frac{\varphi_t}{\varphi_{k_2}} \geq \sum_{t \neq k_1} \frac{\varphi_t}{\varphi_{k_1}} \\
\iff & \frac{\phi + \varphi_{k_1}}{\varphi_{k_2}} \geq \frac{\phi + \varphi_{k_2}}{\varphi_{k_1}} \\
\iff & \varphi_{k_1}^2 + \phi\varphi_{k_1} \geq \varphi_{k_2}^2 + \phi\varphi_{k_2} \\
\iff & \varphi_{k_1} \geq \varphi_{k_2} \\
\iff & \frac{\eta_{k_1}}{\eta_{k_2}} \exp\left( \frac{\langle \boldsymbol{x}, \boldsymbol{\mu}_{k_1} \rangle - \langle \boldsymbol{x}, \boldsymbol{\mu}_{k_2} \rangle}{\sigma^2} \right) \geq 1 \\
\iff & \langle \boldsymbol{x}, \boldsymbol{\mu}_{k_1} \rangle + \sigma^2 \ln \eta_{k_1} \geq \langle \boldsymbol{x}, \boldsymbol{\mu}_{k_2} \rangle + \sigma^2 \ln \eta_{k_2}.
\end{aligned}
\tag{14}
$$

$\square$

### A.2.2 PROOF OF THE PART ONE OF THEOREM 1

*Proof.* For any $i \in [n]$, its features are generated from Gaussian distributions and can be represented as

$$
\boldsymbol{X}_i = \boldsymbol{\mu}_{\varepsilon_i} + \sigma \mathbf{g}_i,
\tag{15}
$$

where $\mathbf{g}_i \sim \mathcal{N}(\mathbf{0}, I)$ is the random Gaussian noise. Then,

$$
\begin{aligned}
\mathbb{P}\left[ \zeta_i(k) \right] &= \mathbb{P}\left[ \mathbb{P}[y = \varepsilon_i \mid \mathbf{x} = \boldsymbol{X}_i] \geq \mathbb{P}[y = k \mid \mathbf{x} = \boldsymbol{X}_i] \right] \\
&= \mathbb{P}\left[ \langle \boldsymbol{X}_i, \boldsymbol{\mu}_{\varepsilon_i} \rangle + \sigma^2 \ln \eta_{\varepsilon_i} \geq \langle \boldsymbol{X}_i, \boldsymbol{\mu}_k \rangle + \sigma^2 \ln \eta_k \right] \\
&= \mathbb{P}\left[ \gamma'^2 + \gamma'\sigma \hat{\boldsymbol{\mu}}_{\varepsilon_i}^T \mathbf{g}_i + \sigma^2 \ln \eta_{\varepsilon_i} \geq \gamma'\sigma \hat{\boldsymbol{\mu}}_k^T \mathbf{g}_i + \sigma^2 \ln \eta_k \right] \\
&= \mathbb{P}\left[ \hat{\boldsymbol{\mu}}_k^T \mathbf{g}_i - \hat{\boldsymbol{\mu}}_{\varepsilon_i}^T \mathbf{g}_i \leq \frac{\gamma'^2 + \sigma^2 \ln \eta_{\varepsilon_i} - \sigma^2 \ln \eta_k}{\gamma'\sigma} \right],
\end{aligned}
\tag{16}
$$

where $\hat{\boldsymbol{\mu}}_k = \boldsymbol{\mu}_k / \|\boldsymbol{\mu}_k\|$. Denote $Z_{\varepsilon_i} = \hat{\boldsymbol{\mu}}_{\varepsilon_i}^T \mathbf{g}_i$ and $Z_k = \hat{\boldsymbol{\mu}}_k^T \mathbf{g}_i$. Then, $Z_{\varepsilon_i}, Z_k \sim \mathcal{N}(0, 1)$ and $\mathbb{E}\left[ Z_{\varepsilon_i} Z_k \right] = 0$. Therefore,

$$
\begin{aligned}
\mathbb{P}\left[ \zeta_i(k) \right] &= \Phi\left( \frac{\gamma'^2 + \sigma^2 \ln \eta_{\varepsilon_i} - \sigma^2 \ln \eta_k}{\sqrt{2}\gamma'\sigma} \right) \\
&= \Phi\left( \frac{\gamma}{2\sigma} + \frac{\sigma \ln \eta_{\varepsilon_i} - \sigma \ln \eta_k}{\gamma} \right) \\
&= \Phi\left( \frac{\gamma}{2\sigma} + \frac{\sigma}{\gamma} \ln \frac{\eta_{\varepsilon_i}}{\eta_k} \right).
\end{aligned}
\tag{17}
$$

Thus, for each $t, k \in [c]$,

$$
\mathbb{E}_{i \in \mathcal{C}_t} \mathbb{P}\left[ \zeta_i(k) \right] = \Phi\left( \frac{\gamma}{2\sigma} + \frac{\sigma}{\gamma} \ln \left( \frac{\eta_t}{\eta_k} \right) \right).
\tag{18}
$$

$\square$

### A.2.3 PROOF OF THE PART TWO OF THEOREM 1

*Proof.* Denote $\hat{\eta}_t = \frac{\eta_t}{\eta_t + \eta_k}$ and $\hat{\eta}_k = \frac{\eta_k}{\eta_t + \eta_k}$. Then, we have $0 < \hat{\eta}_k < \hat{\eta}_t < 1$ and $\hat{\eta}_k + \hat{\eta}_t = 1$. $S(t, k)$ can be rewrited as

$$
\begin{aligned}
S(t, k) &= \hat{\eta}_t \mathbb{E}_{i \in \mathcal{C}_t} \mathbb{P}\left[ \zeta_i(k) \right] + \hat{\eta}_k \mathbb{E}_{i \in \mathcal{C}_k} \mathbb{P}\left[ \zeta_i(t) \right] \\
&= \hat{\eta}_t \Phi\left( \frac{\gamma}{2\sigma} + \frac{\sigma}{\gamma} \ln \frac{\hat{\eta}_t}{\hat{\eta}_k} \right) + \hat{\eta}_k \Phi\left( \frac{\gamma}{2\sigma} - \frac{\sigma}{\gamma} \ln \frac{\hat{\eta}_t}{\hat{\eta}_k} \right).
\end{aligned}
\tag{19}
$$

Denote $\alpha = \frac{\gamma}{\sigma} > 0$. Then, the question is converted to prove the function

$$
f\left( \alpha, \hat{\eta}_t \right) = \hat{\eta}_t \Phi\left( \frac{\alpha}{2} + \frac{1}{\alpha} \ln \frac{\hat{\eta}_t}{\hat{\eta}_k} \right) + \hat{\eta}_k \Phi\left( \frac{\alpha}{2} - \frac{1}{\alpha} \ln \frac{\hat{\eta}_t}{\hat{\eta}_k} \right)
\tag{20}
$$

is an increasing function with respect to both $\alpha$ and $\hat{\eta}_t$. W.o.l.g, let $\hat{\eta}_t \geq \hat{\eta}_k$.

**1.** Firstly, we prove that $f(\alpha, \hat{\eta}_t)$ is an increasing function of $\alpha$, i.e., if $\alpha_1 > \alpha_2 > 0$, we have $f(\alpha_1, \hat{\eta}_t) > f(\alpha_2, \hat{\eta}_t)$ for any $\frac{1}{2} < \hat{\eta}_t < 1$.

When $\alpha_1 > \alpha_2$, we have

$$\Phi\left(\frac{\alpha_1}{2} + \frac{1}{\alpha_1}\ln\frac{\hat{\eta}_t}{\hat{\eta}_k}\right) > \Phi\left(\frac{\alpha_2}{2} + \frac{1}{\alpha_1}\ln\frac{\hat{\eta}_t}{\hat{\eta}_k}\right) \tag{21}$$

and

$$\Phi\left(\frac{\alpha_1}{2} - \frac{1}{\alpha_1}\ln\frac{\hat{\eta}_t}{\hat{\eta}_k}\right) > \Phi\left(\frac{\alpha_2}{2} - \frac{1}{\alpha_1}\ln\frac{\hat{\eta}_t}{\hat{\eta}_k}\right). \tag{22}$$

Thus,

$$f(\alpha_1, \hat{\eta}_t) > \hat{\eta}_t\Phi\left(\frac{\alpha_2}{2} + \frac{1}{\alpha_1}\ln\frac{\hat{\eta}_t}{\hat{\eta}_k}\right) + \hat{\eta}_k\Phi\left(\frac{\alpha_2}{2} - \frac{1}{\alpha_1}\ln\frac{\hat{\eta}_t}{\hat{\eta}_k}\right). \tag{23}$$

Then, we need to prove that

$$\hat{\eta}_t\Phi\left(\frac{\alpha_2}{2} + \frac{1}{\alpha_1}\ln\frac{\hat{\eta}_t}{\hat{\eta}_k}\right) + \hat{\eta}_k\Phi\left(\frac{\alpha_2}{2} - \frac{1}{\alpha_1}\ln\frac{\hat{\eta}_t}{\hat{\eta}_k}\right) > f(\alpha_2, \hat{\eta}_t), \tag{24}$$

which is equivalent to prove that

$$\hat{\eta}_k\int_{\frac{\alpha_2}{2} - \frac{1}{\alpha_2}\ln\frac{\hat{\eta}_t}{\hat{\eta}_k}}^{\frac{\alpha_2}{2} - \frac{1}{\alpha_1}\ln\frac{\hat{\eta}_t}{\hat{\eta}_k}}\phi(y)\,dy - \hat{\eta}_t\int_{\frac{\alpha_2}{2} + \frac{1}{\alpha_1}\ln\frac{\hat{\eta}_t}{\hat{\eta}_k}}^{\frac{\alpha_2}{2} + \frac{1}{\alpha_2}\ln\frac{\hat{\eta}_t}{\hat{\eta}_k}}\phi(y)\,dy > 0, \tag{25}$$

where $\phi(y) = \frac{1}{\sqrt{2\pi}}\exp\left(-\frac{y^2}{2}\right)$. Denote $z = y + \frac{1}{\alpha_2}\ln\frac{\hat{\eta}_t}{\hat{\eta}_k} + \frac{1}{\alpha_1}\ln\frac{\hat{\eta}_t}{\hat{\eta}_k}$. Then, Eq. (25) is equivalent to

$$\hat{\eta}_k\int_{\frac{\alpha_2}{2} + \frac{1}{\alpha_1}\ln\frac{\hat{\eta}_t}{\hat{\eta}_k}}^{\frac{\alpha_2}{2} + \frac{1}{\alpha_2}\ln\frac{\hat{\eta}_t}{\hat{\eta}_k}}\phi\left(z - \frac{1}{\alpha_2}\ln\frac{\hat{\eta}_t}{\hat{\eta}_k} - \frac{1}{\alpha_1}\ln\frac{\hat{\eta}_t}{\hat{\eta}_k}\right)dz - \hat{\eta}_t\int_{\frac{\alpha_2}{2} + \frac{1}{\alpha_1}\ln\frac{\hat{\eta}_t}{\hat{\eta}_k}}^{\frac{\alpha_2}{2} + \frac{1}{\alpha_2}\ln\frac{\hat{\eta}_t}{\hat{\eta}_k}}\phi(y)\,dy > 0$$

$$\Longleftrightarrow \int_{\frac{\alpha_2}{2} + \frac{1}{\alpha_1}\ln\frac{\hat{\eta}_t}{\hat{\eta}_k}}^{\frac{\alpha_2}{2} + \frac{1}{\alpha_2}\ln\frac{\hat{\eta}_t}{\hat{\eta}_k}}\left[\hat{\eta}_k\phi\left(y - \frac{1}{\alpha_2}\ln\frac{\hat{\eta}_t}{\hat{\eta}_k} - \frac{1}{\alpha_1}\ln\frac{\hat{\eta}_t}{\hat{\eta}_k}\right) - \hat{\eta}_t\phi(y)\right]dy > 0. \tag{26}$$

Denote

$$g_1(y) = \hat{\eta}_k\phi\left(y - \frac{1}{\alpha_2}\ln\frac{\hat{\eta}_t}{\hat{\eta}_k} - \frac{1}{\alpha_1}\ln\frac{\hat{\eta}_t}{\hat{\eta}_k}\right) - \hat{\eta}_t\phi(y). \tag{27}$$

Then, we will prove that $g_1(y) > 0$ for all $y \in \left[\frac{\alpha_2}{2} + \frac{1}{\alpha_2}\ln\frac{\hat{\eta}_t}{\hat{\eta}_k}, \frac{\alpha_2}{2} + \frac{1}{\alpha_1}\ln\frac{\hat{\eta}_t}{\hat{\eta}_k}\right]$.

$$g_1(y) > 0$$

$$\Longleftrightarrow \hat{\eta}_k\phi\left(y - \frac{1}{\alpha_2}\ln\frac{\hat{\eta}_t}{\hat{\eta}_k} - \frac{1}{\alpha_1}\ln\frac{\hat{\eta}_t}{\hat{\eta}_k}\right) > \hat{\eta}_t\phi(y)$$

$$\Longleftrightarrow \ln\hat{\eta}_k - \frac{y^2}{2} - \frac{1}{2}\left(\frac{1}{\alpha_1}\ln\frac{\hat{\eta}_t}{\hat{\eta}_k} + \frac{1}{\alpha_2}\ln\frac{\hat{\eta}_t}{\hat{\eta}_k}\right)^2 + \left(\frac{1}{\alpha_1}\ln\frac{\hat{\eta}_t}{\hat{\eta}_k} + \frac{1}{\alpha_2}\ln\frac{\hat{\eta}_t}{\hat{\eta}_k}\right)y > \ln\hat{\eta}_t - \frac{y^2}{2}$$

$$\Longleftrightarrow \left(\frac{1}{\alpha_1}\ln\frac{\hat{\eta}_t}{\hat{\eta}_k} + \frac{1}{\alpha_2}\ln\frac{\hat{\eta}_t}{\hat{\eta}_k}\right)y - \frac{1}{2}\left(\frac{1}{\alpha_1}\ln\frac{\hat{\eta}_t}{\hat{\eta}_k} + \frac{1}{\alpha_2}\ln\frac{\hat{\eta}_t}{\hat{\eta}_k}\right)^2 - \ln\frac{\hat{\eta}_t}{\hat{\eta}_k} > 0 \tag{28}$$

Since $y \in \left[\frac{\alpha_2}{2} + \frac{1}{\alpha_2}\ln\frac{\hat{\eta}_t}{\hat{\eta}_k}, \frac{\alpha_2}{2} + \frac{1}{\alpha_1}\ln\frac{\hat{\eta}_t}{\hat{\eta}_k}\right]$, we have

$$\left(\frac{1}{\alpha_1}\ln\frac{\hat{\eta}_t}{\hat{\eta}_k} + \frac{1}{\alpha_2}\ln\frac{\hat{\eta}_t}{\hat{\eta}_k}\right)y - \frac{1}{2}\left(\frac{1}{\alpha_1}\ln\frac{\hat{\eta}_t}{\hat{\eta}_k} + \frac{1}{\alpha_2}\ln\frac{\hat{\eta}_t}{\hat{\eta}_k}\right)^2 - \ln\frac{\hat{\eta}_t}{\hat{\eta}_k}$$

$$\geq \left(\frac{\alpha_2}{2\alpha_1} - \frac{1}{2}\right)\ln\frac{\hat{\eta}_t}{\hat{\eta}_k} + \left(\frac{1}{2\alpha_1^2} - \frac{1}{2\alpha_2^2}\right)\left(\ln\frac{\hat{\eta}_t}{\hat{\eta}_k}\right)^2 \tag{29}$$

$$> 0.$$

Thus, we have $g_1(y) > 0$ and

$$f(\alpha_1, \hat{\eta}_t) > \hat{\eta}_t \Phi\left(\frac{\alpha_2}{2} + \frac{1}{\alpha_1} \ln \frac{\hat{\eta}_t}{\hat{\eta}_k}\right) + \hat{\eta}_k \Phi\left(\frac{\alpha_2}{2} - \frac{1}{\alpha_1} \ln \frac{\hat{\eta}_t}{\hat{\eta}_k}\right) > f(\alpha_2, \hat{\eta}_t), \tag{30}$$

which means $f(\alpha, \hat{\eta}_t)$ is an increasing function with respect to $\alpha$.

**2.** Secondly, we prove that $f(\alpha, \hat{\eta}_t)$ is also an increasing function with respect to $\hat{\eta}_t$. Denote

$$f_1(\alpha, \hat{\eta}_t) = \hat{\eta}_t \int_{\frac{\alpha}{2}}^{\frac{\alpha}{2} + \frac{1}{\alpha} \ln \frac{\hat{\eta}_t}{\hat{\eta}_k}} \phi(y) \, dy - \hat{\eta}_k \int_{\frac{\alpha}{2} - \frac{1}{\alpha} \ln \frac{\hat{\eta}_t}{\hat{\eta}_k}}^{\frac{\alpha}{2}} \phi(y) \, dy \tag{31}$$

Then,

$$\begin{aligned}
f(\alpha, \hat{\eta}_t) &= \hat{\eta}_t \Phi\left(\frac{\alpha}{2} + \frac{1}{\alpha} \ln \frac{\hat{\eta}_t}{\hat{\eta}_k}\right) + \hat{\eta}_k \Phi\left(\frac{\alpha}{2} - \frac{1}{\alpha} \ln \frac{\hat{\eta}_t}{\hat{\eta}_k}\right) \\
&= \Phi\left(\frac{\alpha}{2}\right) + f_1(\alpha, \hat{\eta}_t).
\end{aligned} \tag{32}$$

Thus, the question is converted to prove that $f_1(\alpha, \hat{\eta}_t)$ is an increasing function with respect to $\hat{\eta}_t$. Denoting $z = \alpha - y$ and $g_2(\hat{\eta}_t, y) = \hat{\eta}_t \phi(y) - \hat{\eta}_k \phi(\alpha - y)$, we have

$$\begin{aligned}
f_1(\alpha, \hat{\eta}_t) &= \hat{\eta}_t \int_{\frac{\alpha}{2}}^{\frac{\alpha}{2} + \frac{1}{\alpha} \ln \frac{\hat{\eta}_t}{\hat{\eta}_k}} \phi(y) \, dy - \hat{\eta}_k \int_{\frac{\alpha}{2}}^{\frac{\alpha}{2} + \frac{1}{\alpha} \ln \frac{\hat{\eta}_t}{\hat{\eta}_k}} \phi(\alpha - z) \, dz \\
&= \int_{\frac{\alpha}{2}}^{\frac{\alpha}{2} + \frac{1}{\alpha} \ln \frac{\hat{\eta}_t}{\hat{\eta}_k}} g_2(\hat{\eta}_t, y) \, dy.
\end{aligned} \tag{33}$$

Firstly, we prove that $g_2(\hat{\eta}_t, y) \geq 0$.

$$\begin{aligned}
& g_2(\hat{\eta}_t, y) \geq 0 \\
\iff & \hat{\eta}_t \phi(y) - \hat{\eta}_k \phi(\alpha - y) \geq 0 \\
\iff & \ln \hat{\eta}_t - \frac{y^2}{2} \geq \ln \hat{\eta}_k - \frac{(\alpha - y)^2}{2} \\
\iff & \alpha y - \frac{\alpha^2}{2} - \ln \frac{\hat{\eta}_t}{\hat{\eta}_k} \leq 0
\end{aligned} \tag{34}$$

Since for $y \in \left[\frac{\alpha}{2}, \frac{\alpha}{2} + \frac{1}{\alpha} \ln \frac{\hat{\eta}_t}{\hat{\eta}_k}\right]$,

$$\alpha y - \frac{\alpha^2}{2} - \ln \frac{\hat{\eta}_t}{\hat{\eta}_k} \leq \alpha\left(\frac{\alpha}{2} + \frac{1}{\alpha} \ln \frac{\hat{\eta}_t}{\hat{\eta}_k}\right) - \frac{\alpha^2}{2} - \ln \frac{\hat{\eta}_t}{\hat{\eta}_k} = 0, \tag{35}$$

we have $g_2(\hat{\eta}_t) \geq 0$.

Secondly, we prove that $g_2(\hat{\eta}_t') > g_2(\hat{\eta}_t)$, if $\hat{\eta}_t' > \hat{\eta}_t$.

$$\begin{aligned}
& g_2(\hat{\eta}_t') > g_2(\hat{\eta}_t) \\
\iff & \hat{\eta}_t' \phi(y) - \hat{\eta}_k' \phi(\alpha - y) > \hat{\eta}_t \phi(y) - \hat{\eta}_k \phi(\alpha - y) \\
\iff & (\hat{\eta}_t' - \hat{\eta}_t) \phi(y) > (\hat{\eta}_k' - \hat{\eta}_k) \phi(\alpha - y)
\end{aligned} \tag{36}$$

Since $\hat{\eta}_t' > \hat{\eta}_t$, we have $\hat{\eta}_t' - \hat{\eta}_t > 0$ and $\hat{\eta}_k' - \hat{\eta}_k < 0$. Thus, $g_2(\hat{\eta}_t') > g_2(\hat{\eta}_t)$ is obtained.

Finally, we go back to the monotonicity of $f_1(\hat{\eta}_t)$.

$$\begin{aligned}
f_1(\hat{\eta}_t') - f_1(\hat{\eta}_t) &= \int_{\frac{\alpha}{2}}^{\frac{\alpha}{2} + \frac{1}{\alpha} \ln \frac{\hat{\eta}_t'}{\hat{\eta}_k'}} g_2(\hat{\eta}_t') - \int_{\frac{\alpha}{2}}^{\frac{\alpha}{2} + \frac{1}{\alpha} \ln \frac{\hat{\eta}_t}{\hat{\eta}_k}} g_2(\hat{\eta}_t) \, dy \\
&\geq \int_{\frac{\alpha}{2}}^{\frac{\alpha}{2} + \frac{1}{\alpha} \ln \frac{\hat{\eta}_t}{\hat{\eta}_k}} g_2(\hat{\eta}_t') - \int_{\frac{\alpha}{2}}^{\frac{\alpha}{2} + \frac{1}{\alpha} \ln \frac{\hat{\eta}_t}{\hat{\eta}_k}} g_2(\hat{\eta}_t) \, dy \\
&\geq \int_{\frac{\alpha}{2}}^{\frac{\alpha}{2} + \frac{1}{\alpha} \ln \frac{\hat{\eta}_t}{\hat{\eta}_k}} [g_2(\hat{\eta}_t') - g_2(\hat{\eta}_t)] \, dy \\
&> 0
\end{aligned} \tag{37}$$

Thus, $f_1(\hat{\eta}_t')$ is an increasing function with respect to $\hat{\eta}_t$. $\qquad\square$

### A.3 PROOF OF THEOREM 2

#### A.3.1 CONCENTRATION PROPERTIES

The following lemmas are derived by utilizing the Chernoff bound (Vershynin, 2018) similarly. These results indicate that, as the number of nodes goes to infinity, the number of nodes within each class, node degrees, and neighbors in each class tend to concentrate around their respective means.

**Lemma A2** (Concentration of the Number of Nodes within Each Class). *For each class $k \in [c]$, with probability at least $1 - 1/\mathrm{Ploy}(n)$, we have*

$$|\mathcal{C}_k| = \eta_k n \pm O(\sqrt{n \log n}). \tag{38}$$

*Proof.* In HSBM, $\varepsilon_1, \ldots, \varepsilon_n$ are random variables independently sampled from the set $[c]$, with probability of $\boldsymbol{\eta}$. Thus, $\mathbb{1}_{\varepsilon_i = k}$ are Bernoulli random variables with parameters $\eta_k$. Then, $|\mathcal{C}_k| = \sum_{i=1}^{n} \mathbb{1}_{\varepsilon_i = k}$ is the sum of independent Bernoulli random variables with the mean of $\mathbb{E}\left[|\mathcal{C}_k|\right] = \eta_k n$. By applying the Chernoff bound (Vershynin, 2018),

$$\mathbb{P}\left\{|\mathcal{C}_k - \eta_k n| \geq \delta \eta_k n\right\} \leq 2 \exp\left(-C_1 \eta_k n \delta^2\right), \tag{39}$$

for some $C_1 > 0$. Let $\delta = \sqrt{\frac{C_2 \log n}{n}}$, where $C_2 > 0$ is a large constant. Then, we have with a probability at least $1 - 2n^{-C_1'}$,

$$|\mathcal{C}_k| = \eta_k n \pm C_2' \sqrt{n \log n}, \tag{40}$$

where $C_1' > 0$ and $C_2' > 0$. □

**Lemma A3** (Concentration of the Node Degrees). *For each node $i \in [n]$, its averaged degree is defined as*

$$\bar{D}_t \triangleq \mathbb{E}\left[D_{ii} | \varepsilon_i = t\right] = n\bar{p}_t, \tag{41}$$

*where $\bar{p}_{\varepsilon_i} = \sum_k m_{\varepsilon_i k} \eta_k$ represents the averaged edge connection probability. Then, with probability at least $1 - 1/\mathrm{Ploy}(n)$, we have*

$$D_{ii} = n\bar{p}_{\varepsilon_i}\left(1 \pm O\left(\frac{1}{\sqrt{\log n}}\right)\right) = \bar{D}_{\varepsilon_i}\left(1 \pm O\left(\frac{1}{\sqrt{\log n}}\right)\right) \tag{42}$$

*and*

$$\frac{1}{D_{ii}} = \frac{1}{n\bar{p}_{\varepsilon_i}}\left(1 \pm O\left(\frac{1}{\sqrt{\log n}}\right)\right) = \frac{1}{\bar{D}_{\varepsilon_i}}\left(1 \pm O\left(\frac{1}{\sqrt{\log n}}\right)\right). \tag{43}$$

*Proof.* For each node $i \in [n]$, its degree $D_{ii} = \sum_{j \in [n]} a_{ij}$ is a sum of $n$ independent Bernoulli random variables, with the mean of $\mathbb{E}\left[D_{ii}\right] = n\bar{p}_{\varepsilon_i}$. Then, by applying the Chernoff bound (Vershynin, 2018), for $\delta \in (0, 1]$ we have

$$\mathbb{P}\left\{|D_{ii} - n\bar{p}_{\varepsilon_i}| \geq \delta n\bar{p}_{\varepsilon_i}\right\} \leq 2 \exp\left(-C_1 n\bar{p}_{\varepsilon_i} \delta^2\right), \tag{44}$$

For some $C_1 > 0$. Note that according to Assumption 1, $d_{\varepsilon_i} = \omega\left(\log^2 n / n\right)$. Then, denoting $\delta = O\left(\frac{1}{\sqrt{\log n}}\right)$, with a probability at least $1 - 2n^{-C_1'}$, we have

$$D_{ii} = n\bar{p}_{\varepsilon_i}\left(1 \pm O\left(\frac{1}{\sqrt{\log n}}\right)\right) \tag{45}$$

and

$$\frac{1}{D_{ii}} = \frac{1}{n\bar{p}_{\varepsilon_i}}\left(1 \pm O\left(\frac{1}{\sqrt{\log n}}\right)\right), \tag{46}$$

where $C_1 > 0$. □

**Lemma A4** (Concentration of the Number of Neighbors in Each Class). *For each node $i \in [n]$ and each class $k \in [c]$, with probability at least $1 - 1/\mathrm{Ploy}(n)$, we have*

$$|\mathcal{C}_k \cap \Gamma_i| = D_{ii} \cdot \hat{m}_{\varepsilon_i k}\left(1 \pm O\left(\frac{1}{\sqrt{\log n}}\right)\right). \tag{47}$$

*Proof.* For each $k \in [c]$ and $i \in [n]$, $|\mathcal{C}_k \cap \Gamma_i| = \sum_{j \in [n]} \mathbb{1}_{\varepsilon_j = k} a_{ij}$ is a sum of $n$ independent Bernoulli random variables, with the mean of $n\eta_k \cdot m_{\varepsilon_i k}$. By applying the Chernoff bound, we have

$$\mathbb{P}\left\{|\mathcal{C}_k \cap \Gamma_i| = \eta_k n m_{\varepsilon_i k}\left(1 \pm \frac{10}{\sqrt{\log n}}\right)\right\} \geq 1 - \exp\left(-\frac{1}{3} \cdot \frac{n}{C} \cdot m_{\varepsilon_i k} \cdot \frac{100}{\log n}\right) \tag{48}$$

Based on the fact that $n > \log^2 n$, we have

$$1 - \exp\left(-\frac{1}{3} \cdot \frac{n}{C} \cdot m_{\varepsilon_i k} \cdot \frac{100}{\log n}\right) \geq 1 - n^{-\frac{100 m_{\varepsilon_i k}}{3C}} \tag{49}$$

Then, with a probability at least $1 - 1/\text{Ploy}(n)$, we have

$$|C_k \cap \Gamma_i| = \eta_k n m_{\varepsilon_i k}\left(1 \pm \frac{10}{\sqrt{\log n}}\right) \tag{50}$$

and

$$|C_k \cap \Gamma_i| = D_{ii} \hat{m}_{\varepsilon_i k}\left(1 \pm O\left(\frac{1}{\sqrt{\log n}}\right)\right). \tag{51}$$

$\square$

### A.3.2 Feature Distributions After a GC Layer

**Lemma A5.** *For any $i \in [n]$, by applying a GC operation, its aggregated features are $\tilde{X}_i = \left(D^{-1}AX\right)_i$. Then, with probability at least $1 - 1/\text{Ploy}(n)$ over $A$ and $\{\varepsilon_i\}_{i \in [n]}$,*

$$\tilde{X}_i \sim \mathcal{N}\left(\tilde{\mu}_{\varepsilon_i}(i), \tilde{\sigma}_{\varepsilon_i}(i)^2\right), \tag{52}$$

*where*

$$\tilde{\mu}_{\varepsilon_i}(i) = \tilde{\mu}_{\varepsilon_i}(1 \pm o(1)), \quad \tilde{\mu}_{\varepsilon_i} = \sum_{k \in [c]} \hat{m}_{\varepsilon_i k} \mu_k, \tag{53}$$

$$\tilde{\sigma}_{\varepsilon_i}(i) = \tilde{\sigma}_{\varepsilon_i}(1 \pm o(1)), \quad \tilde{\sigma}_{\varepsilon_i} = \frac{\sigma}{\sqrt{\bar{D}_{\varepsilon_i}}}, \tag{54}$$

*and $o(1) = O\left(\frac{1}{\sqrt{\log n}}\right)$.*

*Proof.* For each node $i \in [n]$, the aggregated node features are

$$\tilde{X}_i = \frac{1}{D_{ii}} \sum_{j \in \Gamma_i} X_j = \frac{1}{D_{ii}} \sum_{j \in \Gamma_i} \mu_{\varepsilon_j} + \frac{\sigma}{D_{ii}} \sum_{j \in \Gamma_i} \mathbf{g}_j, \tag{55}$$

where $\mathbf{g}_j \sim \mathcal{N}(\mathbf{0}, \mathbf{I})$ are random noises. Then, the first item in Eq. (55) can be derived as

$$\begin{aligned}
\frac{1}{D_{ii}} \sum_{j \in \Gamma_i} \mu_{\varepsilon_j} &= \frac{1}{D_{ii}}\left(\sum_{k \in [c]} \sum_{j \in \mathcal{C}_k \cap \Gamma_i} \mu_k\right) \\
&= \frac{1}{D_{ii}} \sum_{k \in [c]} |\mathcal{C}_k \cap \Gamma_i| \mu_k \\
&= \frac{1}{d_{\varepsilon_i}} \sum_{k \in [c]} m_{\varepsilon_i k} \eta_k \mu_k\left(1 \pm O\left(\frac{1}{\sqrt{\log n}}\right)\right) \\
&= \sum_{k \in [c]} \hat{m}_{\varepsilon_i k} \mu_k\left(1 \pm O\left(\frac{1}{\sqrt{\log n}}\right)\right).
\end{aligned} \tag{56}$$

For the second item in Eq. (55), let $\mathbf{g}'_i = \frac{1}{\sqrt{D_{ii}}} \sum_{j \in \Gamma_i} \mathbf{g}_j$. Since $\mathbf{g}_j$ are sampled independently for all $j \in [n]$, $\mathbf{g}'_i \sim \mathcal{N}(\mathbf{0}, \mathbf{I})$. Thus,

$$\frac{\sigma}{D_{ii}} \sum_{j \in N_i} \mathbf{g}_j = \frac{\sigma}{\sqrt{D_{ii}}} \mathbf{g}'_i \sim \mathcal{N}\left(\mathbf{0}, \tilde{\sigma}_{\varepsilon_i}^2\left(1 \pm O\left(\frac{1}{\sqrt{\log n}}\right)\right)\mathbf{I}\right), \tag{57}$$

where $\tilde{\sigma}_{\varepsilon_i} = \frac{\sigma}{\sqrt{\bar{D}_{\varepsilon_i}}}$. Therefore, the aggregated features can be represented as

$$\tilde{\boldsymbol{X}}_i = \tilde{\boldsymbol{\mu}}_{\varepsilon_i} \left( 1 \pm O\left( \frac{1}{\sqrt{\log n}} \right) \right) + \tilde{\sigma}_{\varepsilon_i} \left( 1 \pm O\left( \frac{1}{\sqrt{\log n}} \right) \right) \mathbf{g}_i, \tag{58}$$

where

$$\tilde{\boldsymbol{\mu}}_{\varepsilon_i} = \sum_{k \in [c]} \hat{m}_{\varepsilon_i k} \boldsymbol{\mu}_k \tag{59}$$

and

$$\tilde{\sigma}_{\varepsilon_i} = \frac{\sigma}{\sqrt{\bar{D}_{\varepsilon_i}}}. \tag{60}$$

$\square$

Recall that the features of each node are sampled from Gaussian distributions in HSBM as stated in Sec. 3. Lemma A5 indicates that aggregating the Gaussian features according to $A$ and $\{\varepsilon_i\}_{i \in [n]}$ is equivalent to sampling the aggregated features from a new Gaussian distribution. Meanwhile, these new aggregated Gaussian distributions are concentrated within each class. Note that for simplicity, the correlations among $\tilde{\boldsymbol{X}}_j$ are neglected due to their sightly impacts. A comprehensive analysis that takes full consideration of the correlations can be found in Appendix A.5.2.

**Lemma A6.** *For each node $i \in [n]$, the Bayesian optimal classifier over the aggregated node features $\tilde{\boldsymbol{X}}_i$ is*

$$\tilde{h}^*(\tilde{\boldsymbol{x}}) = \underset{k \in [c]}{\operatorname{argmax}} \left( \tilde{\varphi}_k \right), \tag{61}$$

*as $n \to \infty$, where*

$$\tilde{\varphi}_k = \eta_k \frac{1}{\sqrt{2\pi}\tilde{\sigma}_k} \exp\left( -\frac{(\tilde{\boldsymbol{x}} - \tilde{\boldsymbol{\mu}}_k)^T (\tilde{\boldsymbol{x}} - \tilde{\boldsymbol{\mu}}_k)}{2\tilde{\sigma}_k^2} \right). \tag{62}$$

*Proof.* The posteriori probability is

$$\begin{aligned} \mathbb{P}[y = k \mid \tilde{\boldsymbol{x}}] &= \frac{\mathbb{P}[y = k]\mathbb{P}[\tilde{\boldsymbol{x}} \mid y = k]}{\sum_{t \neq k} \mathbb{P}[y = t]\mathbb{P}[\tilde{\boldsymbol{x}} \mid y = t]} \\ &= \frac{1}{1 + \sum_{t \neq k} \frac{\eta_t \mathbb{P}[\tilde{\boldsymbol{x}}|y=t]}{\eta_k \mathbb{P}[\tilde{\boldsymbol{x}}|y=k]}}. \end{aligned} \tag{63}$$

Denote $\tilde{\varphi}_k = \eta_k \mathbb{P}[\tilde{\boldsymbol{x}} \mid y = k]$. For any $k_1, k_2 \in [c]$ and $k_1 \neq k_2$, we have

$$\mathbb{P}[y = k_1 \mid \tilde{\boldsymbol{x}}] \geq \mathbb{P}[y = k_2 \mid \tilde{\boldsymbol{x}}] \iff \tilde{\varphi}_{k_1} \geq \tilde{\varphi}_{k_2}. \tag{64}$$

$\square$

### A.3.3 PROOF OF THEOREM 2

*Proof.* According to Lemma A5, the aggregated node features follow the distribution

$$\tilde{\boldsymbol{X}}_i \sim \mathcal{N}\left( \tilde{\boldsymbol{\mu}}_{\varepsilon_i}(i), \tilde{\sigma}_{\varepsilon_i}(i)^2 \right), \quad \forall i \in [n]. \tag{65}$$

Denote $k, t \in [c]$. Then, we have

$$\begin{aligned} \tilde{\boldsymbol{\mu}}_k^T \tilde{\boldsymbol{\mu}}_t &= \left( \sum_{o \in [c]} \hat{m}_{ko} \boldsymbol{\mu}_o \right)^T \left( \sum_{o \in [c]} \hat{m}_{to} \boldsymbol{\mu}_o \right) \\ &= \sum_{o \in [c]} \hat{m}_{ko} \hat{m}_{to} \boldsymbol{\mu}_o^T \boldsymbol{\mu}_o \\ &= \gamma'^2 < \hat{m}_k, \hat{m}_t > . \end{aligned} \tag{66}$$

For each node $i \in [n]$, the probability of the node being misclassified regarding class $k$ is

$$\mathbb{P}\left[\zeta_i(k)\right] = \mathbb{P}\left[\tilde{\varphi}_{\varepsilon_i} > \tilde{\varphi}_k\right]. \tag{67}$$

Denote $B = \sqrt{\frac{\bar{D}_k}{\bar{D}_{\varepsilon_i}}}$. Since $d_k \asymp d_t$, $B = 1 \pm o(1)$. Then, we have

$$
\begin{aligned}
&\tilde{\varphi}_{\varepsilon_i} > \tilde{\varphi}_k \\
\Longleftrightarrow\ & \eta_{\varepsilon_i} \sqrt{\bar{D}_{\varepsilon_i}} \exp\left(-\frac{||\tilde{\boldsymbol{x}} - \tilde{\boldsymbol{\mu}}_{\varepsilon_i}||^2}{2\tilde{\sigma}_{\varepsilon_i}^2}\right) > \eta_k \sqrt{\bar{D}_k} \exp\left(-\frac{||\tilde{\boldsymbol{x}} - \tilde{\boldsymbol{\mu}}_k||^2}{2\tilde{\sigma}_k^2}\right) \\
\Longleftrightarrow\ & \ln\left(\eta_{\varepsilon_i} \sqrt{\bar{D}_{\varepsilon_i}}\right) - \frac{||\tilde{\boldsymbol{x}} - \tilde{\boldsymbol{\mu}}_{\varepsilon_i}||^2}{2\tilde{\sigma}_{\varepsilon_i}^2} > \ln\left(\eta_k \sqrt{\bar{D}_k}\right) - \frac{||\tilde{\boldsymbol{x}} - \tilde{\boldsymbol{\mu}}_k||^2}{2\tilde{\sigma}_k^2} \\
\Longleftrightarrow\ & \ln\left(\eta_{\varepsilon_i} \sqrt{\bar{D}_{\varepsilon_i}}\right) - \frac{\bar{D}_{\varepsilon_i}||\tilde{\boldsymbol{x}} - \tilde{\boldsymbol{\mu}}_{\varepsilon_i}||^2}{2\sigma^2} > \ln\left(\eta_k \sqrt{\bar{D}_{\varepsilon_i}} B\right) - \frac{\bar{D}_{\varepsilon_i} B^2 ||\tilde{\boldsymbol{x}} - \tilde{\boldsymbol{\mu}}_k||^2}{2\sigma^2} \\
\Longleftrightarrow\ & \frac{2\sigma^2}{\bar{D}_{\varepsilon_i}} \ln\left(\frac{\eta_{\varepsilon_i}}{\eta_k B}\right) - ||\tilde{\boldsymbol{x}}||^2 \left(1 - B^2\right) + 2\tilde{\boldsymbol{x}}^T \tilde{\boldsymbol{\mu}}_{\varepsilon_i} - 2\tilde{\boldsymbol{x}}^T \tilde{\boldsymbol{\mu}}_k B^2 - ||\tilde{\boldsymbol{\mu}}_{\varepsilon_i}||^2 + ||\tilde{\boldsymbol{\mu}}_k||^2 B^2 > 0 \\
\Longleftrightarrow\ & 2\tilde{\sigma}_{\varepsilon_i}^2 \ln\left(\frac{\eta_{\varepsilon_i}}{\eta_k}\right) \pm o(1) + ||\tilde{\boldsymbol{\mu}}_{\varepsilon_i}||^2 \varsigma_n + 2\tilde{\sigma}_{\varepsilon_i} \tilde{\boldsymbol{\mu}}_{\varepsilon_i}^T \mathbf{g}_i \varsigma_n - 2\tilde{\boldsymbol{\mu}}_{\varepsilon_i}^T \tilde{\boldsymbol{\mu}}_k B^2 \varsigma_n - 2\tilde{\sigma}_{\varepsilon_i} \tilde{\boldsymbol{\mu}}_k^T \mathbf{g}_i \varsigma_n + ||\tilde{\boldsymbol{\mu}}_k||^2 B^2 > 0 \\
\Longleftrightarrow\ & 2\tilde{\sigma}_{\varepsilon_i}^2 \ln\left(\frac{\eta_{\varepsilon_i}}{\eta_k}\right) \varsigma_n + \gamma'^2 ||B\hat{\mathbf{m}}_k - \hat{\mathbf{m}}_{\varepsilon_i}||^2 \varsigma_n > 2\tilde{\sigma}_{\varepsilon_i} \gamma' \sum_{o \in [c]} (B\hat{m}_{ko} - \hat{m}_{\varepsilon_i o}) \frac{\boldsymbol{\mu}_o^T}{||\boldsymbol{\mu}_o||} \mathbf{g}_i \\
\Longleftrightarrow\ & \frac{\tilde{\sigma}_{\varepsilon_i}}{\gamma' ||B\hat{\mathbf{m}}_k - \hat{\mathbf{m}}_{\varepsilon_i}||} \ln\left(\frac{\eta_{\varepsilon_i}}{\eta_k}\right) \varsigma_n + \frac{\gamma'}{2\tilde{\sigma}_{\varepsilon_i}} ||B\hat{\mathbf{m}}_k - \hat{\mathbf{m}}_{\varepsilon_i}|| \varsigma_n > \sum_{o \in [c]} \frac{B\hat{m}_{ko} - \hat{m}_{\varepsilon_i o}}{||B\hat{\mathbf{m}}_k - \hat{\mathbf{m}}_{\varepsilon_i}||} \frac{\boldsymbol{\mu}_o^T}{||\boldsymbol{\mu}_o||} \mathbf{g}_i,
\end{aligned}
\tag{68}
$$

where $\varsigma_n = 1 \pm o_n(1)$ and $o_n(1)$ is an error term. In the second line, we take the logarithm of both sides of the inequality. In the third line, we replace $\bar{D}_k$ as $B\bar{D}_{\varepsilon_i}$. In the fifth line, we utilize the fact that $(1 - B) = \pm o(1)$ and $\ln(B) = \pm o(1)$.

Then, for each $o \in [c]$, denoting $Z_o = \frac{\boldsymbol{\mu}_o^T}{||\boldsymbol{\mu}_o||} \mathbf{g}_i$, $Z_o \sim \mathcal{N}(0, 1)$. For all $o_1 \neq o_2$, $\mathbb{E}[Z_{o_1} Z_{o_2}] = 0$. Thus, we have

$$\sum_{o \in [c]} \frac{B\hat{m}_{ko} - \hat{m}_{\varepsilon_i o}}{||B\hat{\mathbf{m}}_k - \hat{\mathbf{m}}_{\varepsilon_i}||} \frac{\boldsymbol{\mu}_o^T}{||\boldsymbol{\mu}_o||} \mathbf{g}_i \sim \mathcal{N}(0, 1). \tag{69}$$

Then, the probability

$$
\begin{aligned}
\mathbb{P}\left[\zeta_i(k)\right] &= \Phi\left(\frac{\gamma'}{2\tilde{\sigma}_{\varepsilon_i}} \frac{||B\hat{\mathbf{m}}_k - \hat{\mathbf{m}}_{\varepsilon_i}||}{\varsigma_n} + \frac{\tilde{\sigma}_{\varepsilon_i} \varsigma_n}{\gamma' ||B\hat{\mathbf{m}}_k - \hat{\mathbf{m}}_{\varepsilon_i}||} \ln\left(\frac{\eta_{\varepsilon_i}}{\eta_k}\right)\right) \\
&= \Phi\left(\frac{\gamma}{2\sigma} \frac{F_{\varepsilon_i k}}{\varsigma_n} + \frac{\sigma \varsigma_n}{\gamma F_{\varepsilon_i k}} \ln\left(\frac{\eta_{\varepsilon_i}}{\eta_k}\right)\right),
\end{aligned}
\tag{70}
$$

where $F_{\varepsilon_i k} = \frac{1}{\sqrt{2}} ||\sqrt{\bar{D}_k} \hat{\mathbf{m}}_k - \sqrt{\bar{D}_{\varepsilon_i}} \hat{\mathbf{m}}_{\varepsilon_i}||$ and $\varsigma_n = (1 \pm o(1))$. Thus,

$$\mathbb{E}_{i \in \mathcal{C}_t} \mathbb{P}\left[\zeta_i(k)\right] = \Phi\left(\frac{\gamma}{2\sigma} \frac{F_{tk}}{\varsigma_n} + \frac{\sigma}{\gamma} \frac{\varsigma_n}{F_{tk}} \ln\left(\frac{\eta_t}{\eta_k}\right)\right). \tag{71}$$

$\square$

## A.4 Proof of Theorem 3

*Proof.* For each node $i \in [n]$, the aggregated node features are

$$\tilde{\boldsymbol{X}}_i = \frac{1}{D_{ii}} \sum_{j \in \Gamma_i} \boldsymbol{X}_j = \frac{1}{D_{ii}} \sum_{j \in \Gamma_i} \boldsymbol{\mu}_{\varepsilon_j} + \frac{\sigma}{D_{ii}} \sum_{j \in \Gamma_i} \mathbf{g}_j, \tag{72}$$

where $\mathbf{g}_j \sim \mathcal{N}(\mathbf{0}, \mathbf{I})$ are random noises. Then, the first term in Eq. (72) can be derived as

$$
\begin{aligned}
\frac{1}{D_{ii}} \sum_{j \in \Gamma_i} \boldsymbol{\mu}_{\varepsilon_j} &= \frac{1}{D_{ii}} \left( \sum_{k \in [c]} \sum_{j \in \mathcal{C}_k \cap \Gamma_i} \boldsymbol{\mu}_k \right) \\
&= \frac{1}{D_{ii}} \sum_{k \in [c]} |\mathcal{C}_k \cap \Gamma_i| \boldsymbol{\mu}_k \\
&= \sum_{k \in [c]} \left( \hat{m}_{\varepsilon_i k} + \Delta_{ik} \right) \boldsymbol{\mu}_k \varsigma_n \\
&= \tilde{\boldsymbol{\mu}}_{\varepsilon_i} \varsigma_n + \sum_{k \in [c]} \Delta_{ik} \boldsymbol{\mu}_k \varsigma_n \\
&= \tilde{\boldsymbol{\mu}}_{\varepsilon_i} \varsigma_n + \gamma' \boldsymbol{\Delta}_i \boldsymbol{Q} \varsigma_n,
\end{aligned}
\tag{73}
$$

where

$$
\tilde{\boldsymbol{\mu}}_{\varepsilon_i} = \sum_{k \in [c]} \hat{m}_{\varepsilon_i k} \boldsymbol{\mu}_k, \quad \varsigma_n = \left( 1 \pm O \left( \frac{1}{\sqrt{\log n}} \right) \right), \quad \boldsymbol{Q} = \begin{pmatrix} \hat{\boldsymbol{\mu}}_0 \\ \hat{\boldsymbol{\mu}}_1 \\ \vdots \\ \hat{\boldsymbol{\mu}}_{c-1} \end{pmatrix}.
\tag{74}
$$

Since $\Delta_i \sim \mathcal{N}(\mathbf{0}, \delta^2 \boldsymbol{I})$ and $\boldsymbol{Q}$ is an orthogonal matrix, we have

$$
\gamma' \boldsymbol{\Delta}_i \boldsymbol{Q} \sim \mathcal{N} \left( \mathbf{0}, \frac{\gamma^2 \delta^2}{2} \mathbf{I} \right).
\tag{75}
$$

For the second term in Eq. (72), let $\mathbf{g}_i' = \frac{1}{\sqrt{D_{ii}}} \sum_{j \in \Gamma_i} \mathbf{g}_j$. Since $\mathbf{g}_j$ are sampled independently for all $j \in [n]$, $\mathbf{g}_i' \sim \mathcal{N}(\mathbf{0}, \mathbf{I})$. Thus,

$$
\frac{\sigma}{D_{ii}} \sum_{j \in N_i} \mathbf{g}_j = \frac{\sigma}{\sqrt{D_{ii}}} \mathbf{g}_i' = \tilde{\sigma}_{\varepsilon_i} \varsigma_n \mathbf{g}_i'
\tag{76}
$$

where $\tilde{\sigma}_{\varepsilon_i} = \frac{\sigma}{\sqrt{\bar{D}_{\varepsilon_i}}}$. Therefore, the aggregated features can be represented as

$$
\begin{aligned}
\tilde{\boldsymbol{X}}_i &= \left( \tilde{\boldsymbol{\mu}}_{\varepsilon_i} + \frac{\gamma \delta}{\sqrt{2}} \mathbf{g}_i'' + \frac{\sigma}{\sqrt{\bar{D}_{\varepsilon_i}}} \mathbf{g}_i' \right) \varsigma_n \\
&= \left( \tilde{\boldsymbol{\mu}}_{\varepsilon_i} + \sqrt{\frac{\gamma^2 \delta^2}{2\sigma^2} + \frac{1}{\bar{D}_{\varepsilon_i}}} \sigma \mathbf{g}_i''' \right) \varsigma_n,
\end{aligned}
\tag{77}
$$

where $\mathbf{g}_i'', \mathbf{g}_i''' \sim \mathcal{N}(\mathbf{0}, \mathbf{I})$. Denote $\tilde{\sigma}_{\varepsilon_i}' = \rho_{\varepsilon_i} \sigma$, $\rho_{\varepsilon_i} = \sqrt{\frac{\gamma^2 \delta^2}{2} + \frac{1}{\bar{D}_{\varepsilon_i}}}$, and $\varrho = \frac{\rho_{\varepsilon_i}}{\rho_k}$. Then,

$$
\begin{aligned}
&\tilde{\varphi}_{\varepsilon_i} > \tilde{\varphi}_k \\
\iff &\frac{\eta_{\varepsilon_i}}{\tilde{\sigma}_{\varepsilon_i}'} \exp \left( -\frac{||\tilde{\boldsymbol{x}} - \tilde{\boldsymbol{\mu}}_{\varepsilon_i}||^2}{2\tilde{\sigma}_{\varepsilon_i}'^2} \right) > \frac{\eta_k}{\tilde{\sigma}_k'} \exp \left( -\frac{||\tilde{\boldsymbol{x}} - \tilde{\boldsymbol{\mu}}_k||^2}{2\tilde{\sigma}_k'^2} \right) \\
\iff &\ln \left( \frac{\eta_{\varepsilon_i}}{\rho_{\varepsilon_i}} \right) - \frac{||\tilde{\boldsymbol{x}} - \tilde{\boldsymbol{\mu}}_{\varepsilon_i}||^2}{2\rho_{\varepsilon_i}^2 \sigma^2} > \ln \left( \frac{\eta_k}{\rho_k} \right) - \frac{||\tilde{\boldsymbol{x}} - \tilde{\boldsymbol{\mu}}_k||^2}{2\rho_k^2 \sigma^2} \\
\iff &2\tilde{\sigma}_{\varepsilon_i}'^2 \ln \left( \frac{\eta_{\varepsilon_i}}{\eta_k} \right) \pm o(1) + ||\tilde{\boldsymbol{\mu}}_{\varepsilon_i}||^2 \varsigma_n + 2\tilde{\sigma}_{\varepsilon_i}' \tilde{\boldsymbol{\mu}}_{\varepsilon_i}^T \mathbf{g}_i \varsigma_n - 2\tilde{\boldsymbol{\mu}}_{\varepsilon_i}^T \tilde{\boldsymbol{\mu}}_k \varrho \varsigma_n - 2\tilde{\sigma}_{\varepsilon_i}' \tilde{\boldsymbol{\mu}}_k^T \mathbf{g}_i \varsigma_n + ||\tilde{\boldsymbol{\mu}}_k||^2 \varrho > 0 \\
\iff &2\tilde{\sigma}_{\varepsilon_i}'^2 \ln \left( \frac{\eta_{\varepsilon_i}}{\eta_k} \right) \varsigma_n + \gamma'^2 ||\varrho \hat{\mathbf{m}}_k - \hat{\mathbf{m}}_{\varepsilon_i}||^2 \varsigma_n > 2\tilde{\sigma}_{\varepsilon_i}' \gamma' \sum_{o \in [c]} \left( \varrho \hat{m}_{ko} - \hat{m}_{\varepsilon_i o} \right) \frac{\boldsymbol{\mu}_o^T}{||\boldsymbol{\mu}_o||} \mathbf{g}_i \\
\iff &\frac{\tilde{\sigma}_{\varepsilon_i}'}{\gamma' ||\varrho \hat{\mathbf{m}}_k - \hat{\mathbf{m}}_{\varepsilon_i}||} \ln \left( \frac{\eta_{\varepsilon_i}}{\eta_k} \right) \varsigma_n + \frac{\gamma'}{2\tilde{\sigma}_{\varepsilon_i}'} ||\varrho \hat{\mathbf{m}}_k - \hat{\mathbf{m}}_{\varepsilon_i}|| \varsigma_n > \sum_{o \in [c]} \frac{\varrho \hat{m}_{ko} - \hat{m}_{\varepsilon_i o}}{||\varrho \hat{\mathbf{m}}_k - \hat{\mathbf{m}}_{\varepsilon_i}||} \frac{\boldsymbol{\mu}_o^T}{||\boldsymbol{\mu}_o||} \mathbf{g}_i.
\end{aligned}
\tag{78}
$$

Similar to Eq. (69), we have

$$\sum_{o\in[c]} \frac{\varrho\hat{m}_{ko} - \hat{m}_{\varepsilon_i o}}{||\varrho\hat{\mathbf{m}}_k - \hat{\mathbf{m}}_{\varepsilon_i}||} \frac{\boldsymbol{\mu}_o^T}{||\boldsymbol{\mu}_o||}\mathbf{g}_i \sim \mathcal{N}(0,1). \tag{79}$$

Then, the probability

$$\mathbb{P}[\zeta_i(k)] = \Phi\left(\frac{\gamma'}{2\tilde{\sigma}'_{\varepsilon_i}} \frac{||\varrho\hat{\mathbf{m}}_k - \hat{\mathbf{m}}_{\varepsilon_i}||}{\varsigma_n} + \frac{\tilde{\sigma}'_{\varepsilon_i}}{\gamma'} \frac{\varsigma_n}{||\varrho\hat{\mathbf{m}}_k - \hat{\mathbf{m}}_{\varepsilon_i}||} \ln\left(\frac{\eta_{\varepsilon_i}}{\eta_k}\right)\right)$$

$$= \Phi\left(\frac{\gamma}{2\sigma} \frac{F'_{\varepsilon_i k}}{\varsigma_n} + \frac{\sigma}{\gamma} \frac{\varsigma_n}{F'_{\varepsilon_i k}} \ln\left(\frac{\eta_{\varepsilon_i}}{\eta_k}\right)\right), \tag{80}$$

where $F'_{\varepsilon_i k} = \frac{1}{\sqrt{2}}||\frac{1}{\rho_k}\hat{\mathbf{m}}_k - \frac{1}{\rho_{\varepsilon_i}}\hat{\mathbf{m}}_{\varepsilon_i}||$ and $\varsigma_n = (1 \pm o(1))$. Thus,

$$\mathbb{E}_{i\in\mathcal{C}_t}\mathbb{P}[\zeta_i(k)] = \Phi\left(\frac{\gamma}{2\sigma} \frac{F'_{tk}}{\varsigma_n} + \frac{\sigma}{\gamma} \frac{\varsigma_n}{F'_{tk}} \ln\left(\frac{\eta_{\varepsilon_i}}{\eta_k}\right)\right). \tag{81}$$

$\square$

## A.5    PROOF OF THEOREM 4

### A.5.1    SINGLE NODE FEATURE DISTRIBUTIONS AFTER STACKING $l$ GC LAYERS

**Lemma A7.** *For any node $i \in [n]$, by applying $l$ GC operations, its aggregated features are obtained as $\boldsymbol{X}_i^{(l)} = \left(\left(\boldsymbol{D}^{-1}\boldsymbol{A}\right)^l \boldsymbol{X}\right)_i$. Then, with probability at least $1 - 1/\text{Ploy}(n)$ over $A$ and $\{\varepsilon_i\}_{i\in[n]}$,*

$$\boldsymbol{X}_i^{(l)} = \boldsymbol{\mu}_{\varepsilon_i}^{(l)}(i) + \sigma_i^{(l)}\mathbf{g}^{(l)}, \tag{82}$$

*where*

$$\boldsymbol{\mu}_{\varepsilon_i}^{(l)}(i) = \boldsymbol{\mu}_{\varepsilon_i}^{(l)}(1 \pm lo_n(1)), \quad \boldsymbol{\mu}_{\varepsilon_i}^{(l)} = \sum_{k\in[c]} \hat{m}_{\varepsilon_i k}^{(l)}\boldsymbol{\mu}_k, \tag{83}$$

$$\frac{1}{\sqrt{n}}\sigma \leq \sigma_{\varepsilon_i}^{(l)} \leq \sigma, \tag{84}$$

$o_n(1) = O\left(\frac{1}{\sqrt{\log n}}\right)$, *and* $\mathbf{g}^{(l)} \sim \mathcal{N}(\mathbf{0}, \boldsymbol{I})$.

*Proof.* By applying $l$ GC operations, the aggregated features of node $i$ are

$$\boldsymbol{X}_i^{(l)} = \left(\boldsymbol{D}^{-1}\boldsymbol{A}\right)_i^l \boldsymbol{X} = \sum_{j\in[n]} \left(\boldsymbol{D}^{-1}\boldsymbol{A}\right)_{ij}^l \boldsymbol{\mu}_{\varepsilon_j} + \left(\boldsymbol{D}^{-1}\boldsymbol{A}\right)_{ij}^l \mathbf{g}_j, \tag{85}$$

where $\mathbf{g}_j \sim \mathcal{N}(\mathbf{0}, \mathbf{I})$ are random noises. Then, we now prove that the first term in Eq. (85) can be derived as

$$\sum_{j\in[n]} \left(\boldsymbol{D}^{-1}\boldsymbol{A}\right)_{ij}^l \boldsymbol{\mu}_{\varepsilon_j} = \boldsymbol{\mu}_{\varepsilon_i}^{(l)}(1 \pm lo_n(1)), \quad \text{where } \boldsymbol{\mu}_{\varepsilon_i}^{(l)} = \sum_{k\in[c]} \hat{m}_{\varepsilon_i k}^{(l)}\boldsymbol{\mu}_k. \tag{86}$$

When $l = 1$, Eq. (86) holds, as is claimed in Lemma A5.

When $l > 1$, assuming that Eq. (86) holds for $l - 1$, we have

$$\sum_{j\in[n]} \left(\boldsymbol{D}^{-1}\boldsymbol{A}\right)_{ij}^l \boldsymbol{\mu}_{\varepsilon_j} = \sum_{j,k\in[n]} \left(\boldsymbol{D}^{-1}\boldsymbol{A}\right)_{ik} \left(\boldsymbol{D}^{-1}\boldsymbol{A}\right)_{kj}^{l-1} \boldsymbol{\mu}_{\varepsilon_j}$$

$$= \sum_{k\in[n]} \left(\boldsymbol{D}^{-1}\boldsymbol{A}\right)_{ik} \sum_{t\in[c]} \hat{m}_{\varepsilon_k t}^{(l-1)}\boldsymbol{\mu}_t(1 \pm (l-1)o_n(1))$$

$$= \sum_{k\in[c]} D_{ii}^{-1}|\mathcal{C}_k \cap \Gamma_i| \sum_{t\in[c]} \hat{m}_{kt}^{(l-1)}\boldsymbol{\mu}_t(1 \pm (l-1)o_n(1)) \tag{87}$$

$$= \sum_{k\in[c]} \hat{m}_{\varepsilon_i k}(1 \pm o_n(1)) \sum_{t\in[c]} \hat{m}_{kt}^{(l-1)}\boldsymbol{\mu}_t(1 \pm (l-1)o_n(1))$$

$$= \sum_{t\in[c]} \hat{m}_{\varepsilon_i t}^{(l)}\boldsymbol{\mu}_t(1 \pm lo_n(1)).$$

For the second term, since $\mathbf{g}_j$ are sampled independently for all $j \in [n]$,

$$\left(\boldsymbol{D}^{-1}\boldsymbol{A}\right)_{ij}^{l}\mathbf{g}_j \sim \mathcal{N}\left(\mathbf{0}, \left(\sigma_i'^{(l)}\right)^2\mathbf{I}\right), \tag{88}$$

where $\left(\sigma_i'^{(l)}\right)^2 = \sum_j \left(\left(\boldsymbol{D}^{-1}\boldsymbol{A}\right)_{ij}^{l}\right)^2$. Therefore, similar to the lemma 3 in Wu et al. (2023),

$$\frac{1}{n} \leq \left(\sigma_i'^{(l)}\right)^2 \leq 1 \tag{89}$$

Denote $\sigma_i^{(l)} = \sigma_i'^{(l)}\sigma$. Then, the second term can be represented as

$$\left(\boldsymbol{D}^{-1}\boldsymbol{A}\right)_{ij}^{l}\mathbf{g}_j = \sigma_i^{(l)}\mathbf{g}_i^{(l)}, \tag{90}$$

where $\mathbf{g}_i^{(l)} \sim \mathcal{N}\left(\mathbf{0}, \mathbf{I}\right)$, which completes the proof. $\square$

Lemma A7 presents the feature distributions of a single node. However, it is not feasible to independently sample nodes from their feature distributions due to the high correlation among the $\mathbf{g}_i^{(l)}$ values for all nodes $i \in [n]$. In the following subsection, we will explore the analysis of this correlation and transform the correlated distribution into an independent one.

### A.5.2 CORRELATION OF THE NODE FEATURES

For convenience, we introduce the matrix $\boldsymbol{G}$ defined as

$$\boldsymbol{G} = \begin{pmatrix} \mathbf{g}_0 \\ \mathbf{g}_1 \\ \vdots \\ \mathbf{g}_{c-1} \end{pmatrix}, \tag{91}$$

which represents the aggregated matrix containing Gaussian noise from all nodes.

**Lemma A8.** *For any $i \in [n]$, by applying GC operation $l$ times, its aggregated noisy features are obtained as $\boldsymbol{G}_i^{(l)} = \left(\left(\boldsymbol{D}^{-1}\boldsymbol{A}\right)^{l}\boldsymbol{G}\right)_i$. With probability at least $1 - 1/\mathrm{Ploy}(n)$ over $A$ and $\{\varepsilon_i\}_{i\in[n]}$, we have*

$$\mathbb{E}\left[\mathrm{Var}\left(\boldsymbol{G}_0^{(l)}, \boldsymbol{G}_1^{(l)}, ..., \boldsymbol{G}_{n-1}^{(l)}\right)\right] = \frac{1}{2n^2}\|\boldsymbol{Q}^{(l)}\|_F^2, \tag{92}$$

*where $\boldsymbol{Q}^{(l)} \in \mathbb{R}^{n\times n}$ is the distance matrix of $\left(\boldsymbol{D}^{-1}\boldsymbol{A}\right)^{l}$ and $Q_{ij}^{(l)} = \|\left(\boldsymbol{D}^{-1}\boldsymbol{A}\right)_i^{l} - \left(\boldsymbol{D}^{-1}\boldsymbol{A}\right)_j^{l}\|_2$.*

*Proof.* Denoting $\boldsymbol{S} = \left(\boldsymbol{D}^{-1}\boldsymbol{A}\right)^{l}$, the covariance of random variable $\boldsymbol{G}_i^{(l)}$ and $\boldsymbol{G}_j^{(l)}$ is

$$
\begin{aligned}
\mathrm{Cov}\left(\boldsymbol{G}_i^{(l)}, \boldsymbol{G}_j^{(l)}\right) &= \mathrm{Cov}\left(\sum_{k\in[n]}\boldsymbol{S}_{ik}\boldsymbol{G}_k, \sum_{t\in[n]}\boldsymbol{S}_{jt}\boldsymbol{G}_t\right) \\
&= \sum_{k\in[n]}\sum_{t\in[n]}\boldsymbol{S}_{ik}\boldsymbol{S}_{jt}\,\mathrm{Cov}\left(\boldsymbol{G}_k, \boldsymbol{G}_t\right) \\
&= \sum_{k\in[n]}\boldsymbol{S}_{ik}\boldsymbol{S}_{jk}\,\mathrm{Cov}\left(\boldsymbol{G}_k, \boldsymbol{G}_k\right) \\
&= \boldsymbol{S}_i^T\boldsymbol{S}_j.
\end{aligned} \tag{93}
$$

Then, the expectation of the variance of these node features is

$$
\begin{aligned}
&\mathbb{E}\left[\mathrm{Var}\left(\boldsymbol{G}_0^{(l)}, \boldsymbol{G}_1^{(l)}, ..., \boldsymbol{G}_{n-1}^{(l)}\right)\right] \\
&= \frac{1}{n^2}\sum_{i<j}\mathbb{E}\left[\boldsymbol{G}_i^{(l)T}\boldsymbol{G}_i^{(l)}\right] - 2\mathbb{E}\left[\boldsymbol{G}_i^{(l)T}\boldsymbol{G}_j^{(l)}\right] + \mathbb{E}\left[\boldsymbol{G}_j^{(l)T}\boldsymbol{G}_j^{(l)}\right].
\end{aligned} \tag{94}
$$

Since $\mathbb{E}\left[\boldsymbol{G}_i^{(l)}\right] = \boldsymbol{0}$, the above equation becomes

$$
\begin{aligned}
&\frac{1}{n^2} \sum_{i<j} \mathrm{Cov}\left(\boldsymbol{G}_i^{(l)}, \boldsymbol{G}_i^{(l)}\right) - 2\,\mathrm{Cov}\left(\boldsymbol{G}_i^{(l)}, \boldsymbol{G}_j^{(l)}\right) + \mathrm{Cov}\left(\boldsymbol{G}_j^{(l)}, \boldsymbol{G}_j^{(l)}\right) \\
=&\frac{1}{n^2} \sum_{i<j} ||\boldsymbol{S}_i - \boldsymbol{S}_j||_2^2 = \frac{1}{2n^2} \sum_{i,j} ||\boldsymbol{S}_i - \boldsymbol{S}_j||_2^2 = \frac{1}{2n^2} ||\boldsymbol{Q}||_F^2,
\end{aligned}
\tag{95}
$$

where each $Q_{ij} = ||\boldsymbol{S}_i - \boldsymbol{S}_j||_2$. $\qquad\square$

Lemma A8 shows the expected variance of features among different nodes. By denoting $\bar{\mathbf{g}}^{(l)} = \mathbb{E}\left[\mathrm{Mean}\left(\boldsymbol{G}_0^{(l)}, \boldsymbol{G}_1^{(l)}, ..., \boldsymbol{G}_{n-1}^{(l)}\right)\right]$, we can dismantle $\boldsymbol{G}_i^{(l)}$ as

$$
\boldsymbol{G}_i^{(l)} = \bar{\mathbf{g}}^{(l)} + \frac{||\boldsymbol{Q}^{(l)}||_F}{\sqrt{2}n} \mathbf{g}_i^{(l)},
\tag{96}
$$

where $\mathbf{g}_i^{(l)} \sim \mathcal{N}\left(\boldsymbol{0}, \mathbf{I}\right)$ and $\forall i \neq j, \mathbb{E}\left[\mathbf{g}_i^{(l)T} \mathbf{g}_j^{(l)}\right] = 0$. Therefore, for each node $i \in [n]$, it aggregated features can be approximately represented as

$$
\boldsymbol{X}_i^{(l)} = \boldsymbol{\mu}_{\varepsilon_i}^{(l)}(i) + \sigma\left(\bar{\mathbf{g}}^{(l)} + \frac{||\boldsymbol{Q}^{(l)}||_F}{\sqrt{2}n} \mathbf{g}_i^{(l)}\right).
\tag{97}
$$

Subsequently, by employing a stronger density assumption, i.e., $\sum_t m_{\varepsilon_i t} m_{t \varepsilon_j} = \omega(\log n^2/n)$, we obtain the approximate formula to $||\boldsymbol{Q}^{(l)}||_2^2$.

**Lemma A9.** *Denote $\boldsymbol{Q}^{(l)} \in \mathbb{R}^{n \times n}$ is the distance matrix of $\left(\boldsymbol{D}^{-1}\boldsymbol{A}\right)^l$ and $Q_{ij}^{(l)} = ||\left(\boldsymbol{D}^{-1}\boldsymbol{A}\right)_i^l - \left(\boldsymbol{D}^{-1}\boldsymbol{A}\right)_j^l||_2$. Assume that each class possesses exactly $\frac{n}{c}$ nodes and $\sum_t m_{\varepsilon_i t} m_{t \varepsilon_j} = \omega(\log n^2/n)$. Then, with probability at least $1 - 1/\mathrm{Ploy}(n)$, we have*

$$
||\boldsymbol{Q}^{(l)}||_F^2 = \frac{n}{c} \sum_{k_1, k_2 \in [c]} ||\hat{\mathbf{m}}_{k_1}^{(l)} - \hat{\mathbf{m}}_{k_2}^{(l)}||_2^2 \left(1 \pm lo_n\left(1\right)\right),
\tag{98}
$$

*for $l > 1$.*

*Proof.* In HSBM, edges are constructed from the probability matrix $\mathbf{M} \in \mathbb{R}^{c \times c}$. Each edge, connecting nodes $i$ and $j$, is sampled from a Bernoulli distribution, with a parameter $m_{\varepsilon_i \varepsilon_j}$. We generalize $\mathbf{M}$ to the edge-wise edge generation matrix, i.e., $\boldsymbol{\mathcal{M}} \in \mathbb{R}^{n \times n}$, where $\boldsymbol{\mathcal{M}}_{ij} = m_{\varepsilon_i \varepsilon_j}$. Similarly, we also generalize the neighborhood distribution matrix $\hat{\mathbf{M}} \in \mathbb{R}^{c \times c}$ to a edge-wise matrix, i.e., $\hat{\boldsymbol{\mathcal{M}}} \in \mathbb{R}^{n \times n}$, where $\hat{\boldsymbol{\mathcal{M}}}_{ij} = \hat{m}_{\varepsilon_i \varepsilon_j}$. Then, the expectations of adjacency matrix and the normalized adjacency matrix are

$$
\mathbb{E}\left[\boldsymbol{A}\right] = \boldsymbol{\mathcal{M}}, \quad \mathbb{E}\left[\boldsymbol{D}^{-1}\boldsymbol{A}\right] = \frac{c}{n}\hat{\boldsymbol{\mathcal{M}}},
\tag{99}
$$

respectively. Consider the adjacency matrix $\boldsymbol{A}$ drawn from HSBM, where exactly $n/c$ nodes are in each class. Denote $\boldsymbol{\mathcal{A}} = \mathbb{E}\left[\boldsymbol{D}^{-1}\boldsymbol{A}\right]$. Then,

$$
\boldsymbol{\mathcal{A}} = \frac{c}{n}\begin{pmatrix} \hat{m}_{00}\boldsymbol{E}_{\frac{n}{c}} & \cdots & \hat{m}_{0(c-1)}\boldsymbol{E}_{\frac{n}{c}} \\ \vdots & \ddots & \vdots \\ \hat{m}_{(c-1)0}\boldsymbol{E}_{\frac{n}{c}} & \cdots & \hat{m}_{(c-1)(c-1)}\boldsymbol{E}_{\frac{n}{c}} \end{pmatrix},
\tag{100}
$$

where $\boldsymbol{E}_{\frac{n}{c}}$ is the all-ones matrix. Following $l$ iterations of multiplication, it yields

$$
\boldsymbol{\mathcal{A}}^l = \frac{c}{n}\begin{pmatrix} \hat{m}_{00}^{(l)}\boldsymbol{E}_{\frac{n}{c}} & \cdots & \hat{m}_{0(c-1)}^{(l)}\boldsymbol{E}_{\frac{n}{c}} \\ \vdots & \ddots & \vdots \\ \hat{m}_{(c-1)0}^{(l)}\boldsymbol{E}_{\frac{n}{c}} & \cdots & \hat{m}_{(c-1)(c-1)}^{(l)}\boldsymbol{E}_{\frac{n}{c}} \end{pmatrix}.
\tag{101}
$$

We now prove the concentration of the element of $l$-powered normalized adjacency matrix when $l > 1$, i.e., with a probability at least $1 - 1/\operatorname{Ploy}(n)$,

$$\left(\boldsymbol{D}^{-1}\boldsymbol{A}\right)^l_{ij} = \boldsymbol{\mathcal{A}}^l_{ij}\left(1 \pm lo_n\left(1\right)\right). \tag{102}$$

When $l = 2$, by utilizing the average node degree $\bar{D}$ to approximate the degree of each nodes, i.e., $\bar{D}_{ii} \approx \bar{D}$, the two-powered normalized adjacency matrix satisfies

$$\left(\boldsymbol{D}^{-1}\boldsymbol{A}\right)^2_{ij} = \frac{1}{\bar{D}^2}\sum_{k\in[n]} a_{ik}a_{kj}. \tag{103}$$

Here, $\sum_{k\in[n]} a_{ik}a_{kj}$ is the sum of independent Bernoulli random variables with a mean of $v_{ij} = \mathbb{E}\left[\sum_{k\in[n]} a_{ik}a_{kj}\right]$. Then, $v_{ij} = \frac{n}{c}\sum_{t\in[c]} m_{\varepsilon_i t}m_{t\varepsilon_j}$. By applying the Chernoff bound (Vershynin, 2018), we have

$$\mathbb{P}\left\{\left|\sum_{k\in[n]} a_{ik}a_{kj} - v_{ij}\right| \geq \delta v_{ij}\right\} \leq 2\exp\left(-C_1 v_{ij}\delta^2\right), \tag{104}$$

where $C_1 > 0$ is an absolute constant. We choose $\delta = \sqrt{\frac{C_2 \log n}{v_{ij}}}$, for a large constant $C_2 > 0$. Note that since $\sum_t m_{\varepsilon_i t}m_{t\varepsilon_j} = \omega(\log n^2/n)$, we have $v_{ij} = \omega(\log n^2)$ and $\delta = O\left(\frac{1}{\sqrt{\log n}}\right) = o_n(1)$. Then, with probability at least $1 - 1/\operatorname{Ploy}(n)$, we have

$$\sum_{k\in[n]} a_{ik}a_{kj} = \frac{n}{c}\sum_{t\in[c]} m_{\varepsilon_i t}m_{t\varepsilon_j}\left(1 \pm o_n(1)\right). \tag{105}$$

Therefore,

$$\begin{aligned}
\left(\boldsymbol{D}^{-1}\boldsymbol{A}\right)^2_{ij} &= \frac{c}{n}\sum_{t\in[c]} \hat{m}_{\varepsilon_i t}\hat{m}_{t\varepsilon_j}\left(1 \pm o_n\left(1\right)\right) \\
&= \frac{c}{n}\hat{m}^{(2)}_{\varepsilon_i\varepsilon_j}\left(1 \pm o_n\left(1\right)\right) \\
&= \boldsymbol{\mathcal{A}}^2_{ij}\left(1 \pm o_n\left(1\right)\right).
\end{aligned} \tag{106}$$

When $l > 2$, if

$$\left(\boldsymbol{D}^{-1}\boldsymbol{A}\right)^{l-1}_{ij} = \boldsymbol{\mathcal{A}}^{l-1}_{ij}\left(1 \pm (l-1)o\left(1\right)\right) \tag{107}$$

exists, with a probability at least $1 - 1/\operatorname{Ploy}(n)$, we have

$$\begin{aligned}
\left(\boldsymbol{D}^{-1}\boldsymbol{A}\right)^l_{ij} &= \sum_{k\in[n]}\left(\boldsymbol{D}^{-1}\boldsymbol{A}\right)_{ik}\boldsymbol{\mathcal{A}}^{l-1}_{kj}\left(1 \pm (l-1)o_n\left(1\right)\right) \\
&= \sum_{t\in[c]}\bar{D}^{-1}|\mathcal{C}_t \cap \Gamma_i|\frac{c}{n}\hat{m}^{(l-1)}_{t\varepsilon_j}\left(1 \pm (l-1)o_n\left(1\right)\right) \\
&= \frac{c}{n}\sum_{t\in[c]}\hat{m}_{\varepsilon_i t}\hat{m}^{(l-1)}_{t\varepsilon_j}\left(1 \pm lo\left(1\right)\right) \\
&= \frac{c}{n}\hat{m}^{(l)}_{\varepsilon_i\varepsilon_j}\left(1 \pm lo_n\left(1\right)\right) \\
&= \boldsymbol{\mathcal{A}}^l_{ij}\left(1 \pm lo_n\left(1\right)\right).
\end{aligned} \tag{108}$$

Note that, here, the error term $l \cdot o_n(1)$ increases linearly as $l$ increases. By a assumption that for each nodes, its neighborhood distribution is exactly the sampled one, i.e., $\forall i \in [n], t \in [c]$, $\bar{D}^{-1}|\mathcal{C}_t \cap \Gamma_i| = \hat{m}_{\varepsilon_i t}$, this error term decreases to $o_n(1)$, which is not correlated the $l$.

Then,

$$\begin{aligned}
\|\boldsymbol{Q}\|^2_F &= \sum_{i,j\in[n]}\|\left(\boldsymbol{D}^{-1}\boldsymbol{A}\right)^l_i - \left(\boldsymbol{D}^{-1}\boldsymbol{A}\right)^l_j\|^2_2 \\
&= \sum_{i,j\in[n]}\frac{c}{n}\|\hat{\boldsymbol{m}}^{(l)}_{\varepsilon_i} - \hat{\boldsymbol{m}}^{(l)}_{\varepsilon_j}\|^2_2\left(1 \pm lo_n\left(1\right)\right) \\
&= \frac{n}{c}\sum_{k,t\in[c]}\|\hat{\boldsymbol{m}}^{(l)}_k - \hat{\boldsymbol{m}}^{(l)}_t\|^2_2\left(1 \pm lo_n\left(1\right)\right).
\end{aligned} \tag{109}$$

$\square$

### A.5.3 PROOF OF THEOREM 4

**Theorem A1** (A general version of Theorem 4). *Given* $(\boldsymbol{X}, \boldsymbol{A}) = \mathrm{HSBM}(n, c, \sigma, \{\boldsymbol{\mu}_k\}, \boldsymbol{\eta}, \mathbf{M}, \{\mathbf{0}\})$, *by assuming that* $\forall k, t \in [c]$, $\bar{D}_k = \bar{D}_t$, *with probability of at least* $1 - 1/\mathrm{Ploy}(n)$, *we have*

$$\mathbb{E}_{i \in \mathcal{C}_t} \mathbb{P}\left[\zeta_i(k)\right] = \Phi\left(\frac{\gamma}{2\sigma} \frac{F_{tk}^{(l)}}{\varsigma_n^{(l)}} + \frac{\sigma}{\gamma} \frac{\varsigma_n^{(l)}}{F_{tk}^{(l)}} \ln\left(\frac{\eta_t}{\eta_k}\right)\right), \tag{110}$$

*over data* $\boldsymbol{X}^{(l)} = \left(\boldsymbol{D}^{-1}\boldsymbol{A}\right)^l \boldsymbol{X}$, *where*

$$F_{tk}^{(l)} = \frac{n}{||\boldsymbol{Q}^{(l)}||_F} ||\hat{\mathbf{m}}_k^{(l)} - \hat{\mathbf{m}}_t^{(l)}||_2, \tag{111}$$

$\boldsymbol{Q}^{(l)} \in \mathbb{R}^{n \times n}$ *is the distance matrix of* $\left(\boldsymbol{D}^{-1}\boldsymbol{A}\right)^l$, $Q_{ij}^{(l)} = ||\left(\boldsymbol{D}^{-1}\boldsymbol{A}\right)_i^l - \left(\boldsymbol{D}^{-1}\boldsymbol{A}\right)_j^l||_2$, *and* $\varsigma_n^{(l)} = (1 \pm lo_n(1))$.

*Then, by assuming that each class possesses exactly* $\frac{n}{c}$ *nodes,* $\forall k, t \in [c]$, $\eta_t = \eta_k$, $\sum_t m_{tk} \asymp \sum_t m_{kt}$, *and* $\sum_t m_{\varepsilon_i t} m_{t \varepsilon_j} = \omega(\log n^2/n)$, *for* $l > 1$, *Eq.* (111) *can be further represented as*

$$F_{tk}^{(l)} = \sqrt{\frac{c||\hat{\mathbf{m}}_k^{(l)} - \hat{\mathbf{m}}_t^{(l)}||_2^2}{\sum_{k_1, k_2 \in [c]} ||\hat{\mathbf{m}}_{k_1}^{(l)} - \hat{\mathbf{m}}_{k_2}^{(l)}||_2^2} \frac{\bar{D}}{\log n}}. \tag{112}$$

*Proof.* According to Appendix A.5.2, the aggregated features can be reformulated as

$$\boldsymbol{X}_i^{(l)} = \boldsymbol{\mu}_{\varepsilon_i}^{(l)}(i) + \sigma \bar{\mathbf{g}}^{(l)} + \bar{\sigma} \mathbf{g}_i^{(l)}, \tag{113}$$

where $\bar{\sigma} = \frac{||\boldsymbol{Q}^{(l)}||_F}{\sqrt{2}n} \sigma$, $\mathbf{g}_i^{(l)} \sim \mathcal{N}(\mathbf{0}, \mathbf{I})$, and $\forall i \neq j, \mathbb{E}\left[\mathbf{g}_i^{(l)T} \mathbf{g}_j^{(l)}\right] = 0$. For each node $i \in [n]$, the probability of node is misclassified into category $k$ is

$$\mathbb{P}\left[\zeta_i(k)\right] = \mathbb{P}\left[\varphi_{\varepsilon_i}^{(l)} > \varphi_k^{(l)}\right], \tag{114}$$

where $\varphi_k^{(l)} = \eta_t \cdot \mathbb{P}[\boldsymbol{X}_i^{(l)} \mid y = k]$. Then,

$$\varphi_{\varepsilon_i}^{(l)} > \varphi_k^{(l)}$$

$$\iff \eta_{\varepsilon_i} \exp\left(-\frac{||\boldsymbol{X}_i^{(l)} - \sigma\bar{\mathbf{g}}^{(l)} - \boldsymbol{\mu}_{\varepsilon_i}^{(l)}||^2}{2\bar{\sigma}^2}\right) > \eta_k \exp\left(-\frac{||\boldsymbol{X}_i^{(l)} - \sigma\bar{\mathbf{g}}^{(l)} - \boldsymbol{\mu}_k^{(l)}||^2}{2\bar{\sigma}^2}\right)$$

$$\iff \ln \eta_{\varepsilon_i} - \frac{||\boldsymbol{X}_i^{(l)} - \sigma\bar{\mathbf{g}}^{(l)} - \boldsymbol{\mu}_{\varepsilon_i}^{(l)}||^2}{2\bar{\sigma}^2} > \ln \eta_k - \frac{||\boldsymbol{X}^{(l)} - \sigma\bar{\mathbf{g}}^{(l)} - \boldsymbol{\mu}_k^{(l)}||^2}{2\bar{\sigma}^2}$$

$$\iff 2\bar{\sigma}^2 \ln\left(\frac{\eta_{\varepsilon_i}}{\eta_k}\right) + ||\boldsymbol{\mu}_{\varepsilon_i}^{(l)}||^2 \varsigma_n^{(l)} + 2\bar{\sigma}\boldsymbol{\mu}_{\varepsilon_i}^{(l)T} \bar{\mathbf{g}}_i^{(l)} \varsigma_n^{(l)} - 2\boldsymbol{\mu}_{\varepsilon_i}^{(l)T} \boldsymbol{\mu}_k^{(l)} \varsigma_n^{(l)} - 2\bar{\sigma}^2 \boldsymbol{\mu}_k^{(l)T} \bar{\mathbf{g}}_i^{(l)} \varsigma_n^{(l)} + ||\boldsymbol{\mu}_k^{(l)}||^2 > 0$$

$$\iff 2\bar{\sigma}^2 \ln\left(\frac{\eta_{\varepsilon_i}}{\eta_k}\right) \varsigma_n^{(l)} + \gamma'^2 ||\hat{\mathbf{m}}_k^{(l)} - \hat{\mathbf{m}}_{\varepsilon_i}^{(l)}||^2 \varsigma_n^{(l)} > 2\bar{\sigma}\gamma' \sum_{o \in [c]} \left(\hat{m}_{ko}^{(l)} - \hat{m}_{\varepsilon_i o}^{(l)}\right) \frac{\boldsymbol{\mu}_o^T}{||\boldsymbol{\mu}_o||} \bar{\mathbf{g}}_i^{(l)}$$

$$\iff \frac{\bar{\sigma}}{\gamma' ||\hat{\mathbf{m}}_k^{(l)} - \hat{\mathbf{m}}_{\varepsilon_i}^{(l)}||} \ln\left(\frac{\eta_{\varepsilon_i}}{\eta_k}\right) \varsigma_n^{(l)} + \frac{\gamma'}{2\bar{\sigma}} ||\hat{\mathbf{m}}_k^{(l)} - \hat{\mathbf{m}}_{\varepsilon_i}^{(l)}|| \varsigma_n^{(l)} > \sum_{o \in [c]} \frac{\hat{m}_{ko}^{(l)} - \hat{m}_{\varepsilon_i o}^{(l)}}{||\hat{\mathbf{m}}_k^{(l)} - \hat{\mathbf{m}}_{\varepsilon_i}^{(l)}||} \frac{\boldsymbol{\mu}_o^T}{||\boldsymbol{\mu}_o||} \mathbf{g}_i. \tag{115}$$

Similar to Eq. (69), we have

$$\sum_{o \in [c]} \frac{\hat{m}_{ko}^{(l)} - \hat{m}_{\varepsilon_i o}^{(l)}}{||\hat{\mathbf{m}}_k^{(l)} - \hat{\mathbf{m}}_{\varepsilon_i}^{(l)}||} \frac{\boldsymbol{\mu}_o^T}{||\boldsymbol{\mu}_o||} \mathbf{g}_i \sim \mathcal{N}(0, 1). \tag{116}$$

Then, the probability

$$
\begin{aligned}
\mathbb{P}\left[\zeta_i(k)\right] &= \Phi\left(\frac{\gamma'}{2\bar{\sigma}}||\hat{\mathbf{m}}_k^{(l)} - \hat{\mathbf{m}}_{\varepsilon_i}^{(l)}||\varsigma_n^{(l)} + \frac{\bar{\sigma}}{\gamma'||\hat{\mathbf{m}}_k^{(l)} - \hat{\mathbf{m}}_{\varepsilon_i}^{(l)}||}\ln\left(\frac{\eta_{\varepsilon_i}}{\eta_k}\right)\varsigma_n^{(l)}\right) \\
&= \Phi\left(\frac{\gamma}{2\sigma}\frac{F_{\varepsilon_i k}^{(l)}}{\varsigma_n^{(l)}} + \frac{\sigma}{\gamma}\frac{\varsigma_n^{(l)}}{F_{\varepsilon_i k}^{(l)}}\ln\left(\frac{\eta_{\varepsilon_i}}{\eta_k}\right)\right),
\end{aligned}
\tag{117}
$$

where

$$
F_{tk}^{(l)} = \frac{n}{||\boldsymbol{Q}^{(l)}||_F}||\hat{\mathbf{m}}_k^{(l)} - \hat{\mathbf{m}}_t^{(l)}||_2
\tag{118}
$$

and $\varsigma_n^{(l)} = (1 \pm lo_n(1))$.

By assuming that each class possesses exactly $\frac{n}{c}$ nodes and $\sum_t m_{\varepsilon_i t} m_{t\varepsilon_j} = \omega(\log n^2/n)$, the above formula can be further represented as

$$
F_{tk}^{(l)} = \sqrt{\frac{nc||\hat{\mathbf{m}}_k^{(l)} - \hat{\mathbf{m}}_t^{(l)}||_2^2}{\sum_{k_1, k_2 \in [c]}||\hat{\mathbf{m}}_{k_1}^{(l)} - \hat{\mathbf{m}}_{k_2}^{(l)}||_2^2}},
\tag{119}
$$

for $l > 1$, according to Lemma A9. Since $\sum_t m_{tk} \asymp \sum_t m_{kt}$, for each $k_1, k_2 \in [c]$, $\frac{1}{c^2}(\sum_t m_{tk_2})(\sum_t m_{k_1 t}) \geq \sum_t m_{k_1 t} m_{tk_2} = \omega\left(\log n^2/n\right)$. Then, we have $\bar{p}_k = \omega\left(\log n/\sqrt{n}\right)$ and $\bar{D} = \omega\left(\log n\sqrt{n}\right)$. Then, Eq. (120) can be represented as

$$
F_{tk}^{(l)} = \sqrt{\frac{c||\hat{\mathbf{m}}_k^{(l)} - \hat{\mathbf{m}}_t^{(l)}||_2^2}{\sum_{k_1, k_2 \in [c]}||\hat{\mathbf{m}}_{k_1}^{(l)} - \hat{\mathbf{m}}_{k_2}^{(l)}||_2^2}}\frac{\bar{D}}{\log n},
\tag{120}
$$

Note that the $O_n(1)$ error term can be combined with $\varsigma_n$. Then, we have

$$
\mathbb{E}_{i \in \mathcal{C}_t}\mathbb{P}\left[\zeta_i(k)\right] = \Phi\left(\frac{\gamma}{2\sigma}\frac{F_{tk}^{(l)}}{\varsigma_n^{(l)}} + \frac{\sigma}{\gamma}\frac{\varsigma_n^{(l)}}{F_{tk}^{(l)}}\ln\left(\frac{\eta_t}{\eta_k}\right)\right).
\tag{121}
$$

$\square$

## A.6 PROOF OF PROPOSITION 1

*Proof.* When $\hat{\mathbf{M}}$ is non-singular, denote $\hat{\mathbf{M}}\hat{\mathbf{M}}^{-1} = I$. Then, according to the compatibility of matrix norms, we have

$$
||\hat{\mathbf{M}}^{-1}||_F \geq c/||\hat{\mathbf{M}}||_F > 0
\tag{122}
$$

and

$$
||\hat{\mathbf{m}}_k^{(l)} - \hat{\mathbf{m}}_t^{(l)}||_2 \geq ||\hat{\mathbf{m}}_k - \hat{\mathbf{m}}_t||_2||\hat{\mathbf{M}}^{-1}||_F^{1-l} > 0.
\tag{123}
$$

Thus, we obtain that each separability satisfies

$$
0 < F_{tk}^{(l)}.
\tag{124}
$$

Then, since

$$
\frac{\log n}{\sqrt{c\bar{D}}}\sum_{t,k \in [c]}F_{tk}^{(l)} \geq \sum_{t,k \in [c]}\left(\frac{\log n}{\sqrt{c\bar{D}}}F_{tk}^{(l)}\right)^2 = 1,
\tag{125}
$$

we have

$$
\sum_{t,k \in [c]}F_{tk}^{(l)} \geq \sqrt{c}\bar{D}/\log n.
\tag{126}
$$

$\square$

# B  ADDITIONAL RESULTS FOR SYNTHETIC DATA

## B.1  MORE VISUALIZATIONS WITH DIFFERENT $a$

Here, we present additional results for different values of $a$. Reminding that the family of heterophily patterns is constructed based on the parameter $a$, where $\hat{\mathbf{M}}_{ij}$ is $a$ when $j = i$, $2a$ when $j = i + i$, and $\frac{1}{3} - a$ otherwise. Thus, $a$ is chosen from the range of $[0, \frac{1}{3}]$. Specifically, we select six representative heterophily patterns to elaborate the effect of different heterophily patterns, where $a \in [0.04, 0.05, 0.1, 0.12, 0.26, 0.3]$. Note that $\varsigma_n \approx 1.2$ in these heterophily patterns.

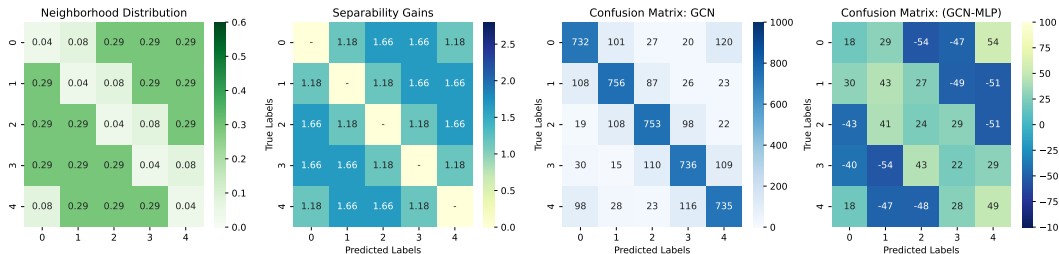

Figure 3: Example of mixed heterophily with $a = 0.04$. The accuracy of GCN is 74.24.

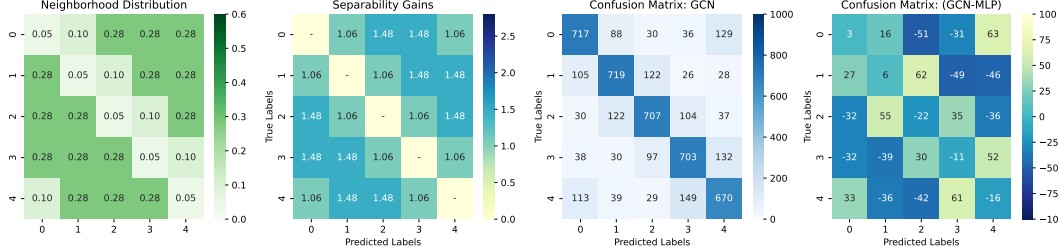

Figure 4: Example of mixed heterophily with $a = 0.05$. The accuracy of GCN is 70.32.

Figs. 3 and 4 depict two instances of mixed heterophily with values of $a$ equal to $0.04$ and $0.05$, respectively. It is evident that these two patterns exhibit very similar neighborhood distributions, resulting in similar separability gains. When $a = 0.04$, nodes within classes 1 and 2 possess a smaller gain, i.e., $1.18$, while nodes within classes 1 and 3 possess a larger gain, i.e., $1.66$. By considering the $\varsigma_n \approx 1.2$, this implies that nodes within class 1 and 2 are more indistinguishable, while nodes within class 1 and 3 are more distinguishable. This presumption can be verified from the differences in the confusion matrices of GCN and MLP. As can be observed, when a GC operation is applied, the number of nodes belonging to class 1 but misclassified into class 2 increases by 27, while the number of nodes misclassified into class 3 decreases by 49. When $a = 0.05$, the separability gains are slightly smaller than those when $a = 0.04$. This results in the overall non-diagonal elements of the differences in the confusion matrices having relatively larger values compared to the situation when $a = 0.04$.

Figs. 5 and 6 show two examples of bad heterophily with $a = 0.1$ and $a = 0.12$, where the neighborhood distributions are similar across different classes. Their separability gains are all smaller than $\varsigma_n$. Therefore, after applying a GC operation, the separability of each class pair degrades. The graph with $a = 0.12$ possesses smaller separability gains than the graph with $a = 0.1$, leading to a larger damage to the separability. For example, when applying a GC operation, the number of nodes belonging to class 0 that are misclassified to class 1 increases 79 and 87 when $a = 0.1$ and $a = 0.12$, respectively.

Figs. 7 and 8 present two examples of good heterophily with $a = 0.26$ and $a = 0.3$, where the neighborhood distributions differ significantly across different classes. In both cases, their separability gains exceed $\varsigma_n$. Consequently, by applying a GC operation, the separability between all pairs of classes increases, resulting in an overall accuracy improvement. The graph generated with

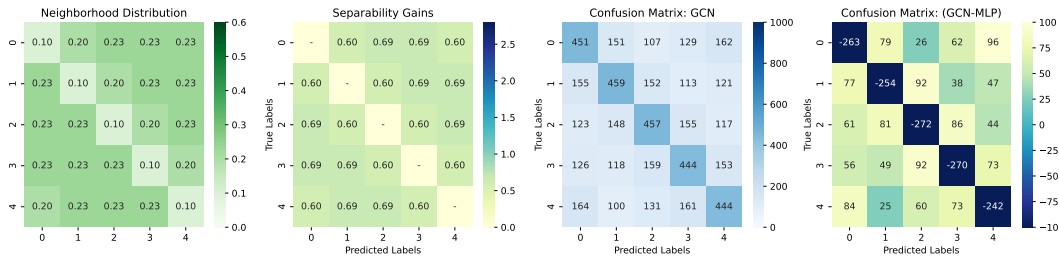

Figure 5: Example of bad heterophily with $a = 0.1$. The accuracy of GCN is 45.10.

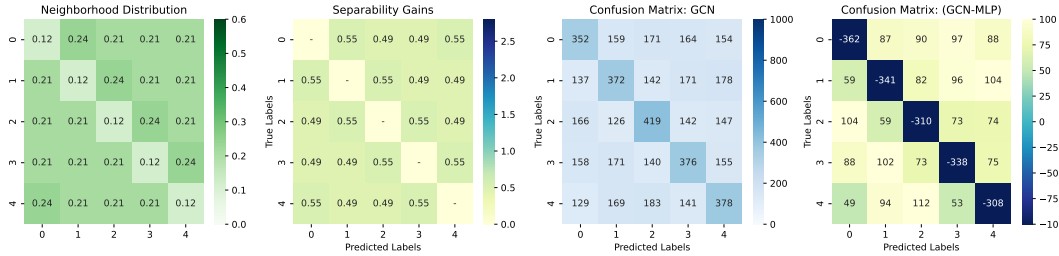

Figure 6: Example of bad heterophily with $a = 0.12$. The accuracy of GCN is 37.94.

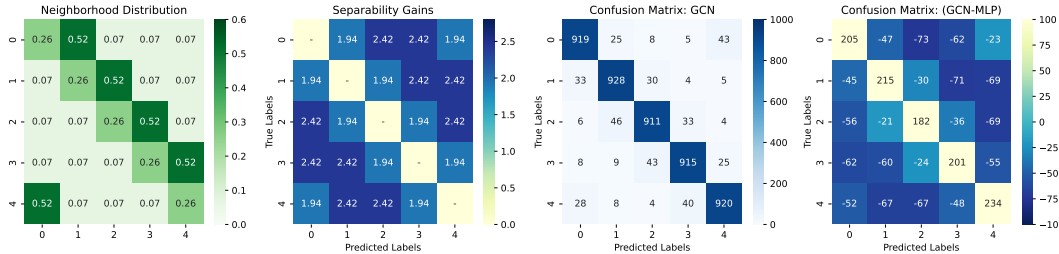

Figure 7: Example of good heterophily with $a = 0.26$. The accuracy of GCN is 91.86.

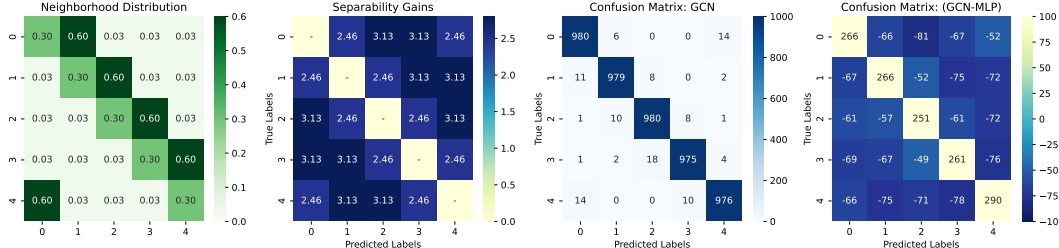

Figure 8: Example of good heterophily with $a = 0.3$. The accuracy of GCN is 97.80.

$a = 0.3$ exhibits an even more distinct neighborhood distribution. As a result, the gains from the GC operation are more substantial, as evidenced by the differences in the confusion matrix.

In summary, the above observations further verify the effectiveness of Theorem 1 and shed light on the underlying mechanism driving the influence of heterophily.

## B.2 MORE HETEROPHILY PATTERNS

The synthetic graphs in Sec. 5.1 are all generated from a family of heterophily patterns with parameter $a$. Here, we validate our theory on two more families of heterophily patterns.

**Homophilous Family** is a family of homophily patterns, where the diagonal elements possess different values from the other elements. It neighborhood distributions are

$$\hat{\mathbf{M}} = \begin{pmatrix} a_1 & b_1 & b_1 & b_1 & b_1 \\ b_1 & a_1 & b_1 & b_1 & b_1 \\ b_1 & b_1 & a_1 & b_1 & b_1 \\ b_1 & b_1 & b_1 & a_1 & b_1 \\ b_1 & b_1 & b_1 & b_1 & a_1 \end{pmatrix}, \tag{127}$$

where $a_1 \in [0, 1]$ and $b_1 = (1 - a_1)/4$.

**Group Family** is a family of group heterophily patterns. There are two groups of classes, i.e., group $(0, 1)$ and group $(2, 3, 4)$. The intra-group classes group possess similar neighborhood distribution, while inter-group classes possess different neighborhood distribution. It neighborhood distributions are

$$\hat{\mathbf{M}} = \begin{pmatrix} a_2 & b_2 & c_2 & c_2 & c_2 \\ b_2 & a_2 & c_2 & c_2 & c_2 \\ d_1 & d_1 & a_1 & b_1 & b_1 \\ d_1 & d_1 & b_1 & a_1 & b_1 \\ d_1 & d_1 & b_1 & b_1 & a_1 \end{pmatrix}, \tag{128}$$

where $a_2 \in [0, 0.2]$, $b = a + 0.2$, $c = (0.8 - 2 * a)/3$, and $d = (0.6 - 3 * a)/2$.

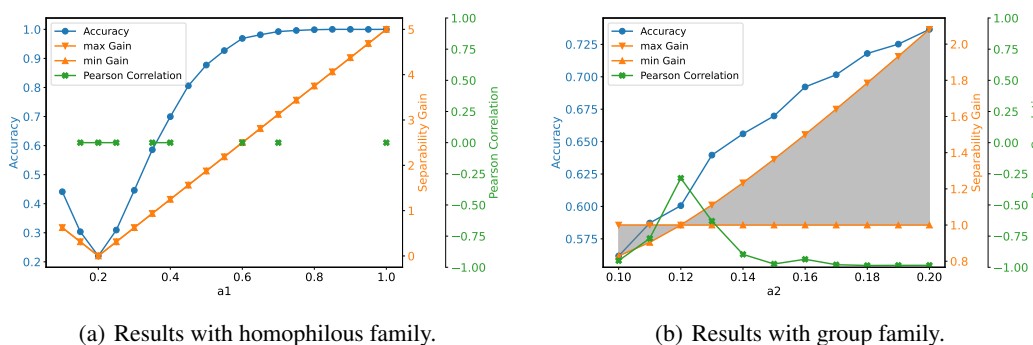

(a) Results with homophilous family.

(b) Results with group family.

Figure 9: Results of additional heterophily patterns.

Fig. 9 show the results of the above two families of heterophily patterns, with different $a_1$ and $a_2$. In Fig. 9(a), curves of maximum gain and minimum gain coincide, which indicates the separability gains are all the same. Therefore, the Pearson correlation between separability gains and confusion matrix gains is meaningless under such circumstance. Similarly, a same situation occurs in Fig. 9(b), when $a_2 = 0.12$. Overall, the separability gains are positively correlated with the accuracy. The separability gains and confusion matrix gains are highly negative correlated when the separability gains are not with one value.

### B.3 LARGE-SCALE SYNTHETIC GRAPHS

Here, we verify our theory on the large-scale synthetic graphs generated from our HSBM model. Similarly, we adopt the settings in Sec. 5.1 with the exception of assigning the number of nodes as $n = 100,000$. Since the averaged node degree for each class is still 25, the expected number of edges in these large-scale graphs is $2,500,000$. As can be observed in Fig. 10, the results of the large-scale synthetic graphs are similar to those in Fig. 2(a), where the number of nodes is only 1000. This observation further validates the effectiveness of our theory.

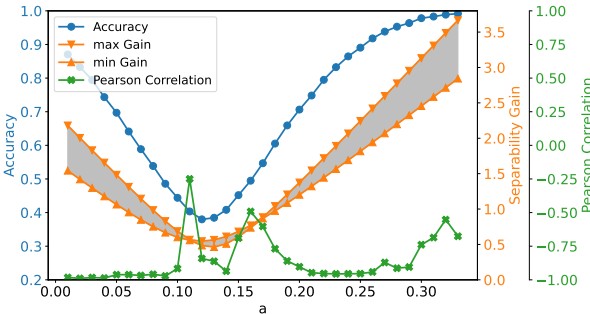

Figure 10: Results on large-scale synthetic data with the number of nodes as $n = 100,000$.

## B.4 PRECISION PROBLEM WITH STACKING MULTIPLE GC OPERATIONS

To verify the *precision limitation* issue, we introduce two additional data formats: float32 and float128. Note that in our development environment, float32, float64, and float128 can accurately represent the numbers up to 7, 16, and 19 digits, respectively. [1]

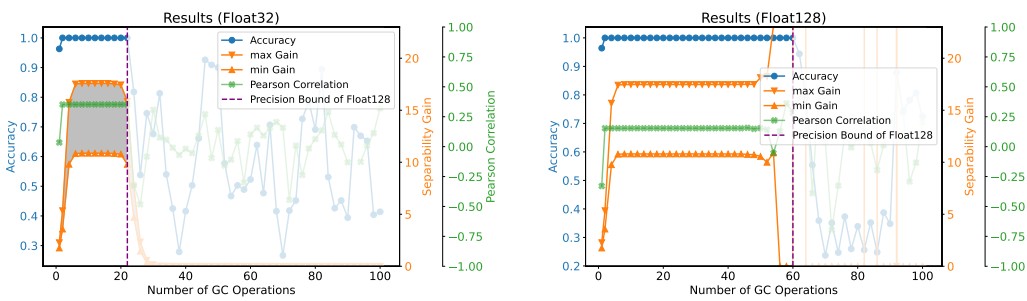

(a) Results on the implementation of float32.

(b) Results on the implementation of float128.

Figure 11: Results with stacking multiple GC operations, implemented on different data formats.

As can be observed in Figs. 2(d), 11(a) and 11(b), both of their accuracy curves and separability gain regions exhibit similar behavior. They maintain the similar high values during an initial period and then suddenly drop at some point. However, they experience this sudden drop at different numbers of GC operations: 22 for float32, 50 for float64, and 60 for float128. This observation suggests that the suddenly decrement of accuracy is caused by the *precision limitation* of the corresponding data format.

For a deeper investigation, Fig. 12 visualizes the variations in three variables under the implementation of different data formats. In Fig. 12, averaged STD represents the standard variance calculated among different dimension of features, i.e., $\tilde{\sigma}^{(l)}$; average Mean Distance represents the average of the distance of the feature within different classes $\tilde{\gamma}^{(l)} \approx \text{Mean}\left(||\tilde{\boldsymbol{\mu}}_i^{(l)} - \tilde{\boldsymbol{\mu}}_i^{(l)}||\right)$. Reminding that for Gaussian node features, $\frac{\tilde{\gamma}^{(l)}}{2\tilde{\sigma}^{(l)}}$ can represents the separability when stacking $l$ GC operations as stated in Theorem 1).

Under the implementations of different data formats, the $\tilde{\sigma}^{(l)}$ and $\tilde{\gamma}^{(l)}$ are decreasing with approximate the same rate during an initial period. Then, for the results of float64, both the standard variance does not decline when their value approach approximately $10^{-15}$, i.e., when the number of GC operations grows to 50, both of them cannot be precisely represented. Therefore, as is expected, the accuracy of GCN decreases suddenly. When we increase the precision of floating-point data

---

[1] This part of the code is implemented in *NumPy* (https://numpy.org/doc/stable/index.html), where *np.float128* provides only as much precision as *np.longdouble*, 64 bits in our development environment.

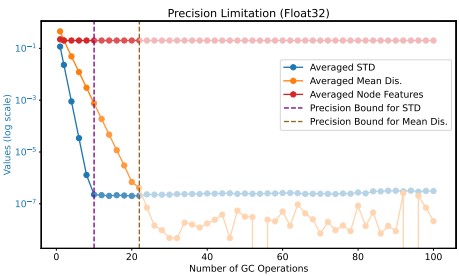 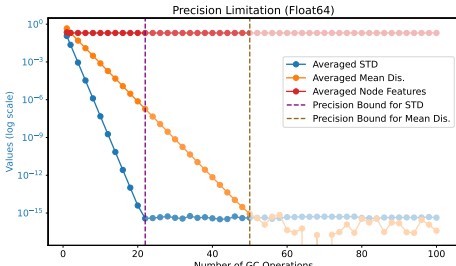

(a) Precision problem of Float32 data format.  (b) Precision problem of Float64 data format.

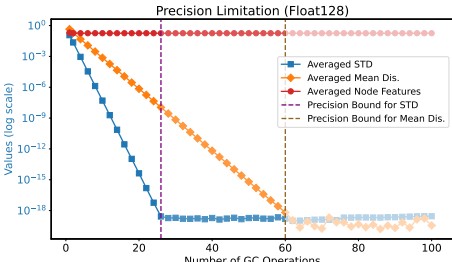

(c) Precision problem of Float128 data format.

Figure 12: Precision bounds for different data formats.

format to float128, the difference of the node features can be obtained until they degrade to $10^{-18}$. Then, the point of descent can be postponed. On the contrary, when utilizing float32, the difference cannot be obtained when they degrade to $10^{-7}$.

Therefore, this observation demonstrates that as the number of GC operations increases, although the nodes remain distinguishable, there will be a sudden decrease in accuracy, due to the *precision problem* associated with the data format.

## C ADDITIONAL RESULTS FOR REAL-WORLD DATA

### C.1 DATASET STATISTICS

We employ eight node classification datasets to verify the effectiveness of our theory, including two citation network (i.e. Cora (Yang et al., 2016) and Arxiv-year (Lim et al., 2021)), two Wikipedia networks (i.e. Chameleon and Squirrel) (Pei et al., 2020), a co-occurrence network (i.e. Actor (Pei et al., 2020)), a co-purchasing network (i.e., Amazon-ratings (Platonov et al., 2023)), a crowdsourcing coworking network(i.e., Workers (Platonov et al., 2023)), and a patent network(i.e., Snap-patents) (Platonov et al., 2023). Their statistics are provided in Tabs. 2 and 3. Their homophily ratios are calculated following Pei et al. (2020). For each dataset, we randomly selected 60%/20%/20% nodes to construct the training/validation/testing sets.

Table 2: Dataset Statistics (Part One).

|  | Nodes | Edges | node features | classes | avg degree | homophily ratio |
|---|---|---|---|---|---|---|
| Cora | 2708 | 10556 | 1433 | 7 | 3.90 | 0.83 |
| Chameleon | 2277 | 62792 | 2325 | 5 | 27.55 | 0.25 |
| Actor | 7600 | 30019 | 932 | 5 | 3.93 | 0.16 |
| Amazon-ratings | 24492 | 93050 | 300 | 5 | 7.60 | 0.32 |
| Squirrel | 5201 | 396846 | 2089 | 5 | 76.27 | 0.22 |
| Workers | 11758 | 519000 | 10 | 2 | 44.14 | 0.59 |
| Arxiv-year | 169343 | 1166243 | 128 | 5 | 6.89 | 0.26 |
| Snap-patents | 2923922 | 13975791 | 269 | 5 | 4.78 | 0.22 |

Table 3: Dataset Statistics (Part Two).

|  | avg degree (class) | nodes (class) |
|---|---|---|
| Cora | [4.35, 4.74, 4.36, 3.46, 3.73, 3.64, 3.66] | [351, 217, 418, 818, 426, 298, 180] |
| Chameleon | [21.94, 24.76, 27.32, 26.50, 39.17] | [456, 460, 453, 521, 387] |
| Actor | [4.13, 3.90, 3.95, 3.77, 4.01] | [ 853, 1337, 1630, 1815, 1965] |
| Amazon-ratings | [3.72, 3.97, 3.82, 3.56, 3.15] | [6560, 9010, 5678, 2183, 1061] |
| Squirrel | [46.38, 61.86, 73.97, 88.06, 111.13] | [1042, 1040, 1039, 1040, 1040] |
| Workers | [40.45, 74.27] | [9192, 2566] |
| Arxiv-year | [9.71, 18.95, 14.21, 7.64, 3.63] | [31971, 21189, 37781, 29799, 48603] |
| Snap-patents | [4.80, 7.15, 7.84, 6.48, 3.13] | [559377, 609501, 551116, 579876, 624052] |

## C.2 DETAILED SETTINGS

We employ a two-layered MLP for the baseline network and incorporate one GC operation to the last layer to construct the corresponding GCN. The cross-entropy loss is utilized as the loss function, and Adam Kingma & Ba (2015) optimizer is employed. We conduct the same grid search to find the best group of hyperparameters. The number of hidden neurons, learning rate, weight decay rate, and dropout rate are obtained by the grid-search strategy. These hyperparameters are search in:

- number of hidden neurons: $\text{hidden} \in [16, 32, 64, 128, 256]$;
- learning rate: $\text{lr} \in [0.001, 0.005, 0.01]$;
- weight decay rate: $\text{wd} \in [0, 1e-5, 5e-4, 1e-4]$;
- dropout rate: $\text{dropout} \in [0, 0.2, 0.5]$.

## C.3 VISUALIZATIONS OF THE RESULTS ON REAL-WORLD DATA

Here, we present detailed visualizations of the neighborhood distribution, topological noise, separability gains, and the learned confusion matrices on these five real-world datasets for a comprehensive understanding.

**Discussion on Cora.** Fig. 13 presents the visualization of results for Cora, a typical homophilous dataset. As can be observed, its neighborhood distribution matrix is a diagonally dominant matrix, where the diagonal elements consistently possess large values. Besides, Cora exhibits relatively low topological noise. Therefore, the separability gain for each class pair is significant. According to the difference in the confusion matrix of GCN and MLP, we can observe that it effectively reduces the number of misclassified nodes across nearly all class pairs.

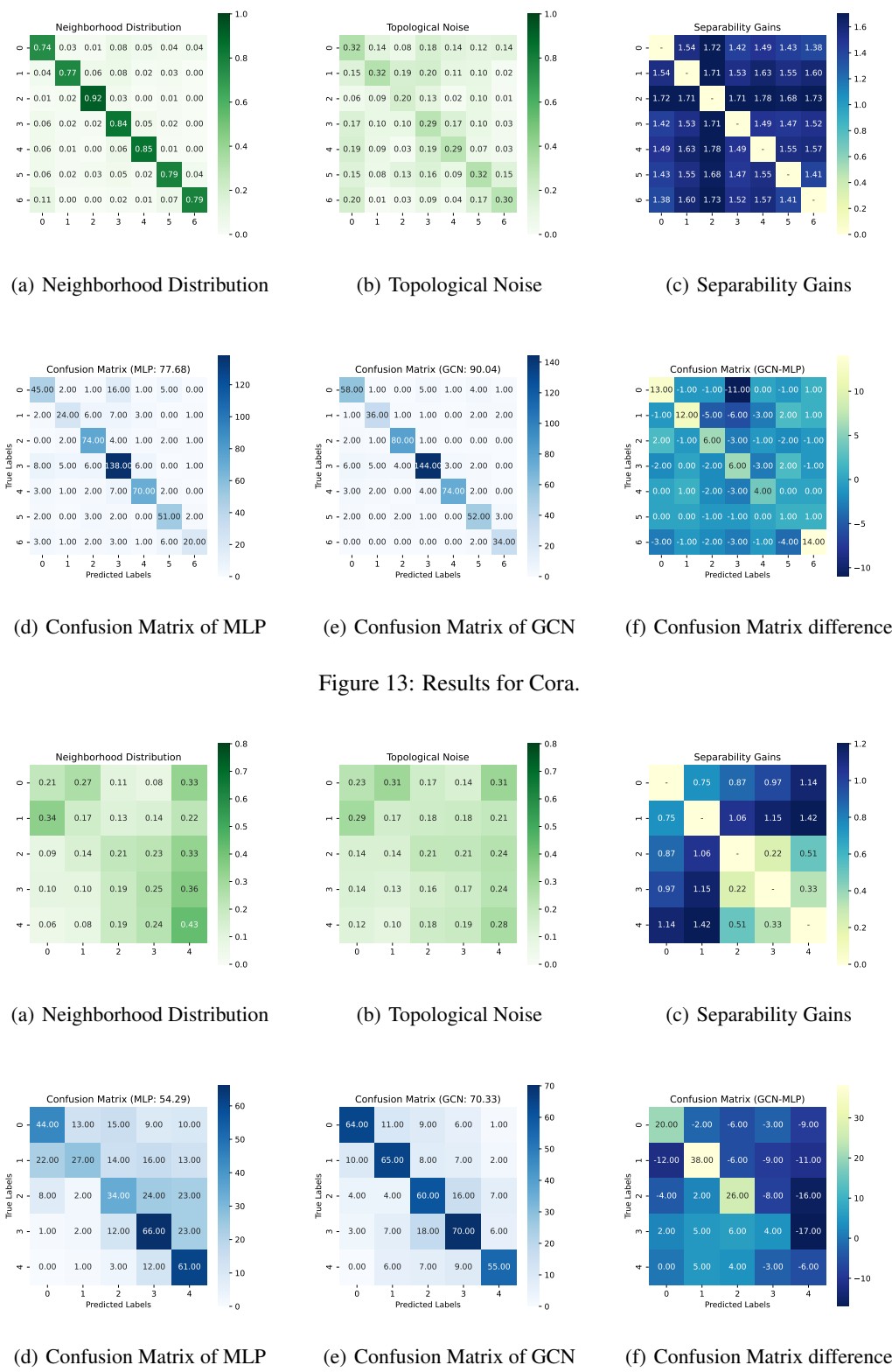

(a) Neighborhood Distribution      (b) Topological Noise      (c) Separability Gains

(d) Confusion Matrix of MLP      (e) Confusion Matrix of GCN      (f) Confusion Matrix difference

Figure 13: Results for Cora.

(a) Neighborhood Distribution      (b) Topological Noise      (c) Separability Gains

(d) Confusion Matrix of MLP      (e) Confusion Matrix of GCN      (f) Confusion Matrix difference

Figure 14: Results for Chameleon.

**Discussion on Chameleon.** Fig. 14 presents the results for Chameleon, a typical heterophilous dataset. Based on its neighborhood distribution matrix, it exhibits similarities to the group heterophily patterns shown in Appendix B.2. Chameleon is divided into two class groups, namely $(0, 1)$ and $(2, 3, 4)$, where intra-group classes share similar neighborhood distributions, while inter-group classes have distinct ones. This characteristic is reflected in its separability gains matrix.

As claimed in Theorem 1, when nodes within different class possess varying proportions, the class with a larger number of samples experience more significant benefits, while the class with fewer samples derive fewer benefits and may even encounter a decrease in the classification accuracy. However, their collective impact consistently aligns with the corresponding separability gain. Therefore, when examining the difference matrix as shown in Fig. 14(f), it is essential to consider the symmetric elements $(i, j)$ and $(j, i)$ together to assess the separability gain effect between classes $i$ and $j$.

As observed, the classification results for all class pairs possess positive gains. Even in the case of the pair $(2, 3)$, which has a separability gain of $0.22$, there is a positive classification gain ($2$ more samples are successfully classified). Thus, Chameleon can be regarded as exhibiting a favorable heterophily pattern with the $\varsigma_n \approx 0.2$.

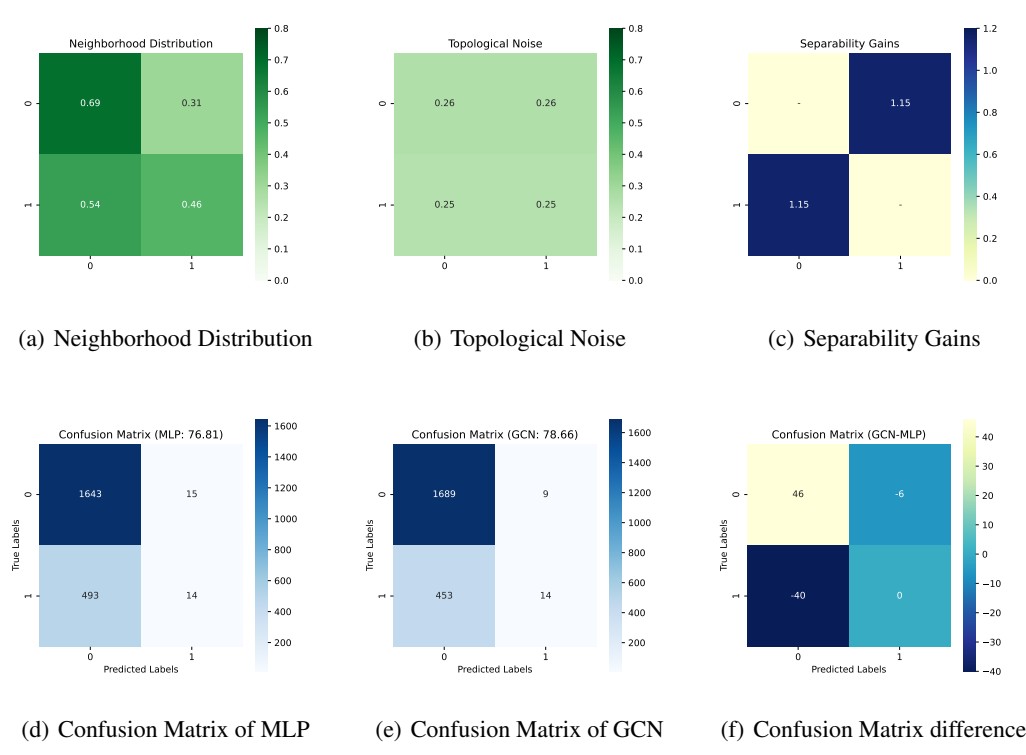

(a) Neighborhood Distribution     (b) Topological Noise     (c) Separability Gains

(d) Confusion Matrix of MLP     (e) Confusion Matrix of GCN     (f) Confusion Matrix difference

Figure 15: Results for Workers.

**Discussion on Workers.** Fig. 15 illustrates the results for Workers, a heterophilous dataset with only two node categories. As can be observed, in Workers, nodes within the two classes exhibit similar neighborhood distribution. However, due to the large averaged degree (approximately 44.14), the separability gain for the two classes is still significant.

As shown in Fig. 15(f), the differences in the confusion matrix between GCN and MLP align with the corresponding separability gains shown in Fig. 17(c). Note that, as shown in Tab. 3, the number of nodes within class $0$ (9192) is larger than the number of nodes within class $1$ (2566). Therefore, both the MLP and GCN tend to classify nodes into class $0$. According to our Theorems 1 and 2, the class with a larger number of samples, i.e., class $0$, experience more significant benefits. Thus, the elements $(1, 0)$ exhibits a larger decrease, while $(0, 1)$ possesses a smaller one.

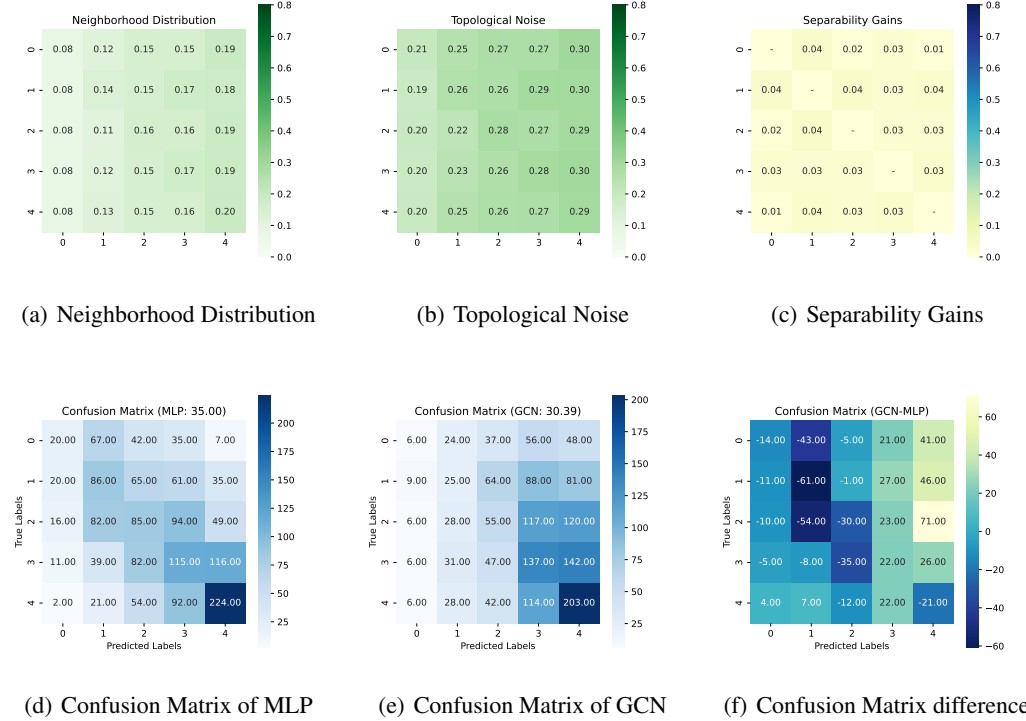

Figure 16: Results for Actor.

**Discussion on Actor.** Fig. 16 illustrates the results for Actor, a heterophilous dataset. As can be observed, in Actor, different classes possess similar neighborhood distributions, and its topological noise is significant. Therefore, its separability gains are minimal, as shown in Fig. 16(c), leading to a deterioration in classification performance when utilizing GCN. Thus, Actor exhibits a bad heterophily pattern.

Eq. (4) in Theorem 2 suggests that when the separability gains are less than $\varsigma_n$, classes with more nodes experience a more significant decrease in separability. In contrast, classes with fewer nodes may experience a slight decline or even an increase in separability. As observed in Fig. 16(f), although GCN exhibits worse overall results compared to MLP, it misclassifies more nodes into classes 3 and 4, while misclassifying fewer nodes into the other three classes. This observation is consistent with the results of Theorem 2.

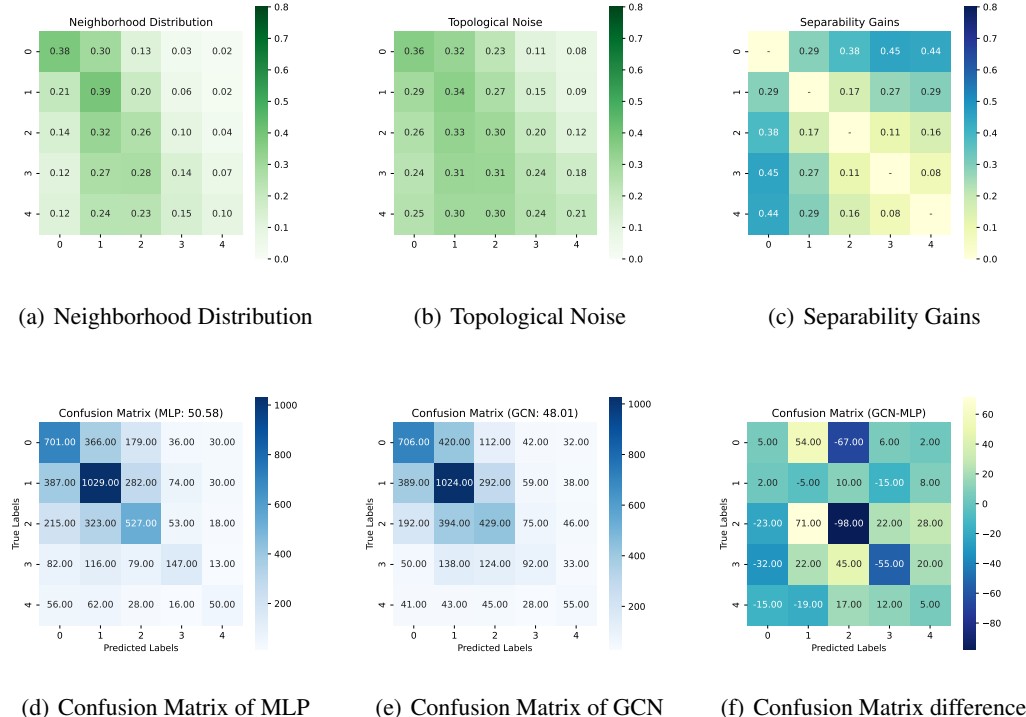

Figure 17: Results for Amazon-ratings.

**Discussion on Amazon-ratings.** Fig. 17 visualizes the results of Amazon-ratings, a heterophilous dataset. As can be observed from Fig. 17(c), by assuming that $\varsigma_n \approx 0.25$, the topological information should boost the classification between class pairs $(0, 3)$, $(0, 4)$, $(0, 2)$, $(0, 1)$, $(1, 4)$, and $(1, 3)$, while damage the others. Therefore, Amazon-ratings possesses a mixed heterophily pattern.

As shown in Fig. 17(f), the differences in the confusion matrix between GCN and MLP largely align with the corresponding separability gains shown in Fig. 17(c). The only exception is the class pair $(0, 1)$, whose separability should increase but experiences a decrease. This discrepancy may be attributed to the distribution of node features. As evident in Fig. 17(d), samples within classes 0 and 1 appear to possess similar node features. In general, the results of Amazon-ratings are consistent with our theoretical findings in Theorem 2.

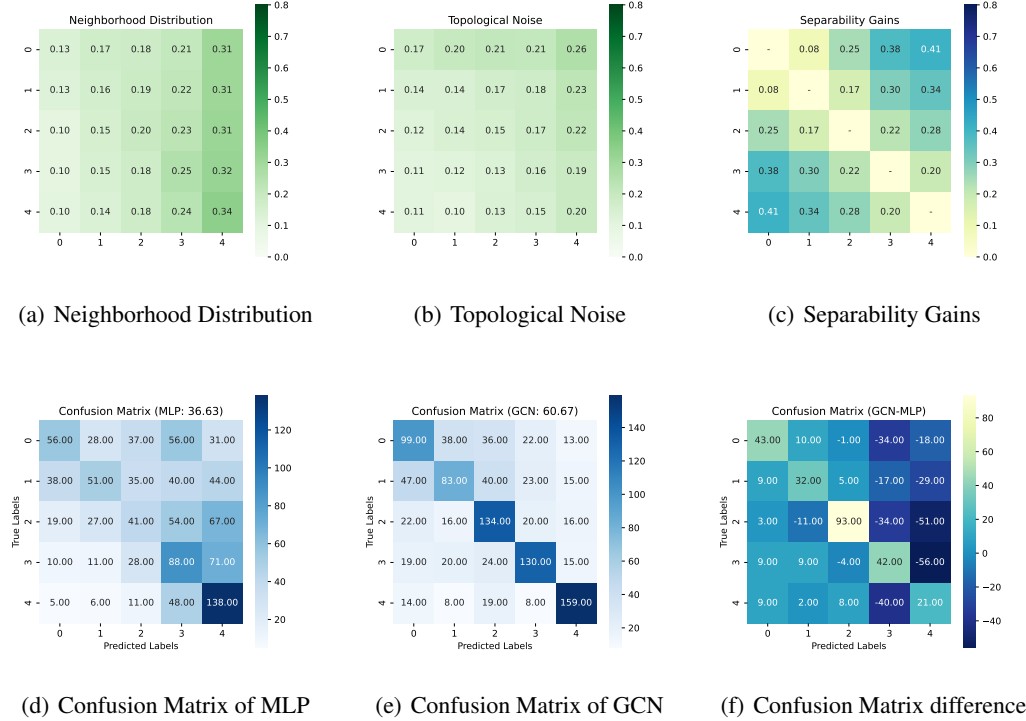

Figure 18: Results for Squirrel.

**Discussion on Squirrel.** Fig. 18 illustrates the results of Squirrel, a heterophilous dataset. As can be observed from Fig. 18(a), Squirrel possess very similar neighborhood distributions across different classes. However, as indicated in Tab. 3, Squirrel has a notably high average degree of 76.27. Therefore, it achieves relatively significant separability gains. By considering the $\varsigma_n \approx 0.12$, Squirrel exhibits a mixed heterophily pattern.

Based on the differences in the confusion matrix between GCN and MLP, as presented in Fig. 18(f), when using GCN, it enhances the classification between every pair of classes except for pair $(0, 1)$. This is because pair $(0, 1)$ has a very small separability gain of 0.08, which is smaller than $\varsigma_n$. This observation aligns with our theoretical results in Theorem 2.

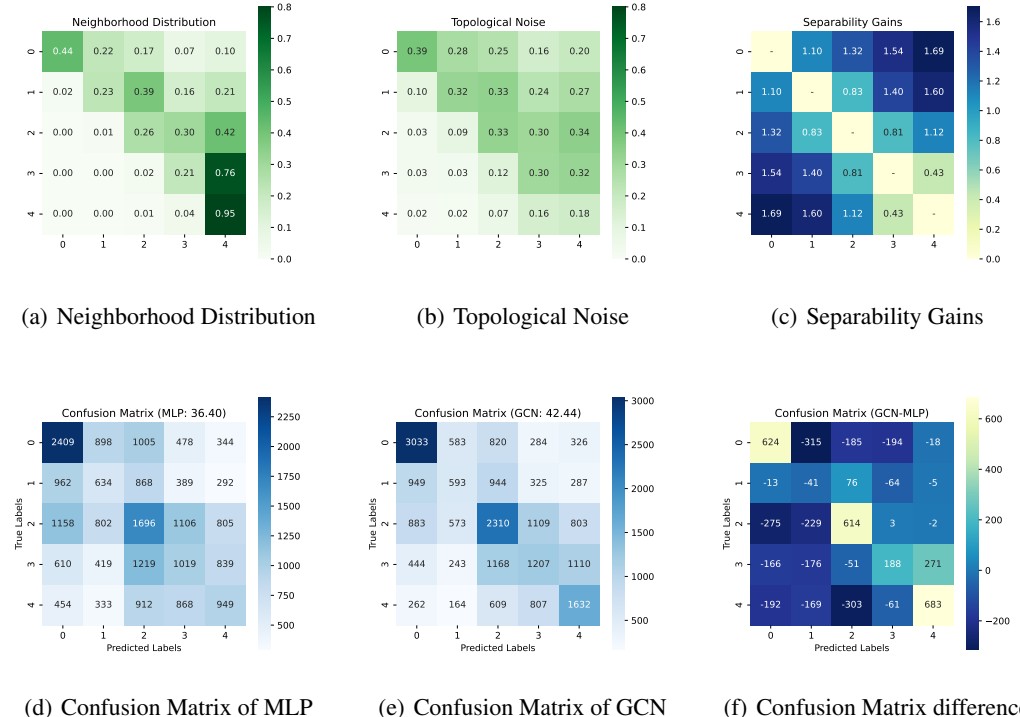

(a) Neighborhood Distribution   (b) Topological Noise   (c) Separability Gains

(d) Confusion Matrix of MLP   (e) Confusion Matrix of GCN   (f) Confusion Matrix difference

Figure 19: Results for Arxiv-year.

**Discussion on Arxiv-year.** Fig. 19 visualizes the results of Arxiv-year, a large-scale heterophilous dataset, which possesses 169,343 nodes and 1,166,243 edges. As can be observed from Fig. 19(c), by assuming that $\varsigma_n \approx 0.5$, the topological information should damage the classification between class pair $(3, 4)$, while boost the others. Therefore, Arxiv-year exhibits a mixed heterophily pattern.

In Fig. 19(f), we jointly consider the symmetric elements $(i, j)$ and $(j, i)$ together to assess the separability gain effect between classes $i$ and $j$, according to Theorem 1. As can be observed, the differences in the confusion matrix between GCN and MLP align with the corresponding separability gains shown in Fig. 19(c), i.e., only the class pair $(3, 4)$ exhibits an increment of the misclassified nodes. This observation in the large-scale real-world dataset validates the effectiveness of our theory.

**Discussion on Snap-patents.** Fig. 20 visualizes the results of Snap-patents, a large-scale heterophilous dataset with 2,923,922 nodes and 13,975,791 edges. As can be observed from Fig. 20(c), by assuming that $\varsigma_n \approx 0.9$, the topological information should damage the classification between class pairs $(0, 1)$, $(1, 2)$, $(2, 3)$, and $(3, 4)$, while boost the others. Therefore, Snap-patents possesses a mixed heterophily pattern.

As shown in Fig. 20(f), the differences in the confusion matrix between GCN and MLP largely align with the corresponding separability gains shown in Fig. 20(c). The only exception is the class pair $(0, 1)$, whose separability should decrease but experiences an increase. This discrepancy may be attributed to the distribution of node features. As evident in Fig. 20(d), MLP tends to classify nodes into class 0, 1, and 3, which indicates that the distribution of nodes features (learned node embeddings) are not similar to Gaussian features, leading to this sightly inconsistency. In general, the results of Snap-patents are largely consistent with our theoretical findings in Theorem 2.

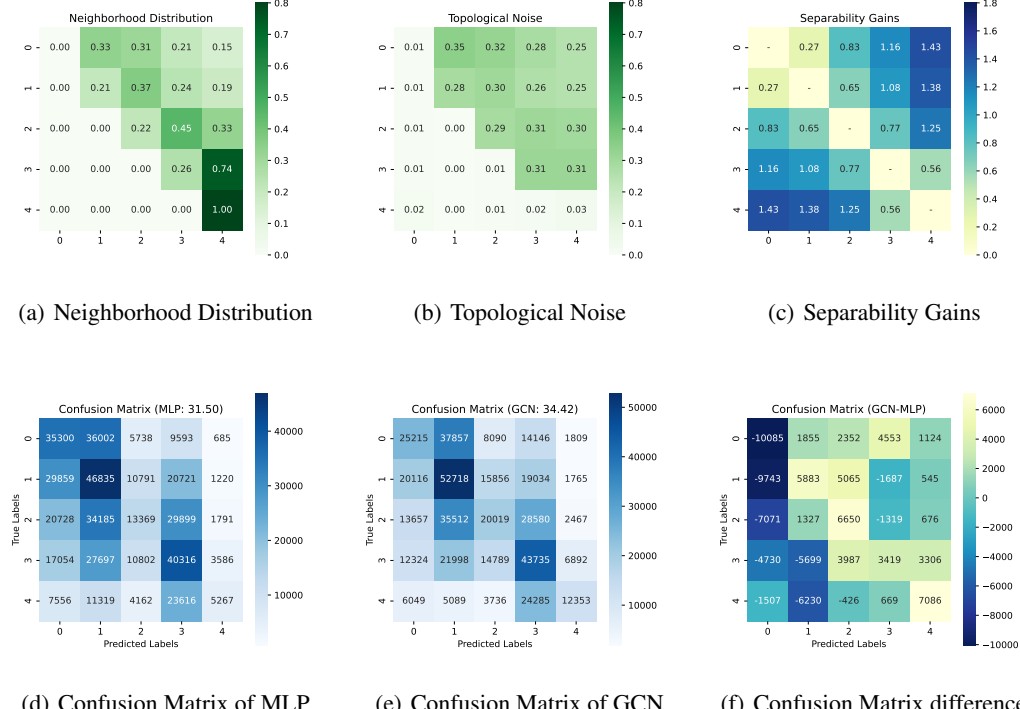

Figure 20: Results for Snap-Patents.

# D NEURAL NETWORK INSTANCE OF BAYES CLASSIFIER

Here, we show that there exists a one-layered fully connected network, which can model the Bayes classifier for the raw node features $\boldsymbol{X}$ or the aggregated node features $\tilde{\boldsymbol{X}}$. Therefore, our theoretical results can be achievable through MLPs.

**Proposition A1** (Instance for the Raw Features). *Given* $(\boldsymbol{X}, \boldsymbol{A}) =$ HSBM $(n, c, \sigma, \{\boldsymbol{\mu}_k\}, \boldsymbol{\eta}, \mathbf{M}, \{\boldsymbol{\Delta}_i\})$, *the Fully Connected Network* $\boldsymbol{Y} = \mathrm{softmax}(\boldsymbol{X}\boldsymbol{W}_1 + \boldsymbol{b}_1)$ *is an instance of the Bayes classifier over* $\boldsymbol{X}$, *which stated in Lemma A1, where the parameter matrix*

$$\boldsymbol{W}_1 = (\boldsymbol{\mu}_0^T \quad \boldsymbol{\mu}_1^T \quad \cdots \quad \boldsymbol{\mu}_{c-1}^T) \tag{129}$$

*and*

$$\boldsymbol{b}_1 = \sigma^2(\ln \eta_0, \ln \eta_1, \cdots, \ln \eta_{c-1}). \tag{130}$$

*Proof.* Given $\boldsymbol{W}_1$ in Eq. (129), we have

$$\boldsymbol{X}\boldsymbol{W}_1 + \boldsymbol{b}_1 = \begin{pmatrix} \langle \mathbf{X}_1, \boldsymbol{\mu}_0 \rangle + \sigma^2 \ln \eta_0 & \langle \mathbf{X}_1, \boldsymbol{\mu}_1 \rangle + \sigma^2 \ln \eta_1 & \cdots & \langle \mathbf{X}_1, \boldsymbol{\mu}_{c-1} \rangle + \sigma^2 \ln \eta_{c-1} \\ \langle \mathbf{X}_2, \boldsymbol{\mu}_0 \rangle + \sigma^2 \ln \eta_0 & \langle \mathbf{X}_2, \boldsymbol{\mu}_1 \rangle + \sigma^2 \ln \eta_1 & \cdots & \langle \mathbf{X}_2, \boldsymbol{\mu}_{c-1} \rangle + \sigma^2 \ln \eta_{c-1} \\ \vdots & \vdots & & \vdots \\ \langle \mathbf{X}_n, \boldsymbol{\mu}_0 \rangle + \sigma^2 \ln \eta_0 & \langle \mathbf{X}_n, \boldsymbol{\mu}_1 \rangle + \sigma^2 \ln \eta_1 & \cdots & \langle \mathbf{X}_n, \boldsymbol{\mu}_{c-1} \rangle + \sigma^2 \ln \eta_{c-1} \end{pmatrix} \tag{131}$$

For each node $i \in [n]$, for any $k_1, k_2 \in [c]$ and $k_1 \neq k_2$, if

$$\langle \boldsymbol{X}_i, \boldsymbol{\mu}_{k_1} \rangle + \sigma^2 \ln \eta_{k_1} \geq \langle \boldsymbol{X}_i, \boldsymbol{\mu}_{k_2} \rangle + \sigma^2 \ln \eta_{k_2}, \tag{132}$$

we have

$$\boldsymbol{Y}_{ik_1} \geq \boldsymbol{Y}_{ik_2}. \tag{133}$$

Therefore, the prediction of this fully connected network is

$$\text{pred}\left(\boldsymbol{X}_i\right) = \underset{k}{\arg\max}\left(\boldsymbol{Y}_{ik}\right) = \underset{k}{\arg\max}\left(\langle\boldsymbol{X}_i, \boldsymbol{\mu}_k\rangle + \sigma^2 \ln \eta_k\right), \tag{134}$$

which completed the proof. $\qquad\square$

**Proposition A2** (Instance for the Aggregated Features). *Given* $(\boldsymbol{X}, \boldsymbol{A})$ = $\text{HSBM}\left(n, c, \sigma, \{\boldsymbol{\mu}_k\}, \boldsymbol{\eta}, \mathbf{M}, \{\mathbf{0}\}\right)$, *when* $\forall k, t \in [c]$, $\bar{D}_k = \bar{D}_t$, *the Fully Connected Network* $\mathbf{Y} = \text{softmax}(\tilde{\boldsymbol{X}}\boldsymbol{W}_2 + \boldsymbol{b}_2)$ *is an instance of the Bayes classifier over* $\tilde{\boldsymbol{X}} = \boldsymbol{D}^{-1}\boldsymbol{A}\boldsymbol{X}$, *which is stated in Lemma A6, where the parameter matrix*

$$\mathbf{W}_2 = (\tilde{\boldsymbol{\mu}}_0^T \quad \tilde{\boldsymbol{\mu}}_1^T \quad \cdots \quad \tilde{\boldsymbol{\mu}}_{c-1}^T) \tag{135}$$

*and*

$$\boldsymbol{b}_1 = \tilde{\sigma}^2(\ln \eta_0, \ln \eta_1, \cdots, \ln \eta_{c-1}). \tag{136}$$

*Proof.* When $\bar{D}_k = \bar{D}_t$, we have $\tilde{\sigma} = \tilde{\sigma}_k$, for all $k \in [c]$. Then,

$$\tilde{\varphi}_{k_1} \geq \tilde{\varphi}_{k_2} \iff \langle\boldsymbol{x}, \tilde{\boldsymbol{\mu}}_{k_1}\rangle + \tilde{\sigma}^2 \ln \eta_{k_1} \geq \langle\boldsymbol{x}, \tilde{\boldsymbol{\mu}}_{k_2}\rangle + \tilde{\sigma}^2 \ln \eta_{k_2}. \tag{137}$$

Similarly, by replacing $\boldsymbol{\mu}$ and $\sigma$ in the proof of Proposition A1 to $\tilde{\boldsymbol{\mu}}$ and $\tilde{\sigma}$, we can complete the proof. $\qquad\square$

# E  SIGNIFICANCE OF THE TWO-CLASSES SEPARABILITIES

**Proposition A3** (Upper Bound of the Overall Error Rate). *The overall error rate is bounded as*

$$\mathbb{E}_{i\in[n]}\mathbb{P}\left[\boldsymbol{X}_i \text{ is misclassified}\right] \leq \sum_{t\in[c]}\sum_{k\in[c],k<t}(\eta_t + \eta_k)(1 - S(t,k)), \tag{138}$$

*where* $1 - S(t,k) \in [0,1]$ *can be viewed as the error rate when only considering the classification of classes* $t$ *and* $k$.

*Proof.*

$$\begin{aligned}
\mathbb{E}_{i\in[n]}\mathbb{P}\left[\boldsymbol{X}_i \text{ is misclassified}\right] &= \sum_{k\in[c]}\eta_k\mathbb{E}_{i\in\mathcal{C}_k}\mathbb{P}\left[\boldsymbol{X}_i \text{ is misclassified}|\varepsilon_i = k\right] \\
&= \sum_{k\in[c]}\eta_k\mathbb{E}_{i\in\mathcal{C}_k}\mathbb{P}\left[\cup_{t\neq k}\zeta_i^c(t)\right] \\
&\leq \sum_{k\in[c]}\eta_k\sum_{t\neq k}\mathbb{E}_{i\in\mathcal{C}_k}\mathbb{P}\left[\cup_t\zeta_i^c(t)\right] \\
&= \sum_{t\in[c]}\sum_{k\in[c],k<t}(\eta_t + \eta_k)(1 - S(t,k)),
\end{aligned} \tag{139}$$

where $\zeta_i^c(t)$ is the complement of $\zeta_i(t)$. $\qquad\square$

Proposition A3 illustrates that the overall error rate of the node classification is bounded by the weighted sum of the error rates of the two-class subtasks. Increasing the separability degrades the upper bound of the overall error rate, while decreasing the separability increases its upper bound. Thus, we can get the following corollary for the good/bad heterophily patterns.

**Corollary A1.** *When applying a GC operation, good heterophily patterns degrade the upper bound of overall error rate, while bad heterophily patterns increase it.*

*Proof.* According to Proposition A3,

$$\mathbb{E}_{i\in[n]}\mathbb{P}\left[\boldsymbol{X}_i \text{ is misclassified}\right] = \sum_{t\in[c]}\sum_{k\in[c],k<t}(\eta_t + \eta_k)(1 - S(t,k)). \tag{140}$$

Good heterophily patterns increase each $S(t,k)$, thereby reducing this upper bound; whereas bad heterophily patterns decrease each $S(t,k)$, thereby increasing this upper bound. $\qquad\square$

# F    SUGGESTIONS ON LEARNING HETEROPHILIOUS GRAPHS

Our work introduces a fresh perspective on heterophily and over-smoothing, coupled with a novel GNN analytical framework, which may further boost the development of innovative methods. For example, several possible insights are as follows.

Firstly, according to Theorem 2, different GNN aggregators may successfully classify different sub-sets of nodes. Therefore, when learning on graphs with heterophily, it may be beneficial to adaptively select specific aggregators for individual nodes, e.g. Luan et al. (2022); Javaloy et al. (2023). Furthermore, this result can also explain why the combination of ego- and neighbor-embeddings can establish a stable and competitive baseline across diverse heterophily patterns Zhu et al. (2020); Platonov et al. (2023).

Secondly, according to Theorem 3, the inconsistency in neighborhood distribution poses a potential challenge when handling heterophilious graphs. Therefore, it may be beneficial to mitigate the influence of topological noise within each class, probably through the design of an attention scheme.

Thirdly, according to Theorem 4, GNN with varying numbers of layers may successfully classify different subsets of nodes. Therefore, it may be beneficial to explore information from high-order neighborhoods, e.g., Zhu et al. (2020).

Besides, we believe that our theoretical results may provide additional inspiration to the community for designing heterophily-specific GNNs, enabling them to successfully handle graphs with diverse heterophily patterns.

