# OpenReview forum: "Understanding Heterophily for Graph Neural Networks"
_ICLR.cc/2024/Conference — Submitted to ICLR 2024_

### Official Review · Reviewer_sSgQ · 2023-10-29

**Soundness:** 3 good
**Presentation:** 2 fair
**Contribution:** 3 good
**Rating:** 6
**Confidence:** 4

**Summary:**

This paper focuses on heterophily in Graph Neural Networks (GNNs) and its impact on node classification. It proposes the Heterophilous Stochastic Block Models (HSBM) to incorporate different heterophily patterns in GNNs. The authors present theoretical analyses of the effects of heterophily patterns on GNNs, considering graph convolution operations. They reveal that the separability gains in GNNs depend on the Euclidean distance of neighborhood distributions and the averaged node degree. The paper also discusses the impact of topological noise on separability and the influence of stacking multiple graph convolution operations. The theoretical results are supported by experiments on synthetic and real-world data.

**Strengths:**

The authors provide sufficient theoretical analysis.

**Weaknesses:**

See below.

**Questions:**

1. Two related works need to be discussed [1,2].

2. "Its primary objective is to accurately categorize samples into their respective classes while distinguishing them from samples belonging to other classes." I think this is the same as the statement about intra- and inter-class node distinguishability in [1]. What is the difference and advantage of your "ideal Bayes classifier" over the optimal Bayes classifier in [1]?

Although this paper gets some similar conclusions as the previous paper about heterophily, I don't think it reduce a lot about is contribution. Thus, I'll give it a 6.



[1] When do graph neural networks help with node classification: Investigating the homophily principle on node distinguishability. arXiv preprint arXiv:2304.14274.

[2] Demystifying Structural Disparity in Graph Neural Networks: Can One Size Fit All?. arXiv preprint arXiv:2306.01323.

---

> ### Author Response · Authors · 2023-11-17
> **Response to Reviewer sSgQ**
>
> Dear reviewer, we sincerely appreciate your comments and valuable suggestions. We respond to your detailed comments as below:
>
> > Comment 1: Two related works need to be discussed [1,2].
>
> **Response to Comment 1:** Thank you for your reminder about these two recent related works, i.e., [1,2]. Both of them, as well as [3] (mentioned in our paper), analyze the heterophily problems of GNNs based on the CSBM models. [3] presents a preliminary result of GCN regarding the impact of different p and q on the two-block CSBMs, while [1] delves into the results when utilizing low-pass/high-pass GNNs. [2] evaluates the performance of GNNs on graphs containing nodes with homophily and heterophily patterns simultaneously. However, these efforts only focus on the simple binary node classifications, which is not general enough to analyze diverse heterophily patterns. Our work analyzes the impact of complicated heterophily patterns in the multi-class classifications, which has not been explored to the best of our knowledge. According to your suggestion, we have compared these two related works in Section 2 (Paragraph: Contextual Stochastic Block Models) in our revised manuscript.
>
>
> > Comment 2: Its primary objective is to accurately categorize samples into their respective classes while distinguishing them from samples belonging to other classes. I think this is the same as the statement about intra- and inter-class node distinguishability in [1]. What is the difference and advantage of your "ideal Bayes classifier" over the optimal Bayes classifier in [1]?
>
> **Response to Comment 2:** The "ideal Bayes classifier" in our paper is the same as the "optimal Bayes classifier" in [1]. It is worth noting that it is the commonly utilized classifier in analyzing the behavior of GNNs based on random graph data models, e.g., [1,2,4,5], due to its simplicity and effectiveness in predicting the performance score on the generated data. Therefore, we utilize it to construct our analytical framework and further state that our theoretical results based on the Bayes classifier can be achievable in practice, as detailed in Appendix D.
>
> *[1] When do graph neural networks help with node classification: Investigating the homophily principle on node distinguishability. arXiv preprint arXiv:2304.14274.*
>
> *[2] Demystifying Structural Disparity in Graph Neural Networks: Can One Size Fit All?. arXiv preprint arXiv:2306.01323.*
>
> *[3] Is homophily a necessity for graph neural networks? In ICLR, 2022.*
>
> *[4] A non-asymptotic analysis of over-smoothing in graph neural networks. In ICLR, 2023.*
>
> *[5] Effects of graph convolutions in multi-layer networks. In ICLR, 2023.*

---

> > ### Comment · Reviewer_sSgQ · 2023-11-22
> >
> > Thanks for the response. The authors have addressed most of my concerns and I'll keep my positive rating for this paper.

---

> > > ### Author Response · Authors · 2023-11-23
> > > **Replying to Reviewer sSgQ**
> > >
> > > Dear reviewer, thank you again for your support and your time in reviewing our work.

---

### Official Review · Reviewer_Ngh1 · 2023-10-31

**Soundness:** 3 good
**Presentation:** 3 good
**Contribution:** 3 good
**Rating:** 6
**Confidence:** 5

**Summary:**

This paper aims to provide a theoretical understanding of the effects of different heterophily patterns on GNNs. The novel Heterophilous Stochastic Block Models (HSBM) is introduced to integrate graph convolution (GC) operations into fully connected networks. The paper offers insights into the factors determining the separability gains when applying GC operations and the influence of topological noise. The study validates the theory with experiments on synthetic and real-world data.

**Strengths:**

* The paper provides solid theoretical insights into the interplay of heterophily patterns and GNNs. It addresses the challenge of understanding the impacts of heterophily on classification tasks.

* The HSBM is a unique proposition that can accommodate a variety of heterophily patterns. This is likely to be beneficial to the research community aiming to handle diverse graph structures.

* The authors present an in-depth analysis of the separability gains in the context of GC operations, shedding light on factors like Euclidean distance of the neighborhood distributions and averaged node degree.

**Weaknesses:**

* While the authors have acknowledged the limitations of their model and analysis, they are significant. The assumptions on which the theory is based, especially Gaussian node features and independence among node features and edges, might not hold true in many real-world scenarios. The future directions suggested by the authors, including extending the analysis to other feature distributions and considering dependencies among nodes and edges, are crucial for the model's broader applicability.

* Discussion on potential real-world applications or case studies where HSBM could be applied would enhance the paper's relevance and appeal to a broader audience.

**Questions:**

*  It would be beneficial for the authors to delve deeper into the implications of the assumptions they have made, providing justifications or scenarios where these assumptions are most likely to hold true.

*  The authors should consider elaborating on the experimental section, providing more detailed results, methodologies, and potential pitfalls or challenges. The benchmarks used in https://arxiv.org/abs/2302.11640 are suggested.

---

> ### Author Response · Authors · 2023-11-17
> **Response to Reviewer Ngh1**
>
> Dear reviewer, we sincerely appreciate your comments and valuable suggestions. We respond to your detailed comments as below:
>
> > Comment 1: It would be beneficial for the authors to delve deeper into the implications of the assumptions they have made, providing justifications or scenarios where these assumptions are most likely to hold true.
>
> **Response to Comment 1:** Thank you for your suggestion. Accordingly, we have added additional discussion regarding our assumptions in the revised manuscript, as below.
>
> Assumption 1 claims that nodes within different classes possess similar averaged degrees. This assumption enables us to approximate $\bar{D}\approx \bar{D}_k$, enhancing the clarity of our results. It's worth noting that many literatures also implicitly or explicitly assume that $\bar{D} = \bar{D}_k$ [1,2,3]. For example, the averaged degrees of different classes are the same when utilizing the CSBM model to generate graphs, with each class exhibiting the same number of nodes. Our assumption allows the averaged degrees of different classes to possess the same order of magnitude, thereby covering broader regimes. Besides, we observe this property is frequently satisfied in many real-world datasets, as confirmed in Appendix C.1. Assumption 2 states that the graph is not too sparse and enable some numerical properties of graphs, e.g., the number of nodes within each class and node degree concentrate to their expectations. This assumption is similar to those in the random graph model literatures, e.g., [3,4].
>
> *[1] A non-asymptotic analysis of over-smoothing in graph neural networks. In ICLR, 2023.*
>
> *[2] Effects of graph convolutions in multi-layer networks. In ICLR, 2023.*
>
> *[3] Graph convolution for semi-supervised classification: Improved linear separability and out-of-distribution generalization. In ICML, 2021.*
>
> *[4] Learnable Graph Convolutional Attention Networks. In ICLR, 2023.*
>
> > Comment 2: The authors should consider elaborating on the experimental section, providing more detailed results, methodologies, and potential pitfalls or challenges. The benchmarks used in https://arxiv.org/abs/2302.11640 are suggested.
>
> **Response to Comment 2:** Thank you for your comment. We note that in the initial version of the manuscript, we provided comprehensive details regarding the experimental settings, results, and analysis for both synthetic data and real-world data in Appendix B and Appendix C. Still, according to your suggestion, we have employed Amazon-ratings and Workers in [5] in the revised manuscript. Additionally, we have also employed two large-scale heterophilious benchmarks, i.e., Arxiv-year and Snap-patents in [6], in the revised manuscript, to verify our theory. Arxiv-year has 169,343 nodes and 1,166,243 edges, while Snap-patents has 2,923,922 nodes and 13,975,791 edges.
>
> As detailed in Appendix C.3 (in the revised manuscript), the results on these datasets further validate the effectiveness of our method. Specifically, according to their separability gains, calculated by our method, the Workers dataset possesses a good heterophily pattern, while the other three possess a mixed heterophily pattern. Besides, for these datasets, the differences in the confusion matrix between GCN and MLP align with the corresponding separability gains. These observations, especially in large-scale real-world datasets, further validate the effectiveness of our theory.
>
> *[5] A Critical Look at Evaluation of GNNs Under Heterophily: Are We Really Making Progress? In ICLR, 2023.*
>
> *[6] New Benchmarks for Learning on Non-Homophilous Graphs. In WWW Workshop, 2021.*

---

> > ### Comment · Reviewer_Ngh1 · 2023-11-23
> >
> > The authors have addressed all of my previous concerns, and I commend their efforts in refining the manuscript. I would like to offer one minor suggestion: the recent work on heterophilous graph learning: https://arxiv.org/abs/2306.04265 should be added as another related work.
> >
> > Apart from this, I have no further comments and maintain my positive evaluation of the paper.

---

> > > ### Author Response · Authors · 2023-11-23
> > > **Replying to Reviewer Ngh1**
> > >
> > > Dear reviewer,
> > >
> > > We sincerely appreciate your support for our work. We are glad that our response has addressed all of your concerns. We would be very grateful if you could consider increasing your score in light of our revised version and response.
> > >
> > > According to your further suggestion, we have added the related work [1] in our revised manuscript.
> > >
> > > Thank you again for your time.
> > >
> > > Sincerely, authors.
> > >
> > > [1] Permutation Equivariant Graph Framelets for Heterophilous Graph Learning. arXiv preprint arXiv: 2306.04265.

---

### Official Review · Reviewer_Wh1i · 2023-10-31

**Soundness:** 2 fair
**Presentation:** 3 good
**Contribution:** 2 fair
**Rating:** 3
**Confidence:** 5

**Summary:**

This paper delves into the theoretical effects of various heterophily patterns on GNNs by integrating graph convolution (GC) operations into fully connected networks using the newly proposed Heterophilous Stochastic Block Models (HSBM). The study reveals three key findings:
1) Applying a GC operation results in separability gains determined by the Euclidean distance of neighborhood distributions and the square root of node degree.
2) Topological noise negatively affects separability.
3) Using multiple GC operations reveals that separability gains are linked to the normalized distance of l-powered neighborhood distributions, suggesting sustained node separability as l approaches infinity across different conditions.

**Strengths:**

Theoretical Insight into Heterophily Patterns: A significant strength of this work lies in its analytical treatment of the heterophily pattern. By dissecting and delving into the theoretical facets of heterophily, the paper offers an elevated understanding of datasets characterized by this pattern. This examination provides a foundational framework for future investigations into heterophily-rich datasets.

**Weaknesses:**

Limited Novelty: While the paper takes steps toward analyzing the heterophily problem, the extent of innovation remains somewhat constrained. I found that the conclusions and the main method they use are very similar to this ICML 21 paper: Graph Convolution for Semi-Supervised Classification: Improved Linear Separability and Out-of-Distribution Generalization. In that paper, they also analyze the SBM model and show that graph convolution extends the regime in which the data is linearly separable by a factor of roughly 1/sqrt(D), where D is the expected degree of a node, which was rediscovered by this paper. Besides, the paper Two Sides of the Same Coin: Heterophily and Oversmoothing in Graph Convolutional Neural Networks also mentions that the node degrees play an important role in the heterophily and oversmoothing problem.

Regarding Assumption 1: I noticed that the conclusion derived for Assumption 1 is predominantly based on statistical information from a small subset of datasets. Could you elucidate how you might validate this assumption? Alternatively, is there a possibility to derive this conclusion from a broader statistical perspective or from an extended range of datasets?

Empirical Validation with Real-World Data: In the presented real-world experiments, it seems that only a limited number of datasets were employed to validate the proposed heterophily pattern concepts. Considering the importance of comprehensive empirical validation, would it be feasible to provide additional experiments across a wider range of datasets to reinforce your conclusions?

**Questions:**

See the weakness above.

---

> ### Author Response · Authors · 2023-11-17
> **Response to Reviewer Wh1i (Part One)**
>
> Dear reviewer, we sincerely thank you for your comments and your questions. We respond to your detailed comments as below:
>
> > Comment 1: Limited Novelty: While the paper takes steps toward analyzing the heterophily problem, the extent of innovation remains somewhat constrained. I found that the conclusions and the main method they use are very similar to this ICML 21 paper: Graph Convolution for Semi-Supervised Classification: Improved Linear Separability and Out-of-Distribution Generalization. In that paper, they also analyze the SBM model and show that graph convolution extends the regime in which the data is linearly separable by a factor of roughly 1/sqrt(D), where D is the expected degree of a node, which was rediscovered by this paper. Besides, the paper Two Sides of the Same Coin: Heterophily and Oversmoothing in Graph Convolutional Neural Networks also mentions that the node degrees play an important role in the heterophily and oversmoothing problem.
>
> **Response to Comment 1:** Thank you for your comment. We would like to emphasize that our work is still novel, compared to the mentioned two related papers [1,2] for the following reasons.
>
> 1. [1] analyzes the improvement of Graph Convolutions on CSBM models via the distance of intra-class node features, while our work delves into the impacts of different heterophily patterns. Although the results in [1] and our work both possess 1/sqrt(D), our results further reveal the impact of different heterophily patterns.
>
> 2. The results in [2] reveal the impact of the relative node degree and homophily ratio in the binary node classifications. However, the relative node degree is a node-level metric. It evaluates the node degree compared to its neighbors' degrees, which is not the same as our D, i.e., the averaged node degrees. Therefore, our results are not similar to theirs.
>
> 3. Both [1] and [2] focus on binary node classifications, which is too simple to analyze the impact of complicated heterophily patterns. Real-world networks typically involve multiple node classes, and nodes within different classes exhibit diverse connection preferences across various classes. Our work focuses on these broader scenarios and analyzes the impact of complicated heterophily patterns in the multi-class node classifications, which have not been theoretically explored, to the best of our knowledge.
>
> 4. The above contents exclusively discuss the novelty of our Theorem 1, compared to the related work. It is worth noting that our paper also proposes a new data model and an analytical framework for multi-class classification, analyzes the impacts of node-wise topological noise, and analyzes the separability when applying multiple GNN layers. We believe that these contributions should also be taken into account.
>
> *[1] Graph convolution for semi-supervised classification: Improved linear separability and out-of-distribution generalization. In ICML, 2021.*
>
> *[2] Two Sides of the Same Coin: Heterophily and Over-smoothing in Graph Convolutional Neural Networks. In ICDM, 2022.*
>
> > Comment 2: Regarding Assumption 1: I noticed that the conclusion derived for Assumption 1 is predominantly based on statistical information from a small subset of datasets. Could you elucidate how you might validate this assumption? Alternatively, is there a possibility to derive this conclusion from a broader statistical perspective or from an extended range of datasets?
>
>
> **Response to Comment 2:** Thank you for your comment. Accordingly, we have added additional discussion regarding Assumption 1 in the revised manuscript, as below.
>
> Assumption 1 claims that nodes within different classes possess similar averaged degrees. This assumption enables us to approximate $\bar{D}\approx \bar{D}_k$, enhancing the clarity of our results. It's worth noting that many literatures also implicitly or explicitly assume that $\bar{D} = \bar{D}_k$ [1,3,4,5]. For example, the averaged degrees of different classes are the same when utilizing the CSBM model to generate graphs, with each class exhibiting the same number of nodes. Our assumption allows the averaged degrees of different classes to possess the same order of magnitude, thereby covering broader regimes. Besides, as stated in the initial version of the manuscript, we observed this property is frequently satisfied in many real-world datasets, as confirmed in Appendix C.1.
>
>
> [3] A non-asymptotic analysis of over-smoothing in graph neural networks. In ICLR, 2023.
>
> [4] Effects of graph convolutions in multi-layer networks. In ICLR, 2023.
>
> [5] Graph Attention Retrospective. arXiv preprint arXiv:2202.13060.

---

> > ### Comment · Reviewer_Wh1i · 2023-11-22
> > **Response**
> >
> > I have read the authors' rebuttal. I still feel the main theoretical results and the proposed CSBM model highly resemble the theoretical analysis in [1] and [2], especially [1]. Though the authors argued for multi-class analysis and the impact of heterophily patterns, these contributions seem to be very incremental for me. I think the theoretical analysis in [1] and [2] can also be easily adapted to multi-class settings and can also be used to analyze heterophily patterns. Besides theoretical analysis tools, I don't feel this paper provides new conclusions/insights beyond [1-3].
> >
> > 1. Graph convolution for semi-supervised classification: Improved linear separability and out-of-distribution generalization. In ICML, 2021
> > 2. Is Homophily a Necessity for Graph Neural Networks? In ICLR 2021.
> > 3. Two Sides of the Same Coin: Heterophily and Over-smoothing in Graph Convolutional Neural Networks. In ICDM, 2022.

---

> ### Author Response · Authors · 2023-11-17
> **Response to Reviewer Wh1i (Part Two)**
>
> > Comment 3: Empirical Validation with Real-World Data: In the presented real-world experiments, it seems that only a limited number of datasets were employed to validate the proposed heterophily pattern concepts. Considering the importance of comprehensive empirical validation, would it be feasible to provide additional experiments across a wider range of datasets to reinforce your conclusions?
>
> **Response to Comment 3:** Thank you for your comment. It is worth noting that when verifying our theory on real-world datasets, we visualize and compare the theoretical separability gains with the elements in the empirical confusion matrix. Therefore, for convenience and clarity, it may be better that the employed real-world graphs would not possess too many node categories, which limits the number of proper heterophilious datasets.
>
> Still, according to your suggestions, we have utilized additional real-world datasets in the revised manuscript. Specifically, we have employed three additional heterophilious datasets, namely Workers[6], Arxiv-year[7], and Snap-patents[7]. The Workers dataset has two node categories, while the Arxiv-year and Snap-patents datasets have five node categories each. Besides, the employed Arxiv-year and Snap-patents are large-scale heterophilious graphs. Arxiv-year has 169,343 nodes and 1,166,243 edges, while Snap-patents has 2,923,922 nodes and 13,975,791 edges.
>
> As detailed in Appendix C.3 (in the revised manuscript), the results on these additional datasets further validate the effectiveness of our method. Specifically, according to their separability gains, calculated by our method, the Workers possess a good heterophily pattern, while the other two possess a mixed heterophily pattern. Besides, for these datasets, the differences in the confusion matrix between GCN and MLP align with the corresponding separability gains. These observations, especially in large-scale real-world datasets, further validate the effectiveness of our theory.
>
> *[6] A Critical Look at Evaluation of GNNs Under Heterophily: Are We Really Making Progress? In ICLR, 2023.*
>
> *[7] New Benchmarks for Learning on Non-Homophilous Graphs. In WWW Workshop, 2021.*

---

> ### Author Response · Authors · 2023-11-23
> **Replying to Reviewer Wh1i (Part Three)**
>
> Dear reviewer, thank you for your reply. We would like to further clarify the novelty of our work, compared to your mentioned related work, as below.
>
> **Firstly, we would like to emphasize the results of our Theorem 2 possess new insights compared to [1,2].**
> Our intention in Theorem 2 is to analyze the impact of diverse heterophily patterns; we believe both [1,2] do not provide sufficient theoretical understandings.
> Although the results in both [1] and our Theorem 2 possess a factor of $1/\sqrt{D}$, our Theorem 2 aims at understanding the impacts of different heterophily. Our theorem 2 first reveals the factor of Euclidean distance of neighborhood distributions, and implies the impact of imbalanced class distribution. Although [2] conducts a preliminary investigation experimentally, our results are theoretical, precise, and quantitative.
>
> **Secondly, we would like to emphasize that the results of our Theorem 2 are not incremental, compared to the multi-class settings of [1]/[2].** Although the results in [1,2] could be generalized to multi-class settings, their generalized results could not consider diverse heterophily patterns, due to the limitation of their analytical frameworks and CSBM data models. Our analysis is based on HSBM. It accommodates multiple classes, diverse heterophily patterns, node-wise topological noise, different class-wise averaged degrees, and imbalanced class distribution. Thus, our data model and analytical framework are much more general. We believe our analysis is more applicable to real-world data.
>
> **Thirdly, we would like to emphasize the contributions of our HSBM models, analytical framework (theorem 1), analysis of the impacts of node-wise topological noise (theorem 3), and analysis of the impacts of applying multiple GNN layers (theorem 4).**
> For example, our Theorem 4 suggests that the nodes still possess separability in various regimes, even when over-smoothing occurs, which is a new insight for the over-smoothing problem. We believe that these contributions should also be taken into account.
>
> [1] Graph convolution for semi-supervised classification: Improved linear separability and out-of-distribution generalization. In ICML, 2021.
>
> [2] Is Homophily a Necessity for Graph Neural Networks? In ICLR, 2021.

---

### Official Review · Reviewer_fZmP · 2023-11-02

**Soundness:** 3 good
**Presentation:** 3 good
**Contribution:** 3 good
**Rating:** 6
**Confidence:** 5

**Summary:**

This paper focuses on understanding the heterophily patterns for Graph Neural Networks (GNNs). It proposes an HSBM graph generation model to generate diverse heterophilious patterns and then analyzes the relationship with the performance of GNN and MLPs.

**Strengths:**

The paper provides a detailed analysis of GNNs using the HSBM graph generation model. The theoretical analysis and empirical evaluations demonstrate the effectiveness of the analysis.

**Weaknesses:**

1. The paper lacks comparisons with existing methods. Previous methods [1,2] have provided analysis on heterophily from the perspective of node degree and neighborhood distribution. It would be beneficial to include discussions and comparisons from the standpoint of assumptions, graph generation, and results.

2. The paper does not provide suggestions for learning on heterophilous graphs. The analysis of heterophilous patterns in relation to model performance is provided, but it would be helpful to offer some guidance on model design when dealing with these graphs. Alternatively, the paper could explain why existing methods succeed on these graphs.

3. The paper could benefit from evaluations on larger-scale graphs. In section 5, the evaluations on the HSBM model are too simplistic to demonstrate its effectiveness, and it would be advantageous to increase the graph size.

[1] Two Sides of the Same Coin: Heterophily and Oversmoothing in Graph Convolutional Neural Networks. ICDM 2022
[2] Is Heterophily A Real Nightmare For Graph Neural Networks To Do Node Classification?

**Questions:**

No more

---

> ### Author Response · Authors · 2023-11-17
> **Response to Reviewer fZmP (Part One)**
>
> Dear reviewer, we sincerely appreciate your comments and valuable suggestions. We respond to your detailed comments as below:
>
> > Comment 1: The paper lacks comparisons with existing methods. Previous methods [1,2] have provided analysis on heterophily from the perspective of node degree and neighborhood distribution. It would be beneficial to include discussions and comparisons from the standpoint of assumptions, graph generation, and results.
>
> **Response to Comment 1:** Thank you for your reminder about these two related works [1,2]. Accordingly, we have discussed them in our revised manuscript.
>
> [1] utilizes the relative node degree and homophily ratio to predict the performance of the binary node classification task. However, neither the homophily ratio nor the binary classification task is sufficient to analyze the impact of complicated heterophily patterns. Our work focuses on more general cases and analyzes the impact of complicated heterophily patterns in the multi-class node classifications, which have not been explored to the best of our knowledge. Besides, it seems that [2] is the preprint version of [3], which had already been discussed in Section 1 (Introduction) in the previous manuscript. According to your suggestion, we have added the comparison with [1] in Section 1 (Introduction) in the initial version of the manuscript.
>
> *[1] Two Sides of the Same Coin: Heterophily and Oversmoothing in Graph Convolutional Neural Networks. In ICDM, 2022.*
>
> *[2] Is Heterophily A Real Nightmare For Graph Neural Networks To Do Node Classification?*
>
> *[3] Revisiting heterophily for graph neural networks. In NeurIPS, 2022.*
>
> > Comment 2: The paper does not provide suggestions for learning on heterophilous graphs. The analysis of heterophilous patterns in relation to model performance is provided, but it would be helpful to offer some guidance on model design when dealing with these graphs. Alternatively, the paper could explain why existing methods succeed on these graphs.
>
> **Response to Comment 2:** Thank you for your suggestion. As stated in the initial version of our manuscript, our work introduces a fresh perspective on heterophily and over-smoothing, coupled with a novel GNN analytical framework, which may further boost the development of innovative methods. We did not provide specific suggestion for learning methods on heterophilious graphs in the initial version. According to your suggestion, we have bridged our theoretical results to the model design of heterophily-specific GNNs, in the revised manuscript (Appendix F). Several possible suggestions and insights are as follows.
>
> Firstly, according to Theorem 2, different GNN aggregators may successfully classify different subsets of nodes. Therefore, when learning on graphs with heterophily, it may be beneficial to adaptively select specific aggregators for individual nodes, e.g. [3,4]. Besides, this result can also explain why the combination of ego- and neighbor-embeddings can establish a stable and competitive baseline across diverse heterophily patterns [5,6].
>
> Secondly, according to Theorem 3, the inconsistency in neighborhood distribution poses a potential challenge when handling heterophilious graphs. Therefore, it may be beneficial to mitigate the influence of topological noise within each class, probably through the design of an attention scheme.
>
> Thirdly, according to Theorem 4, GNN with varying numbers of layers may successfully classify different subsets of nodes. Therefore, it may be beneficial to explore information from high-order neighborhoods, e.g., [5].
>
> Besides, we believe that our theoretical results may provide additional inspiration to the community for designing heterophily-specific GNNs, enabling them to successfully handle graphs with diverse heterophily patterns.
>
> *[3] Revisiting heterophily for graph neural networks. In NeurIPS, 2022.*
>
> *[4] Learnable graph convolutional attention networks. In ICLR, 2023.*
>
> *[5] Beyond Homophily in Graph Neural Networks: Current Limitations and Effective Designs. In NIPS, 2020.*
>
> *[6] A critical look at the evaluation of gnns under heterophily: are we really making progress? In ICLR, 2023.*

---

> ### Author Response · Authors · 2023-11-17
> **Response to Reviewer fZmP (Part Two)**
>
> > Comment 3: The paper could benefit from evaluations on larger-scale graphs. In section 5, the evaluations on the HSBM model are too simplistic to demonstrate its effectiveness, and it would be advantageous to increase the graph size.
>
> **Response to Comment 3:** Thank you for your suggestion. Accordingly, we have added additional large-scale datasets, including synthetic data and real-world data, in the revised manuscript, to validate the effectiveness of our theoretical results.
>
>
> For the synthetic data, we have employed a series of large-scale synthetic graphs generated from our HSBM model in the revised manuscript (Appendix B.3). Similarly, we adopt the settings in Sec. 5.1, except for assigning the number of nodes as 100,000. Since the averaged node degree for each class is still 25, the expected number of edges in these large-scale graphs is 2,500,000. The results are provided in Fig. 10 in the revised manuscript, revealing that the results of the large-scale synthetic graphs are similar to those in Fig. 2(a), where the number of nodes is only 1000. This observation further validates the effectiveness of our theory.
>
> For the real-world data, we have utilized two multi-class large-scale heterophilious graphs, i.e., Arxiv-year and Snap-patents [7], in the revised manuscript. Arxiv-year possesses 169,343 nodes and 1,166,243 edges, while Snap-patents possesses 2,923,922 nodes and 13,975,791 edges. As detailed in Appendix C.3 (in the revised manuscript), the results on these datasets further
> validate the effectiveness of our method. Specifically, according to their separability gains, calculated by our method, they both exhibit a mixed heterophily pattern. Their differences in the confusion matrix between GCN and MLP align with the corresponding separability gains. This observation in the large-scale real-world datasets further validates the effectiveness of our theory.
>
> *[7] New Benchmarks for Learning on Non-Homophilous Graphs. In WWW Workshop, 2021.*

---

### Official Review · Reviewer_SYCa · 2023-11-08

**Soundness:** 3 good
**Presentation:** 3 good
**Contribution:** 3 good
**Rating:** 8
**Confidence:** 3

**Summary:**

This paper presents theoretical understandings of the impacts of different heterophily patterns for GNNs by incorporating the graph convolution (GC) operations into fully connected networks via the proposed Heterophilous Stochastic Block Models (HSBM).

**Strengths:**

Good paper with high originality and high quality. The paper is well-written and makes some new contributions to the relevant field.

**Weaknesses:**

More large-scale datasets are suggested to be added.

**Questions:**

I do not have any questions.

---

> ### Author Response · Authors · 2023-11-17
> **Response to Reviewer SYCa**
>
> Dear reviewer, we sincerely appreciate your support and valuable suggestions.
>
> According to your suggestion, we have added additional large-scale datasets, including synthetic data and real-world data, in the revised manuscript, to validate the effectiveness of our theoretical results.
>
> For the synthetic data, we have employed a series of large-scale synthetic graphs generated from our HSBM model in the revised manuscript (Appendix B.3). Similarly, we adopt the settings in Sec. 5.1, except for assigning the number of nodes as 100,000. Since the averaged node degree for each class is still 25, the expected number of edges in these large-scale graphs is 2,500,000. The results are provided in Fig. 10 in the revised manuscript, revealing that the results of the large-scale synthetic graphs are similar to those in Fig. 2(a), where the number of nodes is only 1000. This observation further validates the effectiveness of our theory.
>
> For the real-world data, we have utilized two multi-class large-scale heterophilious graphs, i.e., Arxiv-year and Snap-patents [1], in the revised manuscript. Arxiv-year possesses 169,343 nodes and 1,166,243 edges, while Snap-patents possesses 2,923,922 nodes and 13,975,791 edges. As detailed in Appendix C.3 (in the revised manuscript), the results on these datasets further
> validate the effectiveness of our method. Specifically, according to their separability gains, calculated by our method, they both exhibit a mixed heterophily pattern. Their differences in the confusion matrix between GCN and MLP align with the corresponding separability gains. This observation in the large-scale real-world datasets further validates the effectiveness of our theory.
>
> *[1] New Benchmarks for Learning on Non-Homophilous Graphs. In WWW Workshop, 2021.*

---

### Author Response · Authors · 2023-11-22
**Response Letter to Reviewers**

Dear reviewers,

We appreciate your thorough review of our paper and the insightful comments you've raised. We have diligently addressed all the questions you provided. If there are any additional questions or points you would like us to clarify, please feel free to let us know. We sincerely hope our responses meet your expectations.

Thank you for your time and consideration.

Sincerely, authors.

---

### Meta-Review · Area_Chair_dD6x · 2023-12-07

**Metareview:**

This paper presents a theoretical understanding of the impacts of different heterophily patterns on Graph Neural Networks (GNNs) by incorporating graph convolution operations into fully connected networks through Heterophilous Stochastic Block Models, revealing that the impact of heterophily on classification needs to be evaluated alongside the averaged node degree and that multiple graph convolution operations still maintain node separability in a wide range of regimes.

While most questions have been addressed during the rebuttal and discussion, the biggest concern is on the novelty in developing the theoretical results.

- The main conclusions (theorem 2) and theoretical tools very resemble those in the ICML 21 paper. In particular, they both find the separability changes with 1/sqrt (D) and they both use stochastic block model to analyze. Though authors claim heterophily patterns and multi-class as their innovation and distinctions, however they can be easily extended by prior work. The prior work (ICML 21) analyzes the general setting and can be utilized to analyze the heterophily setting simply by setting the intra-class probability to be smaller than the inter-class probability.

- Besides, the (ICLR 22) paper more explicitly uses the HSBM for theoretical analysis in hereophily setting. This paper’s theoretical framework does not change things fundamentally.

Graph convolution for semi-supervised classification: Improved linear separability and out-of-distribution generalization. In ICML, 2021.

Is Homophily a Necessity for Graph Neural Networks? In ICLR, 2022.

**Justification For Why Not Higher Score:**

The novelty in developing the theoretical results is limited.  This paper’s theoretical framework does not offer new insights to existing results.

**Justification For Why Not Lower Score:**

N/A.

---

### Decision · Program_Chairs · 2024-01-16

Reject